

# LPJ-GUESS/LSMv1.0: A next generation Land Surface Model with high ecological realism

David Martín Belda[1], Peter Anthoni[1], David Wårlind[2], Stefan Olin[2], Guy Schurgers[3,5], Jing Tang[2,3,4], Benjamin Smith[2,6], and Almut Arneth[1]

[1]Karlsruhe Institute of Technology KIT, Institute of Meteorology and Climate Research - Atmospheric Environmental Research (IMK-IFU), 82467 Garmisch-Partenkirchen, Germany
[2]University of Lund, Department of Physical Geography and Ecosystem Science, 223 62, Lund, Sweden
[3]Terrestrial Ecology Section, Department of Biology, Universitetsparken 15, DK-2100, Copenhagen Ø, Denmark
[4]Center for Permafrost (CENPERM), University of Copenhagen, Øster Voldgade 10, DK-1350, Copenhagen K, Denmark
[5]Department of Geosciences and Natural Resource Management, University of Copenhagen, Copenhagen, Denmark
[6]Hawkesbury Institute for the Environment, Western Sydney University, Richmond, NSW, Australia.

**Correspondence:** David Martín Belda (david.belda@kit.edu)

**Abstract.** Land biosphere processes are of central importance to the climate system. Specifically, biological processes interact with the atmosphere through a variety of feedback loops that modulate energy, water and $CO_2$ fluxes between the land surface and the atmosphere across a wide range of temporal and spatial scales. Human land use and land cover modification add a further level of complexity to land-atmosphere interactions. Dynamic Global Vegetation Models (DGVMs) attempt to capture

these land surface processes, and are increasingly incorporated into Earth System Models (ESMs), which makes it possible to study the coupled dynamics of the land-biosphere and the climate. In this work we describe a number of modifications to the LPJ-GUESS DGVM, aimed at enabling direct integration into an ESM. These include energy balance closure, the introduction of a sub-daily time step, a new radiative transfer scheme, and improved soil physics. The implemented modifications allow the model (LPJ-GUESS/LSM) to simulate the diurnal exchange of energy, water and CO2 between the land-ecosystem and

the atmosphere. A site-based evaluation against FLUXNET2015 data shows reasonable agreement between observed and modeled sensible and latent heat fluxes. Differences in predicted ecosystem function between standard LPJ-GUESS and LPJ-GUESS/LSM vary across land cover types, but the emergent ecosystem composition and structure are consistent between the two versions. We find that the choice of stomatal conductance model has a major impact on the model's predictions. The new LSM implementation described in this work lays the foundation for using the well established LPJ-GUESS DGVM as

an alternative LSM in coupled land-biosphere-climate studies, where an accurate representation of ecosystem processes is essential.

## 1 Introduction

The land surface is of central importance in the climate system, as feedbacks between the land-biosphere and the atmosphere impact climate across a wide range of temporal and spatial scales (Pitman, 2003). Biological processes affected by climate

variations can feed back into the climate by modulating the fluxes of energy and water between vegetation and the atmosphere





(Guo et al., 2006; Green et al., 2017). For example, the early, strong greening caused by the warming climate can enhance evapotranspiration, which may result in a seasonal cooling effect or in an amplification of heat waves, depending on regional characteristics and water availability (Peñuelas et al., 2009; Lorenz et al., 2013). On decadal time scales, decreased vegetation cover caused by reduced rainfall can further decrease local precipitation (Zeng et al., 1999). Large scale shifts in vegetation
cover in response to climate change can affect global and regional climate by altering the radiation and water budgets (O'ishi and Abe-Ouchi, 2009; Levis et al., 2000; Wramneby et al., 2010; Wu et al., 2021).

The climate and the biosphere are also coupled biogeochemically through the carbon cycle (Luo, 2007). Increased atmospheric carbon dioxide ($CO_2$) concentration promotes vegetation growth through $CO_2$ fertilization, which increases plant $CO_2$ absorption from the atmosphere. However, higher temperatures caused by a higher atmospheric $CO_2$ concentration enhance
the release of $CO_2$ from respiration (Cramer et al., 2001; Piao et al., 2013). Other important effects relate to extreme events (Zscheischler et al., 2014), disturbances (Kurz et al., 2008; Metsaranta et al., 2010) or interaction with the nitrogen cycle (Arneth et al., 2010; Lamarque et al., 2013; Ciais et al., 2014).

Of particular importance is the added complexity arising from land use and land-cover change. Conversion of forests into cropland or grassland increases surface albedo, which may promote surface cooling in temperate latitudes (e.g. Noblet-
Ducoudré et al., 2012), but is also a significant contributor to anthropogenic $CO_2$ emissions (Arneth et al., 2017; Le Quéré et al., 2018). Observations and model studies suggest that historical land cover changes over the industrial era have had a minor net impact on the climate system at the global scale, but regional effects are large (Brovkin et al., 2004; Pongratz et al., 2010; Christidis et al., 2013). Further complexity arises from the interaction between land use change and the water cycle (e.g. Narisma and Pitman, 2003; Kumar et al., 2013; Lawrence and Vandecar, 2015), atmospheric circulation (Swann et al., 2012;
Wu et al., 2017) and from atmospheric teleconnections (Werth and Avissar, 2002; Medvigy et al., 2013).

Incorporating DGVMs into ESMs allows the interactions between the biosphere and the rest of the climate system to be studied on the long time scales of vegetation dynamics and biogeochemical and biogeographical responses (Quillet et al., 2010; Fisher et al., 2018). There is considerable uncertainty regarding the carbon cycle response to future climate warming scenarios (Friedlingstein et al., 2006, 2014; Jones et al., 2013), much of which has been attributed to uncertainty in the representation
of land surface processes (Huntingford et al., 2009; Booth et al., 2012; Friend et al., 2014) and differences between the global circulation models (GCMs) used to make such projections (Ahlström et al., 2013, 2017; Schurgers et al., 2018). Improved representations of land-biosphere processes and land use change in ESMs are therefore essential to constrain climate change projections (Friend et al., 2014) and thus to support the assessment of mitigation and adaptation strategies.

DGVMs are frequently coupled to ESMs through an intermediary Land Surface Model (LSM), which facilitates the sub-
daily energy, water and gas exchange calculations (e.g. Bonan et al., 2003; Krinner et al., 2005; Smith et al., 2011; Döscher et al., 2021). This approach can, however, entail inconsistencies between the DGVM and the LSM, such as the use of different time steps and temperatures in photosynthetic calculations, duplicated or inconsistent soil water tracking, or different characterization of vegetation types. In this work we modify the LPJ-GUESS DGVM (Smith et al., 2001, 2014) to enable coupling with an atmospheric model without the need for a mediating LSM. LPJ-GUESS simulates a wide range of land-biosphere processes,
including vegetation growth, establishment and mortality, plant functional type (PFT) competition, disturbances, wildfires, and





land use change. This model has been used in a broad range of applications, including coupled biosphere-atmosphere regional (Wramneby et al., 2010; Smith et al., 2011; Zhang et al., 2014, 2018; Wu et al., 2016, 2021) and global (Weiss et al., 2014; Alessandri et al., 2017; Forrest et al., 2020; Döscher et al., 2021) studies, and undergoes active development and evaluation, which makes it a suitable choice to study climate-biosphere interactions.

Coupling LPJ-GUESS with an atmospheric model requires it to be able to calculate diurnal energy and water exchange rates between plant canopies and the atmosphere. To achieve this, we introduced several major modifications to LPJ-GUESS v4.0, namely: (a) a new radiative transfer scheme, capable of representing direct and diffuse light, as well as treating sunlit and shaded leaves separately; (b) representation of the energy balance on a sub-daily time step; and (c) an improved representation of heat and water transport in the soil. Section 2 describes these modifications in detail. A site-based evaluation of the modeled

fluxes against eddy covariance data is presented in Section 3. Finally, the work is discussed and summarized in Section 4.

## 2  Model description

### 2.1  LPJ-GUESS

LPJ-GUESS (Smith et al., 2001, 2014) is a process-based model of vegetation dynamics and ecosystem biogeochemistry and water cycling that incorporates tree demographic processes and competition for light, space and soil resources among co-

occurring PFTs. Capturing establishment, growth and death of individuals allows to better represent the mechanisms underlying competition, population and community structural dynamics, carbon assimilation and ecosystem carbon turnover (Smith et al., 2001; Wolf et al., 2011a). In LPJ-GUESS, natural vegetation is represented as a co-occurring mixture of different PFTs, divided into age classes or *cohorts*, in a modeled area or *patch*. New cohorts can establish in the patch when climatic conditions are within PFT-prescribed bioclimatic limits, and compete with other cohorts for light, water and soil nitrogen. Each cohort

assimilates atmospheric $CO_2$ at a rate, updated daily in the standard model, that depends on the amount of photosynthetically active radiation (PAR) it absorbs, water availability, temperature, and the maximum rate of carboxylation, $V_{max}$. The maximum rate of carboxylation is estimated under the assumption that plants redistribute leaf nitrogen content across the canopy so as to maximize net assimilation at the canopy level (Haxeltine and Prentice, 1996), and is limited by nitrogen availability (Smith et al., 2014). The yearly assimilated carbon is distributed between roots, leaves and, in the case of woody PFTs, sapwood,

according to a set of PFT-specific allometric constraints. The phenological status of the cohorts (for summergreen and raingreen PFTs) is updated daily. Population dynamics (establishment and mortality) and non-fire related disturbances are modeled as stochastic processes, influenced by environmental factors, vegetation structure, growth and competition. Disturbances occur recurrently and destroy all vegetation in a patch, restarting the successional cycle. Wildfires are modeled explicitly (Thonicke et al., 2001). At any given geographical location (*gridcell*), a number of replicate patches with independent successional

histories are simulated.

LPJ-GUESS can represent managed land (croplands, pastures/rangelands and managed forest) and land use change (Lindeskog et al., 2013, 2021; Olin et al., 2015). Each gridcell contains different land cover types or *stands*, which are updated every simulation year (for example, to simulate conversion of forest to cropland). Croplands are represented as single PFT





stands, distinguishing various rainfed and irrigated crop functional types. In pasture stands only grassy PFTs are allowed to es-

tablish. Simulated land management practices include crop sowing, irrigation, fertilization, harvest, rotation and abandonment, and pasture grazing.

## 2.2 Model modifications

Figure 1 shows a comparison of the daily loop in standard LPJ-GUESS and in the new LSM implementation. In both versions, phenology and soil organic matter dynamics are calculated daily, and carbon allocation (growth) and vegetation dynamics

(establishment, mortality and disturbance) are computed at the end of every simulation year.

Radiative transfer in standard LPJ-GUESS is based on Beer's law (Monsi and Saeki, 1953, 2005). The canopy is divided in vertical layers, each absorbing a fraction of the PAR let through by the layer above. The PAR absorbed by each layer is then split among cohorts according to their share of leaf area index (LAI) in that layer. In this way, taller cohorts have access to more PAR and shade the lower layers of the canopy. Daily unstressed values of $V_{\max}$ and canopy conductance $g_{\mathrm{pot}}$ are first

computed for each cohort assuming well watered conditions. The actual evapotranspiration rate in the patch is then calculated as the minimum of a potential rate, determined by atmospheric conditions and $g_{\mathrm{pot}}$, and a supply rate, which depends on the amount of soil water available for uptake and the vegetation rooting profiles. For each cohort, the model calculates a daily assimilation rate that is consistent with its contribution to the total patch evapotranspiration. The soil column consists of a top layer of $0.5\,\mathrm{m}$ and a bottom layer of $1\,\mathrm{m}$ thickness. The fraction of root matter in each soil layer is PFT-specific. Soil water

content is updated taking into account daily precipitation, interception, percolation between the two layers, evapotranspiration and runoff. Daily soil temperature is calculated as a dampened, lagged oscillation around the annual mean of the forcing air temperature, as described in Sitch et al. (2003). More detailed descriptions of the radiative transfer, evapotranspiration, assimilation and soil organic matter calculations can be found in the supplement to Smith et al. (2001), Smith et al. (2014), and references therein. The hydrology scheme is described in Gerten et al. (2004).

In the LSM implementation, radiative transfer, energy balance, assimilation and soil heat and water transport are all solved on a subdaily basis. Based on Dai et al. (2004), each cohort is conceptualized as two big leaves, representing its sunlit and shaded parts. Sunlit leaves receive direct solar radiation and diffuse radiation, while shaded leaves receive only diffuse radiation. The total LAI for each cohort is calculated dynamically by LPJ-GUESS. A stem area index (SAI) was added to account for the impact of stems and branches in the energy balance and radiative transfer calculations. Whole canopy leaf area and plant area

(PAI) indices are obtained by aggregating over cohorts $i$:

$$\mathrm{LAI_c} = \sum_i \mathrm{LAI}^{(i)}; \tag{1}$$

$$\mathrm{PAI_c} = \sum_i \left[ \mathrm{LAI}^{(i)} + \mathrm{SAI}^{(i)} \right]. \tag{2}$$

Based on Kucharik et al. (1998), we set the stem area index of woody PFTs to $10\%$ of their leaf area index at full leaf coverage. Grasses do not have stem area index. The sunlit and shaded fractions of leaf and plant area indices are updated in the radiative

transfer routine on a subdaily basis (Sec. 2.2.2).





We replaced the original two-layer soil column with a new profile consisting of 9 layers. The top 4 layers have thicknesses of 7, 10, 13 and 20 cm, in order of increasing depth, and correspond to the top soil layer in the original soil column. The next three layers have thicknesses of 30, 30 and 40 cm, and correspond to the original bottom layer. These 7 layers constitute the rooting zone. The new water transport scheme assumes, for simplicity, free gravitational drainage at the bottom of the soil

column, which can lead to excessive soil dryness during dry periods. Additionally, no heat flux is allowed through the bottom boundary, an approximation better met at higher soil depths. In order to mitigate spurious effects derived from this choice of boundary conditions, we extended the soil column with two additional layers of 50 and 100 cm, reaching a total depth of 3 m.

The sunlit and shaded leaves of each cohort have different assimilation rates and stomatal conductances. The temperatures of sunlit and shaded leaves are different, but common to all the cohorts in the patch. The vertical layering of the canopy is kept

in the radiation calculations, but the new scheme distinguishes direct and diffuse radiation and two separate wavebands (visible and near infrared). Infrared radiation does not contribute to photosynthetic assimilation, but needs to be accounted for in the energy balance calculations. A separate treatment of diffuse and direct radiation allows to resolve sunlit and shaded leaves. This approach has been shown to lead to predictions of fluxes of energy, water and $CO_2$ that are comparable in accuracy to those made by more complex, and considerably more computationally expensive, multi-layered canopy models (Wang and

Leuning, 1998).

Each cohort exchanges sensible and latent heat with a common canopy air space, which in turn exchanges sensible and latent heat with the atmosphere (Fig. 2). Assimilation and evapotranspiration are calculated consistently in the energy balance routine. Daily averages of absorbed PAR are used to update $V_{\max}$ for each cohort. The new energy balance, radiative transfer and soil physics calculations are detailed in sections 2.2.1 through 2.2.5.

### 2.2.1 Energy balance

The energy balance of the patch canopy is described by the following equations (e.g., Bonan, 2008):

$$S_{\text{sun}} = L_{\text{sun}} + H_{\text{sun}} + \lambda E_{\text{sun}}; \tag{3}$$
$$S_{\text{sha}} = L_{\text{sha}} + H_{\text{sha}} + \lambda E_{\text{sha}}, \tag{4}$$

where the $S$ terms are absorbed shortwave radiation, $L$ is net emitted longwave radiation, $H$ is sensible heat flux towards the

canopy air space, $E$ is water vapor flux towards the canopy air space, and $\lambda$ is latent heat of vaporization (here taken constant; $\lambda = 2.44 \cdot 10^6 \, \text{J kg}^{-1} \, ^\circ\text{C}^{-1}$). The subindices 'sun' and 'sha' refer to the sunlit and shaded parts of the canopy. The calculation of the shortwave and longwave radiation terms is detailed in Secs. 2.2.2 and 2.2.3.

The sensible heat flux from the sunlit part of the canopy to the canopy airspace is formulated as:

$$H_{\text{sun}} = -2\text{PAI}_{\text{c,sun}}\rho c_P g_{\text{b}}(T_{\text{ca}} - T_{\text{sun}}), \tag{5}$$

where $\text{PAI}_{\text{c,sun}}$ is the plant area index of the sunlit canopy, $\rho$ is air density, $c_P$ is the specific heat of air at constant pressure, $g_{\text{b}}$ is average leaf boundary layer conductance (e.g., Bonan, 2008), $T_{\text{ca}}$ is the temperature of the canopy air, and $T_{\text{sun}}$ is the temperature of the sunlit canopy. The factor 2 expresses heat loss from both sides of the leaf and stem elements.



The latent heat flux from the sunlit part of the canopy to the canopy air is:

$$\lambda E_{\mathrm{sun}} = -\rho \lambda g_{\mathrm{w,sun}}[q_{\mathrm{ca}} - q^*(T_{\mathrm{sun}})], \tag{6}$$

where $q_{\mathrm{ca}}$ is the specific humidity of the canopy air, $q^*(T_{\mathrm{sun}})$ is the specific humidity inside the stomatal cavity, taken to be the saturated humidity at the leaf temperature, and $g_{\mathrm{w,sun}}$ is the conductance for water vapor flux from the sunlit part of the canopy to the canopy air space. The latter is calculated as a weighted average of the contributions from evaporation of intercepted water and transpiration through the stomata (Appendix A):

$$g_{\mathrm{w,sun}} = f_{\mathrm{wet}} \eta_{\mathrm{sun}} \mathrm{PAI}_{\mathrm{c,sun}} g_{\mathrm{b}}$$

$$+ (1 - f_{\mathrm{wet}} \eta_{\mathrm{sun}}) \sum_i \mathrm{LAI}_{\mathrm{sun}}^{(i)} \frac{g_{\mathrm{s,sun}}^{(i)} g_{\mathrm{b}}}{g_{\mathrm{s,sun}}^{(i)} + g_{\mathrm{b}}}. \tag{7}$$

In this equation $f_{\mathrm{wet}}$ is the wet fraction of the canopy, the factor $\eta_{\mathrm{sun}}$ limits evaporation to the amount of intercepted water present in the canopy, and $\mathrm{LAI}_{\mathrm{sun}}^{(i)}$ is the leaf area index of the sunlit part of cohort $i$. The stomatal conductance of cohort $i$, $g_{\mathrm{s,sun}}^{(i)}$, is related to its net photosynthetic rate through a semiempirical model. We implemented two selectable stomatal conductance models: the Ball-Berry model (Ball et al., 1987) and the Medlyn model (Medlyn et al., 2011).

Equations analogous to Eqs. (5) through (7) apply to the shaded part of the canopy.

The energy balance equation for the ground surface is:

$$S_{\mathrm{g}} = L_{\mathrm{g}} + H_{\mathrm{g}} + \lambda E_{\mathrm{g}} + G, \tag{8}$$

where $G$ is heat conducted into the ground. The sensible heat from the ground surface to the canopy air space is:

$$H_{\mathrm{g}} = -\rho c_P g_{\mathrm{ab}}(T_{\mathrm{ca}} - T_{\mathrm{g}}), \tag{9}$$

where $g_{\mathrm{ab}}$ is the aerodynamic conductance from the ground surface to the canopy air space, which is calculated following Sakaguchi and Zeng (2009). The latent heat from the ground surface to the canopy air is given by:

$$\lambda E_{\mathrm{g}} = -\rho \lambda \frac{g_{\mathrm{surf}} g_{\mathrm{ab}}}{g_{\mathrm{surf}} + g_{\mathrm{ab}}}[q_{\mathrm{ca}} - \alpha q^*(T_{\mathrm{g}})], \tag{10}$$

where we used the model of Sakaguchi and Zeng (2009) for the surface conductance $g_{\mathrm{surf}}$, and $\alpha q^*(T_{\mathrm{g}})$ is the air specific humidity at the ground surface (Philip, 1957).

The heat conducted into the ground is calculated as:

$$G = -\kappa_{\mathrm{s}}^{(1)} \frac{T_{\mathrm{s}}^{(1)} - T_{\mathrm{g}}}{\Delta z^{(1)}/2}, \tag{11}$$

where $\kappa_{\mathrm{s}}^{(1)}$, $T_{\mathrm{s}}^{(1)}$, and $\Delta z^{(1)}$ are, respectively, the thermal conductivity, the temperature, and the thickness of the top soil layer.

The following two equations express conservation of latent and sensible heat:

$$H^{\uparrow} = H_{\mathrm{sun}} + H_{\mathrm{sha}} + H_{\mathrm{g}}; \tag{12}$$

$$\lambda E^{\uparrow} = \lambda E_{\mathrm{sun}} + \lambda E_{\mathrm{sha}} + \lambda E_{\mathrm{g}}, \tag{13}$$





where $H^\uparrow$ and $\lambda E^\uparrow$ are respectively the sensible and latent heat fluxes into the atmosphere, given by

$$H^\uparrow = -\rho c_P g_{\mathrm{aa}}(T_{\mathrm{atm}} - T_{\mathrm{ca}}); \tag{14}$$

$$\lambda E^\uparrow = -\rho \lambda g_{\mathrm{aa}}(q_{\mathrm{atm}} - q_{\mathrm{ca}}). \tag{15}$$

Here, $T_{\mathrm{atm}}$ and $q_{\mathrm{atm}}$ are the temperature and specific humidity of the air at the atmospheric reference level, and $g_{\mathrm{aa}}$ is the
aerodynamic conductance above the canopy. The latter is calculated by applying Monin-Obukov similarity theory, which requires knowledge of the surface roughness length, $z_0$, and the zero plane displacement, $z_{\mathrm{d}}$. These are calculated as a function of the canopy plant area index, $\mathrm{PAI_c}$, and the canopy height, $h_c$, according to the model of Raupach (1994, 1995):

$$\frac{z_{\mathrm{d}}}{h_c} = 1 - \frac{1 - \exp\left(-\sqrt{7.5\mathrm{PAI_c}}\right)}{\sqrt{7.5\mathrm{PAI_c}}}; \tag{16}$$

$$\frac{z_0}{h_c} = \left(1 - \frac{z_{\mathrm{d}}}{h_c}\right) \exp\left(-\frac{k}{\beta} + 0.193\right), \tag{17}$$

where $k = 0.4$ is the von Karman constant, and $\beta = \min\left(\sqrt{0.003 + 0.15\mathrm{PAI_c}}, 0.3\right)$. Canopy height is calculated, following Forrest et al. (2020), as an average of cohort heights weighted by their foliar projective cover (FPC).

Equations (3), (4) and (8), subject to constraints (12) and (13), are solved simultaneously every time step with a multidimensional Newton-Rhapson method (e.g. Press, 2003).

### 2.2.2   Shortwave radiative transfer

We adapted the two big leaf model of Dai et al. (2004), based on the two-stream model of Dickinson (1983); Sellers (1985), to LPJ-GUESS's multiple cohort, vertically layered canopy. This approach considers direct solar radiation and diffuse atmospheric radiation separately. The intensity of the direct solar radiation beam in the canopy decreases exponentially with cumulative plant area index $P$ (measured from the top of the canopy, increasing downwards) (Monsi and Saeki, 1953, 2005):

$$I_{\mathrm{D}}^\downarrow(P) = I_{\mathrm{D0}}^\downarrow e^{-kP}, \tag{18}$$

where $I_{\mathrm{D0}}^\downarrow$ is incoming direct solar radiation and $k$ is the direct beam extinction coefficient. The profile of diffuse radiation in the canopy results from the multiple scattering and backscattering of incoming radiation by leaves and stems. Corrected profiles (normalized by incoming radiation) of scattered direct beam ($\hat{I}_{\mathrm{b}}^\uparrow$ and $\hat{I}_{\mathrm{b}}^\downarrow$) and scattered atmospheric diffuse radiation ($\hat{I}_{\mathrm{a}}^\uparrow$ and $\hat{I}_{\mathrm{a}}^\downarrow$) are given in analytic form in Dai et al. (2004) (the arrows indicate the direction of propagation).

The direct beam radiation absorbed in a canopy layer $l$ between $P$ and $P + \Delta_l P$ is calculated as the fraction of the decrease
in direct beam intensity in that layer that is not scattered:

$$S_{\mathrm{D}}^{(l)} = -(1 - \omega)\Delta_l I_{\mathrm{D}}^\downarrow, \tag{19}$$

where $\omega$ is the direct beam scattering coefficient, and $\Delta_l$ denotes change across layer $l$. The diffuse radiation absorbed in the layer is the sum of the radiation from the direct beam that is scattered and reabsorbed in the layer and the contribution from the





diffuse beams:

$$S_{\mathrm{d}}^{(l)} = -\omega\Delta_l I_{\mathrm{D}}^{\downarrow}$$
$$+ I_{\mathrm{D0}}^{\downarrow}(\Delta_l \hat{I}_{\mathrm{b}}^{\uparrow} - \Delta_l \hat{I}_{\mathrm{b}}^{\downarrow}) + I_{\mathrm{d0}}^{\downarrow}(\Delta_l \hat{I}_{\mathrm{a}}^{\uparrow} - \Delta_l \hat{I}_{\mathrm{a}}^{\downarrow}), \tag{20}$$

where $I_{\mathrm{d0}}^{\downarrow}$ is incoming atmospheric diffuse radiation. The radiation absorbed by the sunlit and shaded parts of this layer is

$$S_{\mathrm{sun}}^{(l)} = S_{\mathrm{D}}^{(l)} + f_{\mathrm{sun}}^{(l)} S_{\mathrm{d}}^{(l)}; \tag{21}$$
$$S_{\mathrm{sha}}^{(l)} = f_{\mathrm{sha}}^{(l)} S_{\mathrm{d}}^{(l)}, \tag{22}$$

where the sunlit and shaded fractions of the layer are given by

$$f_{\mathrm{sun}}^{(l)} = -\frac{e^{-k(P+\Delta_l P)} - e^{-kP}}{k\Delta_l P}; \tag{23}$$
$$f_{\mathrm{sha}}^{(l)} = 1 - f_{\mathrm{sun}}^{(l)}. \tag{24}$$

The total amount of shortwave radiation absorbed by the sunlit and shaded parts of the canopy is obtained by summing over layers:

$$S_{\mathrm{sun}} = \sum_l S_{\mathrm{sun}}^{(l)}; \tag{25}$$
$$S_{\mathrm{sha}} = \sum_l S_{\mathrm{sha}}^{(l)}. \tag{26}$$

The shortwave radiation absorbed by the ground surface is calculated as the difference between the downward and upward beams at $P = \mathrm{PAI_c}$,

$$S_{\mathrm{g}} = I_{\mathrm{D}}^{\downarrow}(\mathrm{PAI_c}) + I_{\mathrm{D0}}^{\downarrow}[\hat{I}_{\mathrm{b}}^{\downarrow}(\mathrm{PAI_c}) - \hat{I}_{\mathrm{b}}^{\uparrow}(\mathrm{PAI_c})]$$
$$+ I_{\mathrm{d0}}^{\downarrow}[\hat{I}_{\mathrm{a}}^{\downarrow}(\mathrm{PAI_c}) - \hat{I}_{\mathrm{a}}^{\uparrow}(\mathrm{PAI_c})]. \tag{27}$$

The shortwave radiation reflected back at the atmosphere is obtained by evaluating the upward beams at $P = 0$:

$$I^{\uparrow} = I_{\mathrm{b}}^{\uparrow}(0) + I_{\mathrm{a}}^{\uparrow}(0). \tag{28}$$

The optical elements in the canopy have different properties in the visible and near-infrared wave bands, so the equations above are applied separately to these two parts of the spectrum, and the contributions are summed to calculate total absorption. In this study, we set the optical properties of the canopy to the following values, regardless of PFT:

$$\alpha_{\mathrm{leaf,vis}} = 0.1; \ \alpha_{\mathrm{stem,vis}} = 0.16; \tag{29}$$
$$\tau_{\mathrm{leaf,vis}} = 0.05; \ \tau_{\mathrm{stem,vis}} = 0.001; \tag{30}$$
$$\alpha_{\mathrm{leaf,nir}} = 0.45; \ \alpha_{\mathrm{stem,nir}} = 0.39; \tag{31}$$
$$\tau_{\mathrm{leaf,nir}} = 0.25; \ \tau_{\mathrm{stem,nir}} = 0.001, \tag{32}$$





where $\alpha$ is absorptivity, $\tau$ is transmissivity, 'vis' refers to visible radiation and 'nir' refers to near-infrared. These values were taken from the ones assigned to tropical trees by Oleson et al. (2004). Soil optical properties are from the dataset prepared by Lawrence and Chase (2007).

The PAR absorbed by the sunlit leaves of a cohort $i$ is obtained as the sum over layers of the absorbed visible radiation weighted by the cohort's fractional leaf area index:

$$\mathrm{PAR}_{\mathrm{sun}}^{(i)} = \sum_l S_{\mathrm{sun,vis}}^{(l)} \frac{\mathrm{LAI}^{(i,l)}}{\mathrm{PAI}^{(l)}}. \tag{33}$$

The sunlit leaf and plant area indices of cohort $i$ are obtained by aggregating over layers:

$$\mathrm{LAI}_{\mathrm{sun}}^{(i)} = \sum_l f_{\mathrm{sun}}^{(l)} \mathrm{LAI}^{(i,l)}; \tag{34}$$

$$\mathrm{PAI}_{\mathrm{sun}}^{(i)} = \sum_l f_{\mathrm{sun}}^{(l)} \left[ \mathrm{LAI}^{(i,l)} + \mathrm{SAI}^{(i,l)} \right]. \tag{35}$$

The sunlit plant area index for the whole canopy is calculated by summing over cohorts:

$$\mathrm{PAI}_{\mathrm{sun,c}} = \sum_i \mathrm{PAI}_{\mathrm{sun}}^{(i)}. \tag{36}$$

Equations analogous to Eqs. (33) through (36) apply to the shaded parts of the canopy.

### 2.2.3 Longwave radiative transfer

The longwave radiation emitted by the sunlit part of the canopy is (Dai et al., 2004):

$$L_{\mathrm{sun}} = \gamma_{\mathrm{sun}} (2\sigma T_{\mathrm{sun}}^4 - L^\downarrow - \sigma T_{\mathrm{g}}^4); \tag{37}$$

where $\sigma$ is the Stefan-Boltzmann constant, $L^\downarrow$ is the incoming atmospheric longwave radiation, $T_{\mathrm{sun}}$, and $T_{\mathrm{g}}$ are expressed in Kelvin, and

$$\gamma_{\mathrm{sun}} = \left( 1 - e^{-\mathrm{PAI_c}} \right) \frac{\mathrm{PAI}_{\mathrm{sun,c}}}{\mathrm{PAI_c}}. \tag{38}$$

The thermal emissivity of plants and soil is assumed to be 1. The net emission of longwave radiation by the shaded part of the canopy is described by analogous equations.

The longwave radiation emitted by the ground surface is

$$\begin{aligned} L_{\mathrm{g}} = \sigma T_{\mathrm{g}}^4 &- \gamma_{\mathrm{sun}} \sigma T_{\mathrm{sun}}^4 - \gamma_{\mathrm{sha}} \sigma T_{\mathrm{sha}}^4 \\ &+ (1 - \gamma_{\mathrm{sun}} - \gamma_{\mathrm{sha}}) L^\downarrow. \end{aligned} \tag{39}$$

The bulk longwave radiation emitted by the land surface toward the atmosphere is:

$$\begin{aligned} L^\uparrow = \gamma_{\mathrm{sun}} \sigma T_{\mathrm{sun}}^4 &+ \gamma_{\mathrm{sha}} \sigma T_{\mathrm{sha}}^4 \\ 260 \quad &+ (1 - \gamma_{\mathrm{sun}} - \gamma_{\mathrm{sha}}) \sigma T_{\mathrm{g}}^4. \end{aligned} \tag{40}$$





### 2.2.4 Assimilation and stomatal conductance

Photosynthetic assimilation is now calculated within the subdaily energy balance routine. A net photosynthetic rate is computed for the sunlit and shaded leaves of each cohort separately. These rates are related to stomatal conductance through a semi-empirical model. As noted above, we implemented two selectable models. In the Ball-Berry model (Ball et al., 1987), stomatal conductance depends linearly on net assimilation and the fractional humidity at the leaf surface $h_s$, and inversely on $CO_2$ concentration at the leaf surface, $c_s$. The stomatal conductance for sunlit leaves of cohort $i$ is:

$$g_{s,sun}^{(i)} = g_{min} + g_{1,BB} \frac{A_{n,sun}^{(i)} h_{s,sun}}{c_{s,sun}}, \tag{41}$$

where $A_{n,sun}^{(i)}$ is the net photosynthetic rate per unit leaf area, $g_{min}$ is a minimum stomatal conductance, and $g_1$ is a PFT-specific parameter. The Medlyn model (Medlyn et al., 2011) is derived from the assumption that stomata optimize $CO_2$ uptake while minimizing water loss. In this model, stomatal conductance depends inversely on the square root of the vapor pressure deficit at the leaf surface, $D_s$. The stomatal conductance for sunlit leaves of cohort $i$ is:

$$g_{s,sun}^{(i)} = g_{min} + 1.6 \left( 1 + \frac{g_{1,Med}}{\sqrt{D_{s,sun}}} \right) \frac{A_{n,sun}^{(i)}}{c_{s,sun}}. \tag{42}$$

Values of the parameters $g_{1,BB}$ and $g_{1,Med}$ for specific PFTs were obtained following Sellers et al. (1996) for the Ball-Berry model and De Kauwe et al. (2015) for the Medlyn model. Figure 3 shows the different behaviour of the stomatal conductance models as a function of $D_s$.

For a given cohort $i$, the total photosynthetic rate is limited by the maximum rate of carboxylation, $V_{max}^{(i)}$, which depends linearly on the total amount of daily absorbed photosynthetic active radiation, $PAR_{day}^{(i)}$ (Haxeltine and Prentice, 1996):

$$V_{max,day}^{(i)} = f_v(T_{leaf,dt}^{(i)}, \cdots) \times PAR_{day}^{(i)}. \tag{43}$$

In this equation, $V_{max,day}^{(i)}$ is expressed per unit patch area. This potential rate is calculated by LPJ-GUESS for every cohort daily (Fig 1). The slope of the relationship, $f_v$, depends on environmental factors, including temperature and leaf nitrogen content. The daytime-averaged leaf temperature, $T_{leaf,dt}^{(i)}$, is weighted by the daily averaged fractions of sunlit and shaded leaves for cohort $i$:

$$T_{leaf,dt}^{(i)} = \frac{1}{n_{dt}} \sum_{dt} \frac{PAI_{sun}^{(i)} T_{sun} + PAI_{sha}^{(i)} T_{sha}}{PAI^{(i)}}, \tag{44}$$

where $n_{dt}$ is the number of daytime subdaily periods.

Separating the contributions to daily absorbed PAR from sunlit and shaded leaves, maximum carboxylation rates for the sunlit and shaded parts of the cohort are estimated as:

$$V_{max,sun,day}^{(i)} = f_v(T_{leaf,dt}^{(i)}, \cdots) \times PAR_{sun,day}^{(i)}$$
$$V_{max,sha,day}^{(i)} = f_v(T_{leaf,dt}^{(i)}, \cdots) \times PAR_{sha,day}^{(i)} \tag{45}$$





where $\mathrm{PAR}_{\mathrm{sun,day}}^{(i)}$ and $\mathrm{PAR}_{\mathrm{sha,day}}^{(i)}$ are the total daily PAR absorbed by the sunlit and shaded leaves of cohort $i$, respectively.

Combining Eqs. (43) and (45) yields, for sunlit leaves:

$$V_{\mathrm{max,sun,day}}^{(i)} = V_{\mathrm{max,day}}^{(i)} \frac{\mathrm{PAR}_{\mathrm{sun,day}}^{(i)}}{\mathrm{PAR}_{\mathrm{day}}^{(i)}}. \tag{46}$$

The maximum carboxylation rate per unit leaf area is then calculated as:

$$V_{\mathrm{max,sun,leaf}}^{(i)} = 86400^{-1} \beta \frac{V_{\mathrm{sun,day}}^{(i)}}{\mathrm{LAI}_{\mathrm{sun,dt}}^{(i)}}, \tag{47}$$

where $\mathrm{LAI}_{\mathrm{sun,dt}}^{(i)}$ is the daily-averaged sunlit LAI of cohort $i$, and we have introduced a factor $\beta$ to limit the photosynthetic rate

under conditions of water stress. The prefactor $86400^{-1}$ converts the rate from $\mathrm{day}^{-1}$ to $\mathrm{s}^{-1}$. Analogous equations apply to shaded leaves.

The water stress factor $\beta$ is formulated as a sum over soil layers of a water uptake function weighed by a PFT-specific vertical rooting profile:

$$\beta = \sum_j r^{(j)} W_{\mathrm{av}}^{(j)}, \tag{48}$$

where $r^{(j)}$ is the fraction of roots in soil layer $j$. In order to study the impact of the $\beta$ factor on the model predictions, we implemented four different options for the water uptake function $W_{\mathrm{av}}^{(j)}$. In the Noah type (Niu et al., 2011), $W_{\mathrm{av}}^{(j)}$ decreases linearly in each soil layer with volumetric water content $\theta^{(j)}$ down to the wilting point:

$$W_{\mathrm{av}}^{(j)} = \frac{\theta^{(j)} - \theta_{\mathrm{wilt}}}{\theta_{\mathrm{fc}} - \theta_{\mathrm{wilt}}}, \tag{49}$$

where $\theta_{\mathrm{wilt}}$ and $\theta_{\mathrm{fc}}$ are volumetric water content at wilting point and field capacity respectively. In LPJ-GUESS, the wilting

point is assumed to be at a matric potential of $\psi_{\mathrm{wilt}} = -45\mathrm{m}$, and the corresponding soil water content is calculated following Prentice et al. (1992).

The CLM type water uptake function is formulated in terms of matric potential (Oleson et al., 2004):

$$W_{\mathrm{av}}^{(j)} = \frac{\psi_{\mathrm{wilt,CLM}} - \psi^{(j)}}{\psi_{\mathrm{wilt,CLM}} - \psi_{\mathrm{sat}}}, \tag{50}$$

where $\psi^{(j)}$ is the matric potential of layer $j$, $\psi_{\mathrm{sat}}$ is the matric potential at saturation, and $\psi_{\mathrm{wilt,CLM}}$ is the matric potential

at wilting point, set to $-150\mathrm{m}$. In this case, the water uptake response is flatter than in the Noah-type case when the soil is wet, and decreases more steeply when the soil gets drier. We also implemented a modified version of the CLM-type uptake function, with the same functional form but using LPJ-GUESS's $-45\mathrm{m}$ wilting matric potential instead of CLM's $-150\mathrm{m}$.

The SSiB type water uptake function is:

$$W_{\mathrm{av}}^{(j)} = 1 - e^{-c_2 \ln\left[\psi_{\mathrm{wilt}}/\psi^{(j)}\right]}, \tag{51}$$





where the parameter $c_2$ depends on PFT, and takes values between $4.36$ and $6.37$ (Xue et al., 1991). In this study, we set $c_2$ to a fixed value of $5.8$ for all PFTs, which results in high $\beta$ values in most of the water availability range, and a steep decrease when approaching the wilting point.

Figure 4 shows the behavior of the different formulations of $W_{av}^{(j)}$ as a function of volumetric water content.

### 2.2.5  Soil physics

In standard LPJ-GUESS, soil temperature is used in calculations related to ecosystem respiration and nitrogen cycling, while soil water content influences plant water uptake and evapotranspiration. Both quantities affect soil organic matter decomposition rates.

Soil temperature $T_s$ is now calculated by solving the heat transport equation:

$$\frac{\partial T_s}{\partial t} = -\frac{1}{c_h}\frac{\partial}{\partial z}\left(\kappa_s \frac{\partial T_s}{\partial z}\right), \tag{52}$$

where $c_h(z)$ and $\kappa_s(z)$ are soil heat capacity and thermal conductivity respectively. The top boundary condition is given by the heat flux into the ground, $G$, calculated in the energy balance routine (Eq. 11). Heat flow through the bottom boundary is neglected. Thermal conductivity is calculated following the method of Johansen (1975, 1977). Soil heat capacity is computed as a weighted sum of the heat capacities of the dry soil, which depends on texture, and water (de Vries, 1963).

Vertical water transport in the soil column is described by the Richards equation (Richards, 1931), which can be expressed

in the following form:

$$\frac{\partial \theta}{\partial t} = \frac{\partial}{\partial z}\left[\lambda_w \frac{\partial \theta}{\partial z} - \gamma_w\right] + S_\theta(z). \tag{53}$$

Here, $\theta$ is volumetric water content, $\lambda_w(\theta)$ is hydraulic diffusivity, $\gamma_w(\theta)$ is hydraulic conductivity, and $S_\theta(z)$ is a volumetric sink term that accounts for plant water uptake ($S_\theta \leq 0$). Hydraulic diffusivity and conductivity are calculated as a function of soil texture and soil water content by using the expressions derived by Clapp and Hornberger (1978) and Cosby et al. (1984).

Rain water that is not intercepted by the canopy infiltrates into the soil at a rate limited by the soil's infiltration capacity as given by the Green-Ampt equation (Green and Ampt, 1911). Free gravitational drainage is assumed at the bottom of the soil column.

Soil temperature, water content, ecosystem respiration, plant water uptake and evapotranspiration are calculated in the sub-daily loop. Equations (52) and (53) are solved with a Crank-Nicolson scheme (e.g. Press, 2003). Daily averages of water

content and temperature over the layers corresponding to the standard LPJ-GUESS top and bottom layers are then used as inputs to the original soil organic matter and nitrogen cycling routines.





## 3    Model verification and evaluation

### 3.1    Model verification

The revised model was verified by performing energy and water conservation tests. At any given time step, the energy conser-
vation error per unit time and per unit patch area, $\Delta u_{\mathrm{err}}$, is calculated as:

$$\Delta u_{\mathrm{err}} = S^{\downarrow} + L^{\downarrow} - \langle L^{\uparrow} + H^{\uparrow} + \lambda E^{\uparrow} + \Delta u_{\mathrm{soil}} \rangle, \tag{54}$$

where $\langle \cdot \rangle$ indicates an average over patches, and $\Delta u_{\mathrm{soil}}$ is the rate of change of energy stored in the soil column per unit patch
area $(\mathrm{J\,m^{-2}\,s^{-1}})$. The latter is calculated as:

$$\Delta u_{\mathrm{soil}} = \frac{1}{\Delta t} \sum_j c_{\mathrm{h}}^{(j)} \Delta z^{(j)} T_{\mathrm{s}}^{(j)}, \tag{55}$$

where $\Delta t$ is the time step in seconds, and $c_{\mathrm{h}}^{(j)}$, $\Delta z^{(j)}$ and $T_{\mathrm{s}}^{(j)}$ are, respectively, the heat capacity, thickness and temperature
of soil layer $j$. Figure 5 (upper panel) shows the frequency of the energy conservation error relative to the energy input to the
system (i.e., the total incoming irradiance, $S^{\downarrow} + L^{\downarrow}$). The vast majority of the time steps ($\sim 98.4\%$) the error is smaller than is
$0.25\%$ of the incoming radiation. Errors larger than $1\%$ of the incoming radiation occur $\sim 0.014\%$ of time steps, and the error
is never larger than $1.75\%$ of the energy input.

The water conservation error is computed as:

$$\Delta w_{\mathrm{err}} = P - \langle R + E^{\uparrow} + \Delta w_{\mathrm{soil}} + \Delta w_{\mathrm{c}} \rangle, \tag{56}$$

where $P$ is precipitation, $R$ is runoff (including surface runoff and base flow), $E^{\uparrow}$ is evapotranspiration, $\Delta w_{\mathrm{soil}}$ is the change
in soil water content per unit patch area, per unit time, and $\Delta w_{\mathrm{c}}$ is the change in canopy water content. We found that the bulk
of the water conservation error is due to a generally small overestimation of canopy evaporation when the potential evaporation
at a given time step is substantially larger than the available canopy water. To assess the importance of this error in terms of
energy fluxes, we plotted it as a percentage of the energy input to the system (Fig. 5, lower panel). Water conservation errors
larger than $1\%$ of the total energy input occur $\sim 0.35\%$ of the time steps, and errors larger than $5\%$ of the energy input occur
$\sim 0.006\%$ of the time steps.

We therefore conclude that the magnitude of the errors in energy balance closure and water conservation is negligible the
vast majority of time steps. Relatively larger errors in water conservation due to overestimation of canopy evaporation are small
in terms of total energy input.

### 3.2    Evaluation setup

We evaluated the revised model by comparing hourly and monthly simulated fluxes of sensible and latent heat, and annual $CO_2$
fluxes, with flux tower measurements from 21 FLUXNET2015 (Pastorello et al., 2020) sites. The current version of the model
does not simulate snow or frozen soil water, so we restricted our study to sites where the air temperature remained above $0°C$





**Table 1.** Brief description of selected sites. The land cover classification was taken from the FLUXNET site description web pages. The reference level height is taken as the height of the measuring sensors above the canopy. A dash indicates that we weren't able to find an observed LAI value for the site.

| Site | Code | Land Cover | $z_{ref}$ (m) | LAI | Reference |
|------|------|-----------|---------------|-----|-----------|
| Emerald, Australia | AU-Emr | $C_3$ grassland | 6.2 | 0.7 | Schroder et al. (2014) |
| Amoladeras, Spain | ES-Amo | Open shrubland ($C_3$ grassland) | 3.5 | - | López-Ballesteros et al. (2017) |
| Daly River Cleared, Australia | AU-DaP | $C_4$ grassland | 5 | 1.5 | Hutley et al. (2011) |
| Sturt Plains, Australia | AU-Stp | $C_4$ grassland | 5 | 0.5 | Beringer et al. (2011) |
| Tchizalamou, Congo | CG-Tch | Savanna ($C_4$ grassland) | 3.8 | 2.0 | Merbold et al. (2009) |
| Sardinilla Pasture, Panama | PA-SPs | $C_4$ grassland | 2.91 | 5.4 | Wolf et al. (2011b) |
| Daly River Savanna, Australia | AU-DaS | Savanna | 5 | 1.5 | Hutley et al. (2011) |
| Dry River, Australia | AU-Dry | Savanna | 5 | 1.2 | Beringer et al. (2011) |
| Demokeya, Sudan | SD-Dem | Savanna | 4 | 0.9 | Ardö et al. (2008) |
| Adelaide River, Australia | AU-Ade | Woody savanna | 5 | 1.1 | Beringer et al. (2011) |
| Gingin, Australia | AU-Gin | Woody savanna | 7.8 | 0.9 | Beringer et al. (2016) |
| Howard Springs, Australia | AU-How | Woody savanna | 5 | 1.5 | Beringer et al. (2011) |
| Red Dirt Melon Farms, Australia | AU-RDF | Woody savanna | 5 | 1.6 | Bristow et al. (2016) |
| Robson Creek, Australia | AU-Rob | Evergreen broadleaf forest | 12 | 4.3 | Beringer et al. (2016) |
| Santarem-Km67, Brazil | BR-Sa1 | Evergreen broadleaf forest | 13 | 6.5 | Saleska et al. (2003) |
| Santarem-Km83, Brazil | BR-Sa3 | Evergreen broadleaf forest | 19 | 6.5 | Saleska et al. (2003) |
| Guyaflux, French Guiana | GF-Guy | Evergreen broadleaf forest | 23 | 5.9 | Bonal et al. (2008) |
| Ankasa, Ghana | GH-Ank | Evergreen broadleaf forest | 16 | - | Stefani et al. (2009) |
| Pasoh forest, Malaysia | MY-PSO | Evergreen broadleaf forest | 18 | 6.5 | Kosugi et al. (2008) |
| Sardinilla Plantation, Panama | PA-SPn | Deciduous broadleaf forest | 5 | 2.9 | Wolf et al. (2011b) |
| Mongu, Zambia | ZM-Mon | Deciduous broadleaf forest | 10 | 1.6 | Merbold et al. (2009) |

**Table 2.** Summary of the LPJ-GUESS/LSM simulations carried out. Simulations with different stomatal conductance schemes are arranged in columns: Ball-Berry (BB) and Medlyn (Med). Simulations with different water uptake function types are arranged in rows: NOAH, CLM, modified CLM and SSiB.

| | Ball-Berry | Medlyn |
|------|-----------|--------|
| NOAH | NOAH/BB | NOAH/Med |
| CLM | CLM/BB | CLM/Med |
| CLM (mod) | CLMm/BB | CLMm/Med |
| SSiB | SSiB/BB | SSiB/Med |



throughout the measuring period. We additionally discarded wetland sites, which require a more detailed representation of soil and ground water hydrology (Wania et al., 2009). A list of the selected sites is presented in Table 1. The location of the sites is represented on the world map in Fig. 6.

For each site, we ran 8 simulations, covering all possible configurations of the water uptake functions and stomatal con-

ductance schemes described in Sec. 2.2.4 (Table 2). We used the climate data collected at the tower sites to force the model. Half-hourly forcing data was converted to hourly averages, and we set a lower boundary of $10\%$ of the dataset median on the air humidity to correct for physically invalid negative values. Nitrogen deposition data is from Lamarque et al. (2013). Atmospheric $CO_2$ concentration data is from McGuire et al. (2001). Additionally, we ran a standard (non-LSM) LPJ-GUESS simulation to compare both model versions' predictions of monthly evapotranspiration and a number of ecosystem structure

and function variables. The number of replicate patches was set to $100$ in all the simulations to avoid spurious effects of the stochastic ecosystem processes on the modeled fluxes.

All natural PFTs were allowed to establish in forest and savanna sites. Since the focus of the model evaluation was placed on the predicted turbulent fluxes, we restricted the simulated PFTs to grassy types at sites classified as grasslands, which limits modeled surface roughness. This was also done for Spain-Amoladeras and Congo-Tchizalamou. Amoladeras is classified as

an open shrubland on the FLUXNET reference, but the vegetation is short and the most abundant species is *Machrocloa Tenacissima*, a type of grass (López-Ballesteros et al., 2017). Tchizalamou, which is classified as savanna, is actually a $C_4$ grassland (Merbold et al., 2009).

The simulations were spun up for a standard period of 500 years from a bare ground state to bring C and N soil and vegetation pools to near-equilibrium with the climate (see, e.g., Smith et al., 2014). During the spin-up phase, the site climate spanning the

whole measurement period was repeated cyclically, with interannual trends in air temperature removed, and the atmospheric $CO_2$ concentration was kept at the level of the first year of observations at each site.

### 3.3 Analysis

Half-hourly measured fluxes were converted to hourly averages for direct comparison with model outputs. Subdaily FLUXNET data are classified into four quality categories: 0 (measured), 1 (good quality gap fill), 2 (poor quality gap fill) and 3 (downscale

from ERA reanalysis data). In our analysis, we only used subdaily fluxes with a quality flag of 0 or 1. For monthly and annual fluxes, the quality flag varies between 0 and 1, and indicates the fraction of the subdaily values in that month/year whose quality is either 0 or 1. We only used monthly and annual fluxes with a quality flag equal to or greater than $0.75$. Following Stöckli et al. (2008), we further discarded fluxes with friction velocity $u^* < 0.2\,\mathrm{ms}^{-1}$ in order to avoid possibly biased eddy covariance measurements during periods of weak turbulence (Schroder et al., 2014).

To evaluate the agreement between measured and simulated turbulent heat fluxes at each site for all different model configurations we used standard statistical metrics: correlation coefficient ($r$), mean bias, and root mean square error (RMSE). We also considered the standard deviation of the modeled fluxes normalized by the standard deviation of the observed fluxes ($\sigma_{\mathrm{m}}/\sigma_{\mathrm{o}}$), which provides a measure of the agreement between observed and simulated variability.





## 3.4 Results

### 405  3.4.1  Annual and diurnal cycles of turbulent heat fluxes

Figure 7 shows examples of simulated and observed monthly averages of turbulent and latent heat fluxes over the course of a year at four sites: Gingin (AU-Gin), Daly River Savanna (AU-DaS), Santarem Km67 (BR-Sa1) and Guyaflux (GF-Guy). Examples of the monthly-averaged diurnal cycle for the same sites are shown in Figs. 8 and 9. We chose these sites and years to illustrate situations with varying degrees of agreement between simulations and measurements. The simulated fluxes are 410  from the run using the CLM-type water uptake function and the Medlyn model of stomatal conductance.

At the AU-Gin site, the shape of the annual cycles of latent and sensible heat is similar to the observed (Fig. 7, upper left). Sensible heat is largest at the beginning of the year, decreases steeply to its minimum around June-July, and starts increasing again around August. The simulation agrees very well with measurements most of the year, but overestimates sensible heat by $\sim 40\,\mathrm{Wm^{-2}}$ in the first two months. Observed latent heat dominates the turbulent exchange in the wet season (from May 415  to September). Simulated latent heat is overestimated by up to $\sim 25\,\mathrm{Wm^{-2}}$ during the wet season. The shift from larger sensible heat to larger latent heat in May is well captured in the simulation, but, due to the overestimation of latent heat, the shift back to larger sensible heat flux at the beginning of the dry season is delayed by about a month with respect to the observations. The average simulated diurnal cycle of sensible heat is overestimated in January, peaking at $\sim 700\,\mathrm{Wm^{-2}}$ (observed: $\sim 500\mathrm{Wm^{-2}}$), while it agrees very well with observations in May and September, both in terms of magnitude and 420  day-to-day variability.

At the AU-DaS site (Fig. 7, upper right panel), the shapes of measured and simulated annual cycles match relatively well at the beginning and the end of the year, but diverge substantially during the dry season. Simulated monthly averages of latent heat are $\sim 20\,\mathrm{Wm^{-2}}$ above measured values from March to May, and $\sim 30\,\mathrm{Wm^{-2}}$ below the measurements between August and October. The average simulated diurnal cycle peaks at $\sim 300\,\mathrm{Wm^{-2}}$ in May (observed: $\sim 175\,\mathrm{Wm^{-2}}$), and at $\sim 30\,\mathrm{Wm^{-2}}$ 425  in September (observed: $\sim 150\,\mathrm{Wm^{-2}}$; Fig. 8, lower half). This marked divergence from measured values happens in very dry periods, when the simulated soil moisture in the rooting zone drops close to the wilting point and there is not enough precipitation to replenish it until the start of the wet season. As a consequence, sensible heat is greatly overestimated. Simulated monthly averages rise sharply and peak at $\sim 120\text{--}140\,\mathrm{Wm^{-2}}$ from September to October, while measured values stay at $\sim 60\,\mathrm{Wm^{-2}}$ throughout the dry season. The average sensible heat diurnal cycle peaks at $\sim 530\,\mathrm{Wm^{-2}}$ in September, while 430  the observed average diurnal peak is slightly under $\sim 300\,\mathrm{Wm^{-2}}$ (Fig. 8).

Monthly averages of sensible and latent heat at the BR-Sa1 tropical rainforest site show little variability throughout the year (Fig. 7, lower left). Measured sensible heat flux stays at $\sim 20\,\mathrm{Wm^{-2}}$ for most of the year, and increases to $\sim 30\,\mathrm{Wm^{-2}}$ around August and September, when measured precipitation reaches its minimum. During this period, the soil retains enough moisture in the rooting zone to maintain average latent heat levels at $\sim 80\text{--}90\,\mathrm{Wm^{-2}}$. Sensible and latent heat fluxes are systematically 435  overestimated by the model by $\sim 10\text{--}20\,\mathrm{Wm^{-2}}$. Average sensible heat flux peaks daily between $\sim 170\text{--}230\,\mathrm{Wm^{-2}}$ (measured: $\sim 100\mathrm{Wm^{-2}}$). Latent heat flux peaks daily between $\sim 300\text{--}370\,\mathrm{Wm^{-2}}$ (measured: $\sim 280\text{--}320\mathrm{Wm^{-2}}$, Fig. 9).





**Table 3.** Model performance statistics for simulated hourly (left) and monthly (right) sensible heat fluxes for the CLM/BB and the CLM/Med simulations. Bold fonts indicate the model configuration that performed better. The mean and standard deviation of the observed fluxes ($\bar{H}_o$ and $\sigma_o$, respectively), shown for reference, are given in $\mathrm{Wm^{-2}}$. The RMSE and Bias have been normalized by the mean of the observed fluxes for easier cross-site comparison.

| Site | $\bar{H}_o\,(\sigma_o)$ | $r$ | | $\sigma_m/\sigma_o$ | | nRMSE | | nBias | | $\bar{H}_o\,(\sigma_o)$ | $r$ | | $\sigma_m/\sigma_o$ | | nRMSE | | nBias | |
|---|---|---|---|---|---|---|---|---|---|---|---|---|---|---|---|---|---|---|
| | | BB | Med | BB | Med | BB | Med | BB | Med | | BB | Med | BB | Med | BB | Med | BB | Med |
| AU-Emr | 110 (108) | **0.93** | 0.92 | 1.4 | **1.3** | 0.7 | **0.6** | **0.3** | **0.3** | 57 (20) | **0.89** | 0.85 | 1.4 | **1.3** | **0.4** | **0.4** | **0.3** | **0.3** |
| ES-Amo | 103 (133) | **0.96** | 0.94 | **1.5** | **1.5** | **0.8** | 0.9 | 0.1 | **0.0** | 67 (42) | **0.96** | **0.96** | **1.7** | 1.8 | **0.5** | 0.6 | **0.0** | **0.0** |
| AU-DaP | 124 (122) | **0.90** | **0.90** | 1.2 | 1.2 | 0.6 | **0.5** | 0.3 | **0.2** | 56 (29) | 0.78 | **0.85** | 0.8 | **1.0** | 0.5 | **0.3** | 0.3 | **0.2** |
| AU-Stp | 118 (121) | **0.96** | 0.94 | **1.3** | **1.3** | **0.5** | **0.5** | 0.3 | **0.2** | 66 (19) | **0.88** | **0.88** | **1.2** | 1.4 | **0.3** | **0.3** | 0.3 | **0.2** |
| CG-Tch | 98 (74) | **0.88** | 0.86 | 1.2 | **1.0** | **0.4** | **0.4** | 0.1 | **0.0** | 38 (11) | **0.62** | 0.55 | 0.5 | 0.4 | **0.2** | 0.3 | **0.0** | −0.1 |
| PA-SPs | 104 (96) | **0.89** | 0.85 | 1.3 | **1.2** | **0.7** | **0.7** | 0.5 | **0.3** | 26 (19) | **0.94** | 0.93 | 1.2 | **1.1** | 0.6 | **0.5** | 0.6 | **0.4** |
| AU-DaS | 86 (117) | **0.92** | 0.91 | **1.5** | **1.5** | 1.2 | **1.0** | 0.6 | **0.5** | 53 (16) | 0.73 | **0.77** | **1.6** | 2.1 | 0.7 | **0.6** | 0.6 | **0.4** |
| AU-Dry | 94 (117) | **0.94** | 0.92 | 1.7 | **1.5** | 1.5 | **1.1** | 1.0 | **0.7** | 56 (21) | **0.87** | 0.80 | **1.2** | 1.3 | 1.1 | **0.8** | 1.0 | **0.8** |
| SD-Dem | 78 (107) | **0.92** | 0.89 | 1.8 | **1.3** | 1.5 | **0.8** | 0.7 | **0.2** | 53 (16) | 0.03 | **0.12** | **0.7** | **0.7** | 0.7 | **0.4** | 0.6 | **0.2** |
| AU-Ade | 74 (107) | 0.91 | **0.92** | 1.6 | **1.5** | 1.3 | **1.0** | 0.6 | **0.4** | 50 (19) | 0.82 | **0.87** | **1.4** | 1.8 | 0.6 | **0.5** | 0.5 | **0.3** |
| AU-Gin | 111 (159) | **0.96** | **0.96** | **1.3** | **1.3** | **0.6** | **0.6** | 0.2 | **0.1** | 73 (44) | **0.99** | 0.98 | **1.3** | 1.4 | **0.3** | **0.3** | 0.2 | **0.1** |
| AU-How | 71 (102) | 0.88 | **0.90** | 1.7 | **1.6** | 1.6 | **1.3** | 0.9 | **0.7** | 41 (22) | 0.79 | **0.84** | **1.1** | 1.4 | 1.0 | **0.8** | 0.9 | **0.7** |
| AU-RDF | 109 (114) | **0.90** | 0.89 | 1.7 | **1.5** | 1.1 | **0.9** | 0.5 | **0.3** | 59 (14) | 0.10 | **0.37** | **1.4** | 1.9 | 0.6 | **0.5** | 0.4 | **0.2** |
| AU-Rob | 49 (96) | **0.93** | 0.92 | 1.7 | **1.6** | 2.0 | **1.8** | 1.1 | **0.9** | 32 (26) | **0.97** | 0.94 | **1.4** | 1.6 | 1.3 | **1.2** | 1.2 | **1.0** |
| BR-Sa1 | 35 (60) | 0.85 | **0.86** | 1.9 | **1.7** | 2.3 | **1.8** | 1.1 | **0.8** | 20 (4) | **0.58** | 0.53 | 3.2 | **2.6** | 1.3 | **0.9** | 1.2 | **0.8** |
| BR-Sa3 | 42 (59) | **0.91** | 0.90 | 2.2 | **2.0** | 2.5 | **2.2** | 1.6 | **1.3** | 22 (5) | **0.87** | 0.83 | **2.5** | 3.1 | 1.7 | **1.5** | 1.6 | **1.3** |
| GF-Guy | 36 (78) | **0.92** | **0.92** | 1.7 | **1.5** | 2.3 | **2.0** | 1.4 | **1.2** | 22 (17) | **0.95** | 0.93 | **1.1** | **1.1** | 1.6 | **1.3** | 1.6 | **1.3** |
| GH-Ank | 37 (65) | **0.84** | **0.84** | 1.4 | **1.3** | 1.5 | **1.3** | 0.5 | **0.3** | 24 (9) | 0.47 | **0.48** | **1.1** | **1.1** | 0.6 | **0.5** | 0.5 | **0.3** |
| MY-PSO | 87 (117) | **0.94** | 0.93 | 1.2 | **1.0** | 0.7 | **0.5** | 0.4 | **0.1** | 45 (10) | **0.88** | 0.84 | 0.9 | **0.7** | 0.5 | **0.3** | 0.5 | **0.2** |
| PA-SPn | 87 (95) | **0.91** | **0.91** | 1.6 | **1.5** | 1.2 | **1.0** | 0.7 | **0.6** | 29 (15) | **0.93** | **0.93** | **1.1** | **1.1** | 0.9 | **0.8** | 0.9 | **0.7** |
| ZM-Mon | 62 (120) | **0.93** | 0.90 | 1.5 | **1.4** | 1.6 | **1.5** | 0.9 | **0.7** | 48 (15) | 0.18 | **0.28** | **1.4** | 1.7 | 1.0 | **0.9** | 0.9 | **0.8** |
| Average | 82 (103) | **0.91** | 0.90 | 1.5 | **1.4** | 1.3 | **1.1** | 0.7 | **0.5** | 45 (19) | 0.73 | **0.74** | **1.3** | 1.5 | 0.8 | **0.6** | 0.7 | **0.5** |

At the GF-Guy site, another tropical rainforest, monthly averages of sensible heat are overestimated by $\sim 20\,\mathrm{Wm^{-2}}$ throughout the year, while latent heat flux is underestimated by about the same amount. The simulated sensible heat diurnal cycle peaks, on average, $\sim 100\,\mathrm{Wm^{-2}}$ above the measured values, while the peak of the simulated latent heat diurnal cycle is $\sim 130\,\mathrm{Wm^{-2}}$

below measured values. There is a marked decrease in simulated latent heat in October, and a corresponding sharp increase in sensible heat, due to excessively low soil moisture in the rooting zone in the model. The simulated October average diurnal sensible heat cycle peaks at $\sim 350\,\mathrm{Wm^{-2}}$ (measured: $\sim 200\,\mathrm{Wm^{-2}}$), while the average latent heat diurnal cycle peaks at $\sim 200\,\mathrm{Wm^{-2}}$.

### 3.4.2 Influence of different stomatal conductance schemes

Table 3 and Fig. 10 show model performance statistics for sensible heat fluxes, for the CLM/BB and the CLM/Med simulations. Correlations between modeled and observed sensible heat fluxes are very high, and similar for both runs. For hourly fluxes, $r$ is between $\sim 0.85$–92. Correlations between monthly averaged fluxes are weaker, but still high at most sites ($r > 0.75$), but they are very low for SD-Dem, AU-RDF and ZM-Mon. The correlation is lowest at SD-Dem, but the RMSE and Bias are lower for




**Table 4.** Model performance statistics for simulated hourly (left) and monthly (right) latent heat fluxes for the CLM/BB and the CLM/Med simulations. Bold fonts indicate the model configuration that performed better. The mean and standard deviation of the observed fluxes ($\lambda\bar{E}_\mathrm{o}$ and $\sigma_\mathrm{o}$, respectively), shown for reference, are given in $\mathrm{Wm}^{-2}$. The RMSE and Bias have been normalized by the mean of the observed fluxes for easier cross-site comparison.

| Site | $\lambda\bar{E}_\mathrm{o}(\sigma_\mathrm{o})$ | $r$ BB | $r$ Med | $\sigma_\mathrm{m}/\sigma_\mathrm{o}$ BB | $\sigma_\mathrm{m}/\sigma_\mathrm{o}$ Med | nRMSE BB | nRMSE Med | nBias BB | nBias Med | $\lambda\bar{E}_\mathrm{o}(\sigma_\mathrm{o})$ | $r$ BB | $r$ Med | $\sigma_\mathrm{m}/\sigma_\mathrm{o}$ BB | $\sigma_\mathrm{m}/\sigma_\mathrm{o}$ Med | nRMSE BB | nRMSE Med | nBias BB | nBias Med |
|---|---|---|---|---|---|---|---|---|---|---|---|---|---|---|---|---|---|---|
| AU-Emr | 50 (51) | **0.61** | 0.59 | **1.5** | 1.7 | **1.3** | 1.5 | **0.3** | 0.4 | 29 (13) | **0.78** | 0.76 | **2.0** | 2.3 | **0.7** | 0.8 | **0.4** | 0.4 |
| ES-Amo | 20 (25) | **0.72** | 0.66 | **2.3** | 3.1 | **2.7** | 3.7 | **1.5** | 1.6 | 14 (7) | 0.62 | **0.66** | **2.2** | 3.4 | **1.6** | 2.0 | **1.3** | 1.4 |
| AU-DaP | 93 (122) | **0.84** | **0.84** | **0.9** | 1.2 | **0.7** | 0.9 | **0.0** | 0.2 | 53 (45) | 0.92 | **0.93** | **0.9** | 1.1 | **0.3** | 0.4 | **0.0** | 0.2 |
| AU-Stp | 68 (87) | **0.82** | 0.79 | **1.2** | 1.5 | **0.9** | 1.1 | **0.0** | 0.1 | 43 (35) | **0.92** | **0.92** | **1.2** | 1.3 | **0.4** | 0.5 | **0.1** | **0.1** |
| CG-Tch | 86 (81) | **0.85** | 0.83 | **1.0** | 1.2 | **0.5** | 0.7 | **0.1** | 0.3 | 40 (22) | **0.93** | 0.90 | 0.8 | **0.9** | **0.2** | 0.3 | **0.1** | 0.2 |
| PA-SPs | 208 (127) | **0.85** | 0.78 | 0.8 | **1.1** | **0.4** | 0.5 | -0.3 | **-0.2** | 75 (18) | **0.71** | 0.70 | 0.7 | **0.9** | 0.3 | **0.2** | -0.2 | **-0.1** |
| AU-DaS | 100 (101) | 0.80 | **0.83** | **0.8** | 1.2 | **0.6** | 0.7 | -0.2 | **-0.1** | 67 (24) | 0.84 | **0.87** | **1.3** | 1.8 | **0.3** | 0.4 | -0.2 | **0.0** |
| AU-Dry | 92 (94) | **0.81** | 0.75 | **0.8** | 1.4 | **0.6** | 1.0 | -0.2 | **-0.1** | 58 (28) | **0.85** | 0.83 | **1.1** | 1.7 | **0.3** | 0.5 | -0.2 | **-0.1** |
| SD-Dem | 54 (75) | **0.85** | 0.82 | 0.6 | **1.1** | **0.9** | 0.9 | -0.3 | **-0.2** | 40 (33) | 0.94 | **0.95** | 0.6 | **1.0** | 0.5 | **0.4** | **-0.3** | **-0.3** |
| AU-Ade | 120 (133) | 0.84 | **0.90** | 0.8 | **1.0** | 0.6 | **0.5** | -0.2 | **-0.1** | 85 (37) | 0.91 | **0.94** | **0.9** | 1.2 | **0.2** | **0.2** | -0.2 | **0.0** |
| AU-Gin | 63 (60) | 0.72 | **0.75** | **1.0** | 1.3 | **0.7** | 0.8 | **0.0** | 0.1 | 43 (16) | **0.77** | 0.76 | **1.1** | 1.7 | **0.3** | 0.4 | **0.0** | 0.1 |
| AU-How | 139 (134) | 0.84 | **0.86** | **0.6** | 0.8 | 0.7 | **0.6** | -0.3 | **-0.2** | 88 (28) | **0.87** | **0.87** | **0.8** | 1.2 | 0.4 | **0.3** | -0.3 | **-0.2** |
| AU-RDF | 82 (104) | 0.72 | **0.73** | **0.8** | 1.2 | **0.9** | 1.1 | **0.1** | 0.4 | 49 (39) | **0.81** | 0.75 | **0.7** | 1.1 | **0.5** | 0.7 | **0.2** | 0.5 |
| AU-Rob | 104 (94) | 0.73 | **0.75** | **0.9** | 1.1 | **0.7** | **0.7** | -0.2 | **-0.1** | 80 (14) | **0.20** | 0.19 | **0.8** | 1.1 | 0.4 | **0.3** | -0.3 | **-0.2** |
| BR-Sa1 | 132 (134) | 0.86 | **0.89** | **1.0** | 1.1 | **0.5** | **0.5** | **0.1** | 0.2 | 87 (13) | 0.64 | **0.76** | 0.6 | **0.8** | **0.1** | 0.2 | **0.1** | 0.2 |
| BR-Sa3 | 161 (142) | 0.82 | **0.83** | **0.6** | 0.7 | **0.6** | **0.6** | **-0.3** | **-0.3** | 95 (10) | **0.40** | 0.38 | **1.1** | 1.6 | **0.3** | **0.3** | -0.3 | **-0.2** |
| GF-Guy | 162 (152) | 0.87 | **0.88** | **0.6** | 0.7 | **0.6** | **0.6** | -0.4 | **-0.3** | 109 (11) | **0.55** | **0.55** | **0.8** | 1.1 | 0.4 | **0.3** | **-0.3** | **-0.3** |
| GH-Ank | 72 (114) | 0.65 | **0.66** | **0.8** | **0.8** | **1.2** | **1.2** | **0.0** | 0.1 | 51 (21) | 0.02 | **0.07** | **0.6** | **0.6** | **0.5** | **0.5** | **0.1** | 0.1 |
| MY-PSO | 169 (151) | 0.89 | **0.93** | **0.6** | 0.7 | 0.6 | **0.4** | -0.4 | **-0.2** | 97 (7) | **0.73** | 0.65 | 0.7 | **1.0** | 0.3 | **0.2** | -0.3 | **-0.2** |
| PA-SPn | 195 (127) | 0.83 | **0.85** | **0.6** | 0.7 | **0.5** | **0.5** | -0.4 | **-0.3** | 88 (16) | 0.72 | **0.77** | 0.6 | **0.7** | 0.4 | **0.3** | -0.4 | **-0.3** |
| ZM-Mon | 72 (88) | **0.69** | 0.68 | 0.6 | **1.0** | **1.0** | 1.1 | -0.5 | **-0.3** | 59 (22) | 0.44 | **0.59** | **1.0** | 1.5 | **0.6** | **0.6** | -0.5 | **-0.4** |
| Average | 107 (105) | **0.79** | **0.79** | **0.9** | 1.2 | **0.8** | 0.9 | -0.1 | **0.0** | 64 (22) | 0.69 | **0.70** | **1.0** | 1.3 | **0.4** | 0.5 | -0.1 | **0.0** |

the CLM/Med run. At AU-RDF and ZM-mon, the CLM/Med simulation shows better correlations and smaller errors than the one using the BB scheme.

The model tends to overestimate average sensible heat. The hourly and monthly mean bias are non-negative at all sites (except at CG-Tch for monthly fluxes, where it is slightly negative), but normalized RMSE and mean bias are smaller for the CLM/Med run at most sites. The simulations seem to perform comparatively better in grasslands; for the Med simulation, RMSE is between $0.4$ and $0.8$ of the sample average (hourly fluxes), whereas it is in the $0.6$–$1.3$ range at savanna sites, and in the $0.5$–$2.2$ range for forest sites.

The variability of sensible heat flux is also overestimated by the model. In this case, the CLM/Med run performs better than the CLM/BB one for hourly fluxes, but the situation is the reversed for monthly average fluxes. Again, the simulations show better performance in grassland sites; for hourly fluxes, the Med simulation predicts $\sigma_\mathrm{m}/\sigma_\mathrm{o} \sim 1$–$1.5$ in grasslands, $\sim 1.3$–$1.6$ in savanna sites, and $\sim 1.0$–$2.2$ in forest sites.

Model performance statistics for latent heat fluxes are presented in Table 4 and Fig. 11. Correlations for hourly fluxes are between $0.7$ and $0.9$ for most sites. For monthly fluxes, correlations are poorer at forest sites, but errors are comparatively


**Table 5.** Cross-site averaged model performance statistics for simulated hourly and monthly sensible and latent heat fluxes. RMSE and Bias are given in $\mathrm{Wm}^{-2}$. Bold fonts indicate the best performing simulations in each metric.

| | $H$ | | | | $\lambda E$ | | | |
|---|---|---|---|---|---|---|---|---|
| *Hourly averages* | $r$ | $\sigma_\mathrm{m}/\sigma_\mathrm{o}$ | RMSE | Bias | $r$ | $\sigma_\mathrm{m}/\sigma_\mathrm{o}$ | RMSE | Bias |
| Noah/BB | **0.92** | 1.6 | 95 | 49 | 0.77 | 0.9 | 79 | −24 |
| CLM/BB | 0.91 | 1.5 | 88 | 44 | 0.79 | 0.9 | 74 | −19 |
| CLM(mod)/BB | **0.92** | 1.6 | 90 | 47 | 0.79 | 0.9 | 74 | −22 |
| SSiB/BB | 0.91 | 1.6 | 88 | 45 | 0.79 | 0.9 | 74 | −20 |
| Noah/Med | **0.92** | 1.5 | 83 | 40 | **0.81** | **1.0** | **72** | −15 |
| CLM/Med | 0.90 | **1.4** | **75** | **31** | 0.79 | 1.2 | 76 | **−8** |
| CLM(mod)/Med | 0.91 | 1.5 | 78 | 35 | 0.80 | 1.1 | 74 | −11 |
| SSiB/Med | 0.91 | **1.4** | 77 | 32 | 0.79 | 1.2 | 77 | −9 |
| *Monthly averages* | $r$ | $\sigma_\mathrm{m}/\sigma_\mathrm{o}$ | RMSE | Bias | $r$ | $\sigma_\mathrm{m}/\sigma_\mathrm{o}$ | RMSE | Bias |
| Noah/BB | 0.75 | 1.5 | 33 | 28 | 0.65 | 1.2 | 27 | −12 |
| CLM/BB | 0.73 | **1.3** | 30 | 25 | 0.69 | **1.0** | **24** | −10 |
| CLM(mod)/BB | 0.74 | 1.4 | 31 | 26 | **0.71** | **1.0** | **24** | −12 |
| SSiB/BB | 0.74 | 1.4 | 30 | 25 | 0.69 | **1.0** | **24** | −11 |
| Noah/Med | **0.77** | 1.6 | 29 | 23 | 0.70 | 1.2 | **24** | −8 |
| CLM/Med | 0.74 | 1.5 | **26** | **18** | 0.70 | 1.3 | **24** | −4 |
| CLM(mod)/Med | 0.75 | 1.6 | 27 | 20 | **0.71** | 1.3 | **24** | −6 |
| SSiB/Med | 0.75 | 1.5 | **26** | **18** | 0.70 | 1.4 | 25 | −5 |
| LPJ-GUESS | – | – | – | – | 0.64 | 1.6 | 27 | −5 |

small; normalized RMSE is in the 0.1–0.5 range. Hourly correlations are rather similar for both model configurations at most sites.

Latent heat fluxes tend to be underestimated in forest and savanna sites, and overestimated over grasslands. The CLM/BB
configuration seems to perform better at grassland sites, while the CLM/Med configuration performs better at forest sites. Results for savanna sites are mixed in terms of RMSE, but the CLM/Med scheme yields somewhat smaller biases.

The variability of simulated latent heat fluxes is always larger in the CLM/Med run than in the CLM/BB run. Over $C_3$ grasslands, both LSM runs predict a much larger variability than observed. This may be in part due to the fact that observed variability at these sites is very low in absolute terms. At savanna and woody savanna sites, the CLM/Med run predicts a
larger variability than observed, both in the hourly and monthly cases, whilst the CLM/BB simulation tends to produce a lower variability. For forest sites, both runs yield $\sigma_\mathrm{m} \lesssim \sigma_\mathrm{o}$, with the exception of BR-Sa3 in the CLM/Med run, where the variability of the monthly fluxes is $\sim 1.6$ times larger than observed. On average, the CLM/BB simulation shows better agreement with measured variability over grasslands, while CLM/Med performs somewhat better at forest sites.



### 3.4.3 Alternative model configurations

To evaluate the overall performance of the different model configurations, we considered the cross-site averaged statistics of each simulation (Table 5). Since the covered period differs across sites, this method ensures all sites contribute equally to the result.

Figure 12 shows the cross-site averaged metrics on a Taylor diagram. The clumping and clear separation of simulations using different stomatal conductance schemes suggests that this component of the model has a significantly greater influence than the
soil water uptake function on the behaviour of the model, with the exception of the linear (Noah-type) parametrization, which is much more restrictive than the other three in terms of water uptake. In this case, both the BB and Med simulations seem to perform similarly regarding monthly latent heat fluxes, with the variability of the modeled fluxes somewhere in between the BB and Med clumps.

Simulated sensible heat fluxes display similar correlation with observations in all runs. The correlation coefficient is very
high ($r \sim 0.9$) for hourly fluxes, and moderately high ($r \sim 0.75$) for monthly averages.

Sensible heat is overestimated in all model configurations; the average bias is always positive, but the Med simulations perform better in this respect. In the case of hourly averages, BB runs show an average bias of $\sim 46\,\mathrm{W\,m^{-2}}$, while the average value for Med runs is $\sim 35\,\mathrm{W\,m^{-2}}$. Average errors are also smaller in Med simulations. For hourly fluxes, the average RMSE is $\sim 90\,\mathrm{W\,m^{-2}}$ for BB runs, and $\sim 70\,\mathrm{W\,m^{-2}}$ for Med runs. For monthly fluxes, RMSE averages are $\sim 31\,\mathrm{W\,m^{-2}}$ and $\sim$
$27\,\mathrm{W\,m^{-2}}$ respectively.

The model also generally overestimates the variability of sensible heat. For hourly fluxes, the standard deviation of the sample is, on average, $\sim 1.6$ times greater than the measurements for the BB runs, and $\sim 1.5$ for the Med runs. In the case of monthly variability, BB runs perform better; the average standard deviations of modeled fluxes are $\sim 1.4$ and $\sim 1.6$ for BB and Med runs respectively.

Correlations between modeled and measured latent heat fluxes are lower than for sensible heat; $r \sim 0.8$ for hourly fluxes and $\sim 0.7$ for monthly fluxes. All runs show similar RMSE; $\sim 75\,\mathrm{W\,m^{-2}}$ and $\sim 24\,\mathrm{W\,m^{-2}}$ for hourly and monthly fluxes respectively. Latent heat is underestimated on average in all configurations. However, the Med runs perform significantly better than the BB runs on this metric. The average bias is $\sim -11\,\mathrm{W\,m^{-2}}$ (BB: $\sim -21\,\mathrm{W\,m^{-2}}$) for hourly fluxes, and $\sim -6\,\mathrm{W\,m^{-2}}$ (BB: $\sim -11\,\mathrm{W\,m^{-2}}$) for hourly fluxes. The variability of hourly latent heat fluxes is underestimated in the BB runs by about
the same amount that it is overestimated in the Med runs, but in the case of monthly fluxes, BB simulations seem to reproduce the measured variability better ($\sigma_\mathrm{m} \sim \sigma_\mathrm{o}$, while $\sigma_\mathrm{m} \sim 1.3\sigma_\mathrm{o}$ for Med runs).

Monthly averages of latent heat simulated by the non-LSM version of LPJ-GUESS show a slightly worse correlation with measurements than the LSM version of the model. The average bias is $\sim -5\,\mathrm{W\,m^{-2}}$, in line with Med simulations and lower than BB simulations, and the RMSE is slightly higher, but close to the LSM runs. However, the predicted variability is signifi-
cantly exaggerated; the standard deviation of the sample of modeled fluxes is, on average, $\sim 1.6$ times larger than observed.




**Table 6.** Foliar projective cover, averaged over the whole simulated period, of the plant functional types predicted for each site, given as a percentage. The LSM simulations use the CLM type water uptake factor. The dominant PFT for each site is highlighted in bold font.

| Site | TeBE LPJ-G | TeBE BB | TeBE Med | TrBR LPJ-G | TrBR BB | TrBR Med | TrIBE LPJ-G | TrIBE BB | TrIBE Med | TrBE LPJ-G | TrBE BB | TrBE Med | C3G LPJ-G | C3G BB | C3G Med | C4G LPJ-G | C4G BB | C4G Med |
|---|---|---|---|---|---|---|---|---|---|---|---|---|---|---|---|---|---|---|
| AU-Emr | – | – | – | – | – | – | – | – | – | – | – | – | **61** | **23** | **24** | – | – | – |
| ES-Amo | – | – | – | – | – | – | – | – | – | – | – | – | **61** | **67** | **63** | – | – | – |
| AU-DaP | – | – | – | – | – | – | – | – | – | – | – | – | 1 | – | – | **82** | **94** | **95** |
| AU-Stp | – | – | – | – | – | – | – | – | – | – | – | – | 1 | – | – | **71** | **56** | **57** |
| CG-Tch | – | – | – | – | – | – | – | – | – | – | – | – | 1 | – | – | **88** | **98** | **98** |
| PA-SPs | – | – | – | – | – | – | – | – | – | – | – | – | – | – | – | **95** | **98** | **98** |
| AU-DaS | – | – | – | 11 | **33** | 21 | 4 | 14 | 4 | 10 | 3 | 5 | – | – | – | **61** | 30 | **54** |
| AU-Dry | – | – | – | 7 | 28 | – | 2 | 6 | 1 | 2 | 3 | 2 | 1 | – | – | **64** | **36** | **85** |
| SD-Dem | – | – | – | 1 | 9 | – | 1 | 2 | – | 8 | 6 | 5 | – | – | – | **38** | **21** | **45** |
| AU-Ade | – | – | – | 7 | 28 | 22 | 3 | 12 | 3 | 5 | 3 | 6 | 1 | – | – | **63** | **37** | **52** |
| AU-Gin | **43** | **41** | **31** | – | – | – | – | – | – | – | – | – | 9 | 7 | 7 | – | – | – |
| AU-How | – | – | – | 21 | **40** | 31 | 7 | 16 | 12 | 20 | 3 | 5 | – | – | – | **41** | 23 | **40** |
| AU-RDF | – | – | – | 10 | **37** | 21 | 4 | 15 | 9 | 4 | 4 | 8 | 1 | – | – | **68** | 20 | **45** |
| AU-Rob | – | – | – | **57** | **54** | **45** | 29 | 23 | 22 | 2 | 1 | 3 | – | – | – | 4 | 9 | 10 |
| BR-Sa1 | – | – | – | **66** | **65** | **63** | 28 | 12 | 23 | 2 | 3 | 6 | – | – | – | 8 | 6 | 6 |
| BR-Sa3 | – | – | – | **51** | **50** | **44** | 23 | 21 | 21 | 4 | 5 | 3 | – | – | – | 7 | 14 | 11 |
| GF-Guy | – | – | – | **57** | **54** | **50** | 35 | 22 | 23 | 3 | 2 | 3 | – | – | – | 7 | 12 | 12 |
| GH-Ank | – | – | – | **61** | **63** | **59** | 24 | 26 | 20 | 2 | 6 | 7 | 1 | – | – | 8 | 3 | 8 |
| MY-PSO | – | – | – | **53** | **64** | **56** | 26 | 15 | 24 | 4 | 4 | 14 | – | – | – | 7 | 6 | 5 |
| PA-SPn | – | – | – | **60** | **57** | **51** | 27 | 20 | 20 | 2 | 1 | 4 | – | – | – | 6 | 8 | 12 |
| ZM-Mon | 2 | – | – | 3 | 23 | 4 | 3 | 4 | 3 | 3 | 4 | 5 | 2 | 1 | 1 | **54** | **44** | **75** |





**Table 7.** List of Plant Functional Types in the standard configuration of LPJ-GUESS (only PFTs predicted by the simulations in this study are listed)

| Plant functional type | Abbreviation |
| --- | --- |
| Temperate Broadleaf Evergreen | TeBE |
| Tropical Broadleaf Raingreen | TrBR |
| Tropical shade-Intolerant Broadleaf Evergreen | TrIBE |
| Tropical Broadleaf Evergreen | TrBE |
| $C_3$ Grass | C3G |
| $C_4$ Grass | C4G |

### 3.4.4 Ecosystem structure and function

We compared the predictions of the CLM/BB and the CLM/Med simulations to standard LPJ-GUESS for species composition and a number of ecosystem structure and function variables.

Table 6 shows the FPC of the simulated PFTs (Table 7) at each site. All three simulations predict the same type of grass at grassland sites. At AU-Emr, the grass coverage predicted by the LSM runs is substantially lower than in standard LPJ-GUESS. Land surface model simulations predict a larger FPC at most $C_4$ grassland sites, except at AU-Stp, where FPC is $\sim 20\%$ smaller than in the LPJ-GUESS simulation.

All three simulations predict a temperate forest with a $C_3$ grassy understory at AU-Gin. At the rest of the savanna and woody savanna sites, the three runs predict a mixture of tropical trees and $C_4$ grasses with a relatively high proportion of the latter. $C_4$ grass is the dominant PFT in the standard LPJ-GUESS and Med simulations, while in the BB simulations the tree coverage is close to or larger than that of grasses. In the standard LPJ-GUESS run, the simulated landscape is closer to the savanna IGBP classification at most sites (tree coverage lower than $30\%$). The BB simulation predicts a woody savanna (tree coverage higher than $30\%$) at all sites except for SD-Dem, while the Med simulation predicts the expected landscape at all sites (Table 1) except for AU-DaS, where it produces a woody savanna.

At the forest sites, all three simulations predict a mixture of tropical trees and $C_4$ grass, where the coverage of the latter is relatively small. The dominant PFT is TrBR, taking between 50 and $65\%$ of the coverage area. A similar prediction is made for the PA-SPn site, which is classified as a deciduous broadleaf forest. However, for the ZM-Mon site, also a deciduous forest, all three models yield a PFT composition that is closer to that of savanna sites.

Model predictions for the rest of the selected variables are shown in Table 8. The two $C_3$ grassland sites show different behaviour with respect to ecosystem productivity and respiration. At AU-Emr, LSM simulations predict substantially lower gross primary production (GPP) and autotrophic respiration ($R_\mathrm{a}$) than standard LPJ-GUESS, which results in lower estimates of net primary production (NPP). This site is a net carbon source (positive net ecosystem exchange, NEE) in all three simulations, which agrees with observations. Modelled LAI is lower in the LSM simulations, and much closer to the observed value. At ES-





**Table 8.** Comparison of selected variables related to simulated ecosystem structure and function between standard LPJ-GUESS and the LSM version at the selected sites. The LSM values are from the CLM/BB and the CLM/Med simulations. Gross primary production (GPP), autotrophic respiration (Ra), net primary production (NPP), heterotrophic respiration (Rh) and net ecosystem exchange (NEE) are given in $gCm^{-2}y^{-1}$. Bold fonts in the LAI and NEE columns indicate the closest match to the observed value. Bold fonts in the rest of the columns indicate the LSM prediction closest to standard LPJ-GUESS.

| Site | LAI | | | | GPP | | | Ra | | | NPP | | | Rh | | | NEE | | | |
|---|---|---|---|---|---|---|---|---|---|---|---|---|---|---|---|---|---|---|---|---|
| | Obs | LPJ-G | BB | Med | LPJ-G | BB | Med | LPJ-G | BB | Med | LPJ-G | BB | Med | LPJ-G | BB | Med | Obs | LPJ-G | BB | Med |
| AU-Emr | 0.7 | 2.0 | **0.6** | **0.6** | 1184 | 431 | 414 | 922 | **360** | 330 | 262 | 70 | **84** | 303 | 104 | 93 | 53 | **41** | 33 | 10 |
| ES-Amo | – | 2.0 | 2.3 | 2.1 | 767 | 976 | **850** | 479 | 666 | **540** | 288 | **310** | 310 | 262 | **254** | 233 | 182 | –27 | –55 | –76 |
| Average | 0.7 | 2.0 | 1.4 | **1.3** | 976 | **704** | 632 | 700 | **513** | 435 | 275 | 190 | **197** | 283 | **179** | 163 | 118 | **7** | –11 | –33 |
| AU-DaP | 1.5 | **4.0** | 6.9 | 7.8 | 976 | **1544** | 1888 | 296 | **611** | 693 | 681 | **933** | 1194 | 544 | **807** | 973 | –210 | –137 | –126 | **–221** |
| AU-Stp | 0.5 | 3.0 | **1.8** | 2.0 | 896 | 548 | **692** | 294 | 215 | **246** | 602 | 333 | **446** | 559 | 299 | **397** | –52 | –43 | –34 | **–49** |
| CG-Tch | 2.0 | **4.5** | 11.3 | 11.5 | 1020 | **1965** | 2019 | 336 | **757** | 780 | 682 | **1208** | 1238 | 596 | **1089** | 1116 | –148 | –86 | –118 | **–122** |
| PA-SPs | 5.4 | **6.5** | 10.7 | 10.9 | 1447 | **2026** | 2099 | 536 | **794** | 823 | 912 | **1232** | 1276 | 836 | **1133** | 1169 | 277 | **–76** | –98 | –107 |
| Average | 2.4 | **4.5** | 7.7 | 8.1 | 1084 | **1521** | 1674 | 365 | **594** | 636 | 719 | **926** | 1039 | 634 | **832** | 914 | –33 | **–86** | –94 | –125 |
| AU-DaS | 1.5 | **3.4** | 3.7 | 4.2 | 966 | **1123** | 1242 | 332 | 541 | **518** | 633 | **582** | 724 | 470 | **425** | 521 | –284 | –164 | –157 | **–202** |
| AU-Dry | 1.2 | **2.8** | 3.3 | 4.7 | 826 | **1015** | 1286 | 263 | 476 | **436** | 563 | **538** | 850 | 428 | **388** | 689 | –307 | –137 | –150 | **–161** |
| SD-Dem | 0.9 | 1.3 | **1.2** | 1.3 | 460 | **421** | 543 | 173 | **195** | 190 | 287 | **225** | 351 | 246 | **189** | 315 | –73 | –42 | **–36** | –36 |
| Average | 1.2 | **2.5** | 2.7 | 3.4 | 751 | **853** | 1024 | 256 | 404 | **381** | 495 | **449** | 642 | 381 | **334** | 508 | –221 | –114 | –115 | **–133** |
| AU-Ade | 1.1 | **2.8** | 3.9 | 4.2 | 764 | **1126** | 1206 | 235 | 530 | **518** | 529 | **595** | 688 | 400 | **448** | 528 | –272 | –130 | –148 | **–160** |
| AU-Gin | 0.9 | 1.8 | 1.9 | **1.5** | 1178 | 1346 | **1216** | 914 | 1109 | **996** | 264 | **236** | 220 | 233 | **234** | 196 | –317 | **–30** | –1 | –24 |
| AU-How | 1.5 | **3.6** | 4.0 | 4.4 | 1073 | **1223** | 1298 | 427 | **603** | 588 | 647 | **620** | 710 | 476 | **487** | 529 | –692 | –171 | –133 | **–181** |
| AU-RDF | 1.6 | **3.4** | 3.8 | 4.2 | 915 | **1212** | 1267 | 300 | 633 | **574** | 614 | **579** | 692 | 515 | **492** | 589 | 329 | –99 | **–87** | –102 |
| Average | 1.3 | **2.9** | 3.4 | 3.6 | 983 | **1227** | 1247 | 469 | 719 | **669** | 514 | **507** | 578 | 406 | **415** | 460 | –238 | –107 | –92 | **–117** |
| AU-Rob | 4.3 | **4.7** | 4.9 | 4.7 | 1650 | **1663** | 1668 | 775 | **800** | 809 | 873 | 861 | **859** | 613 | **626** | 642 | –744 | **–260** | –236 | –217 |
| BR-Sa1 | 6.5 | 6.4 | 5.2 | **5.7** | 2266 | 1910 | **2022** | 1137 | 1038 | **1058** | 1130 | 873 | **963** | 939 | 740 | **853** | –4 | –190 | –134 | **–110** |
| BR-Sa3 | 6.5 | 4.5 | **4.8** | 4.5 | 1599 | **1505** | 1567 | 794 | **739** | 798 | 805 | **766** | 770 | 647 | **646** | 601 | –105 | –159 | **–120** | –168 |
| GF-Guy | 5.9 | **5.5** | 5.2 | 5.1 | 1926 | 1753 | **1815** | 942 | 882 | **924** | 983 | 871 | **891** | 760 | 683 | **695** | –157 | –223 | **–187** | –195 |
| GH-Ank | – | 5.4 | **5.2** | 5.1 | 1779 | **1648** | 1665 | 860 | **844** | 869 | 920 | **803** | 796 | 876 | **701** | 776 | – | –44 | –103 | **–20** |
| MY-PSO | 6.5 | **5.5** | 4.6 | 5.2 | 2039 | 1619 | **1805** | 993 | 859 | **913** | 1046 | 759 | **892** | 841 | 616 | **739** | –971 | **–205** | –144 | –154 |
| Average | 5.9 | **5.3** | 5.0 | 5.1 | 1876 | 1683 | **1757** | 917 | 860 | **895** | 959 | 822 | **862** | 779 | 669 | **718** | –396 | **–180** | –154 | –144 |
| PA-SPn | 2.9 | 5.0 | **4.7** | 4.8 | 1654 | 1550 | **1605** | 839 | 803 | **810** | 764 | 747 | **794** | 764 | 606 | **640** | –458 | **–75** | –143 | –155 |
| ZM-Mon | 1.6 | **2.2** | 3.5 | 4.2 | 621 | **1002** | 1101 | 202 | **399** | 370 | 418 | **603** | 731 | 312 | **433** | 581 | 143 | **–107** | –171 | –149 |
| Average | 2.3 | **3.6** | 4.1 | 4.5 | 1137 | **1276** | 1353 | 509 | 601 | **590** | 629 | **675** | 762 | 538 | **519** | 611 | –157 | –91 | **–157** | –152 |





Amo, GPP is enhanced in the LSM versions, while heterotrophic respiration ($R_h$) is low compared to standard LPJ-GUESS,
resulting in a slightly enhanced carbon sink in the LSM simulations.

At three out of four $C_4$ grassland sites, the LSM simulations generate $\sim 50$ to $100\%$ higher GPP than LPJ-GUESS. LAI
is extremely high (between $\sim 7$ and $\sim 11$), and much higher than the observed values. $R_a$ increases too, but not as much,
in absolute terms, as GPP. This results in an increased NPP in the LSM simulations ($\sim 30\%$ in the BB run and $\sim 45\%$ in
the Med run, Fig. 13, upper panels). $R_h$ is also enhanced with respect to standard LPJ-GUESS, but again not as much as
NPP, so the $CO_2$ sink is strengthened in both LSM runs (Fig. 13, lower panels). Simulations at the AU-Stp site again show a
pattern different to the other $C_4$ grassland sites; GPP and NPP simulated in the LSM runs are both lower than in the non-LSM
simulation. However, NPP is substantially higher in the Med simulation than in the BB simulation. This results in NEE being
less negative than in LPJ-GUESS for the BB simulation and more negative than LPJ-GUESS for Med. NEE values predicted
by the Med simulation are closest to the observed fluxes, except at PA-SPs, where all simulations fail to predict a net $CO_2$
source.

The BB and Med simulations also behave differently at savanna and woody savanna sites. At most sites, BB predicts LAI
values similar to LPJ-GUESS, while Med produces higher LAIs. Both GPP and $R_a$ increase in the LSM runs, but in the BB
simulation the comparatively larger increase in $R_a$ results in an NPP decrease $\sim 8\%$ for savanna and $\sim 2\%$ for woody savanna
with respect to LPJ-GUESS. In the Med simulation, the absolute GPP increase is larger than that of $R_a$, which results in an NPP
increase of $\sim 30\%$ for savanna and $\sim 12\%$ for woody savanna relative to the LPJ-GUESS simulation (Fig. 13, upper panels). In
the BB simulation, $R_h$ is $\sim 12\%$ smaller than the LPJ-GUESS prediction in savanna sites, and close to LPJ-GUESS in woody
savanna sites, which yields similar NEE averages for savanna sites, but a $\sim 15\%$ reduced carbon sink for woody savanna (Fig.
13, lower left). In the Med simulation, savanna and woody savanna show similar tendencies; $R_h$ increases relative to the LPJ-
GUESS run, but not enough to overtake the increase in NPP. As a result, the average net carbon sink is enhanced by $\sim 17\%$ at
savanna sites and by $\sim 9\%$ at woody savanna sites. Observed NEE is generally more negative than the values predicted by the
simulations, with the exception of AU-RDF, where observations indicate a large net carbon source.

The three simulations slightly underestimate LAI at most evergreen broadleaf forest sites, but the non-LSM version produces
values that are somewhat closer to the observations. Both GPP and $R_a$ are somewhat lower in the LSM simulations, but similar
to the LPJ-GUESS run. The net result is somewhat lower NPP estimates at these sites; a $\sim 14\%$ decrease, relative to LPJ-
GUESS, in the BB simulation and a $\sim 10\%$ decrease in the Med simulation (Fig. 13, upper panels). $R_h$ is also reduced in
LSM simulations, but the balance with NPP still yields an NEE value above LPJ-GUESS's diagnostic (increases of $\sim 15\%$
and $\sim 20\%$ in the BB and the Med simulations, respectively; Fig. 13, lower panels). Observed NEE shows great cross-site
variability. Simulations overestimate the strength of the $CO_2$ sink at the two Brazil sites and the Guiana sites, and greatly
underestimate it at Robson Creek and Pasoh Forest.

The two deciduous broadleaf forest sites show different patterns. At PA-SPn, GPP is somewhat lower in the LSM simulations
than in LPJ-GUESS, while autotrophic respiration is similar for all three simulations, which results in slightly decreased NPP
values in the LSM runs. However, standard LPJ-GUESS predicts substantially higher heterotrophic respiration, so NEE is
decreased in the LSM runs, but still much higher than the observed value of $\sim -458\,\mathrm{gCm^{-2}y^{-1}}$. At the ZM-Mon site, GPP,





$R_a$ and NPP increase in the LSM runs. Heterotrophic respiration is also enhanced in the LSM simulations, but the balance with
NPP still results in more negative NEE than standard LPJ-GUESS. The simulations predict a carbon sink between $-107$ and
$-171\,\mathrm{gCm^{-2}y^{-1}}$, while observations indicate a carbon source of $\sim 143\,\mathrm{gCm^{-2}y^{-1}}$.

Figure 14 shows a comparison between land-cover averages of measured and modeled NEE for $C_4$ grasslands, savanna,
woody savanna and evergreen forests. Average measured NEE is negative for all land cover types, and substantially more
negative than in the simulations for savanna, woody savanna and evergreen broadleaf forests, implying an average underes-
timation of the C sink by the models at these sites. At $C_4$ sites simulations predict NEE values between $-86\,\mathrm{gCm^{-2}y^{-1}}$
and $-125\,\mathrm{gCm^{-2}y^{-1}}$, while observations indicate a less negative value of $-33\,\mathrm{gCm^{-2}y^{-1}}$. For savanna, measured NEE is
$\sim -221\,\mathrm{gCm^{-2}y^{-1}}$, while simulations predict an average between $\sim -114\mathrm{gCm^{-2}y^{-1}}$ and $\sim -133\mathrm{gCm^{-2}y^{-1}}$. For woody
savanna, measured NEE averages to $\sim -238\mathrm{gCm^{-2}y^{-1}}$, while simulated fluxes are $\sim -100\mathrm{gCm^{-2}y^{-1}}$. Measured fluxes at
evergreen broadleaf forests are, on average, $\sim -396\,\mathrm{gCm^{-2}y^{-1}}$, while simulations predict average fluxes between $-144$ and
$-180\,\mathrm{gCm^{-2}y^{-1}}$. However, this is the result of very large negative values measured at AU-Rob, and MY-PSO (Table 8). In
general, differences in simulated fluxes between standard LPJ-GUESS and the two LPJ-GUESS/LSM simulations seem to be
small compared to the magnitude of observed fluxes, and the interannual and cross-site variability of the measured fluxes is
much greater than in the simulations.

## 4   Discussion and summary

In this work we described a number of modifications to the LPJ-GUESS DGVM aimed at making the model suitable for direct
coupling with an atmospheric model. The newly incorporated energy balance module resolves the diurnal cycle of energy
and water fluxes between the canopy and the atmosphere, as opposed to LPJ-GUESS's daily calculations. This enables the
shorter time step used by atmospheric models to be matched. The simple, Beer's law-based PAR absorption calculations were
replaced with a more sophisticated two-stream radiative transfer scheme (Sellers, 1985; Dai et al., 2004), which allows for
separate treatment of sunlit and shaded leaves in the canopy. The representation of soil physical processes was modified in two
ways. Firstly, the original $1.5\,\mathrm{m}$ deep, two-layer soil column was replaced with a $3\,\mathrm{m}$ deep, 9-layer column. Secondly, the soil
heat and water transport schemes were replaced with less parametrized formulations. Soil heat transport is now calculated by
solving the heat diffusion equation, while soil water transport is solved by applying Richard's equation. These formulations are
better fit to resolve near-surface heat and water fluxes on the sub-daily time scales introduced in the model.

### 4.1   Evaluation of the simulated heat fluxes

The new model was evaluated by comparing simulated fluxes of sensible and latent heat with flux tower measurements at 21
FLUXNET sites. Stöckli et al. (2008) used a similar analysis (including the filtering of fluxes with $u^* < 0.2\,\mathrm{m\,s^{-1}}$) to evaluate
the improvement in performance of CLM 3.5 after introducing nitrogen limitation of photosynthesis, a ground water model and
an updated formulation of surface resistance. Owing to the different site selection, a rigorous comparison of that study with the
results presented in section 3.4.2 is not possible, but average statistics of model performance can provide an overview of how





**Table 9.** Average correlation and RMSE between observed and simulated sensible and latent heat fluxes for LPJ-GUESS/LSM, CLM 3.0, and CLM 3.5. RMSE values (between brackets) are given in $\mathrm{Wm^{-2}}$. Hourly and monthly values are labeled (h) and (m) respectively. CLM values correspond to averages over temperate, tropical and grassland sites as reported in Stöckli et al. (2008). LPJ-GUESS/LSM values correspond to the CLM/Med simulation.

|  | LPJ-GUESS/LSM | CLM 3.0 | CLM 3.5 |
|---|---|---|---|
| $H$ (h) | 0.90 (75) | 0.70 (101) | 0.77 (72) |
| $H$ (m) | 0.74 (26) | 0.66 (47) | 0.61 (30) |
| $\lambda E$ (h) | 0.79 (76) | 0.54 (83) | 0.80 (59) |
| $\lambda E$ (m) | 0.70 (24) | 0.53 (40) | 0.85 (29) |

the models compare. Table 9 shows averaged values of the correlation coefficient and RMSE for CLM 3.0, CLM 3.5 and LPJ-GUESS/LSM. Our model seems to yield stronger correlations between measured and observed sensible heat fluxes, for similar RMSE values, while CLM 3.5 appears to perform better in terms of RMSE for hourly latent heat fluxes, and the correlation between measured and observed monthly latent heat fluxes is stronger. In order to ascertain the significance of these findings a comparison using the same site-measured fluxes and forcing climate would be needed. Nevertheless, the values presented in Table 9 suggest that the performance of our model is closer to CLM 3.5 than to CLM 3.0.

Sensible heat is generally overestimated by the LSM model. Poor performance in sensible heat flux estimation is a common issue of many land surface models (Best et al., 2015). The reason for this is not well understood. It has been suggested that the models, the majority of which use similar methods to calculate the turbulent fluxes, do not extract all the information available in the climate forcing data. However, eddy covariance measurements often fail to close the energy balance, and might systematically underestimate sensible heat much more than latent heat, which would appear as an overestimation of sensible heat in the simulations. A detailed discussion of these issues is provided in Haughton et al. (2016).

One issue in our simulations is the marked underestimation of latent heat flux during extremely dry periods, when the rooting zone is nearly depleted of water available for plant uptake. This, in turn, causes a strong spike in sensible heat (Fig. 7, upper right panel). All eight model configurations show this behaviour. One possible reason for this is the choice of free drainage boundary conditions at the bottom of the soil column. Simulating ground water in the model may promote the retention of some soil moisture during dry periods and thus help alleviate this problem (Stöckli et al., 2008). Deeper root profiles and lateral access to soil water may also be important to support evapotranspiration in dry periods (Schenk and Jackson, 2002).

We implemented two different stomatal conductance schemes: the Ball-Berry model (Ball et al., 1987) and the Medlyn model (Medlyn et al., 2011). One notable difference between these two models concerns the behaviour of the stomatal conductance when the vapour pressure deficit at the leaf surface (VPD) is small. In the Ball-Berry model, stomatal conductance increases linearly with decreasing VPD, while in the Medlyn model stomatal conductance increases much more rapidly as VPD approaches zero (Fig. 3). Larger stomatal conductance leads to generally higher evapotranspiration values (less negative bias values, Table 5), and enhanced GPP (Fig. 13) in simulations using the Medlyn model. A statistical evaluation of the impact of





**Table 10.** Daily average climate measured at the four $C_4$ grassland sites in this study. $T_{\mathrm{atm,day}}$: daily average temperature (°C); $P_{\mathrm{day}}$: daily average precipitation (mm/day); $I^{\downarrow}_{0,\mathrm{day}}$: daily average incoming solar irradiance ($\mathrm{W\,m^{-2}}$).

|        | $T_{\mathrm{atm,day}}$ | $P_{\mathrm{day}}$ | $I^{\downarrow}_{0,\mathrm{day}}$ |
|--------|------|-----|-------|
| AU-DaP | 25.5 | 3.8 | 249.5 |
| AU-Stp | 26.2 | 2.0 | 262.0 |
| CG-Tch | 24.3 | 4.7 | 148.7 |
| Pa-SPs | 25.3 | 6.4 | 177.1 |

**Table 11.** Observed and simulated LAI at the four $C_4$ grassland sites when all natural PFTs are allowed to grow in the patch. The BB and Med values correspond to LSM simulations using the CLM-type water uptake function.

| Site | Observed | Total | | | Tree | | | Grass | | |
|------|----------|-------|-----|-----|-------|-----|-----|-------|-----|-----|
|      |          | LPJ-G | BB  | Med | LPJ-G | BB  | Med | LPJ-G | BB  | Med |
| AU-DaP | 1.5 | 3.5 | 3.6 | 4.0 | 1.0 | 2.1 | 1.5 | 2.5 | 1.6 | 2.4 |
| AU-Stp | 0.5 | 2.6 | 1.0 | 0.9 | 0.3 | 0.8 | 0.6 | 2.3 | 0.3 | 0.3 |
| CG-Tch | 2.0 | 4.4 | 4.5 | 4.7 | 4.2 | 4.4 | 4.3 | 0.2 | 0.1 | 0.3 |
| PA-SPs | 5.4 | 5.2 | 4.4 | 4.4 | 4.9 | 3.8 | 3.9 | 0.3 | 0.6 | 0.6 |

these differences on the model output was not carried out, but the clumping of symbols representing the two different stomatal conductance models seen in Fig. 12 suggests that the stomatal conductance scheme has a significantly greater impact on the model's behaviour than the choice of soil water uptake function, except when the latter is very restrictive (for example, the Noah-type in our simulations, Eq. 49).

## 4.2   Why does $C_4$ productivity increase so much in LSM simulations?

The results presented in Section 3.4.4 show that predictions of PFT composition are similar for standard LPJ-GUESS and the LSM simulations. However, ecosystem productivity, respiration and carbon dioxide exchange vary between LSM simulations using different stomatal conductance schemes and with respect to standard LPJ-GUESS. Very notably, both the gross and net productivities of $C_4$ grasses are substantially enhanced in the LSM simulations. This results in unrealistically high simulated LAI values at three out of four sites where the grasses are allowed to grow without competition.

We found that the main reason for this behaviour is the occurrence of higher photosynthetic rates in the LSM simulations due to the mitigation of biochemical N limitation at higher leaf temperatures. Standard LPJ-GUESS uses a daily average of the forcing air temperature as a proxy for leaf temperature in the $V_{\mathrm{max}}$ calculation. By contrast, LPJ-GUESS/LSM simulates leaf temperature explicitly, and uses a daytime average (Eq. 44) to estimate $V_{\mathrm{max}}$. This average leaf temperature can be several degrees above the forcing air temperature, which makes it possible to reach the optimal maximum carboxylation rate at lower





leaf nitrogen concentrations (see Haxeltine and Prentice, 1996). This makes it easier for the plants to attain the optimal $V_{\max}$ with the available nitrogen, which enhances productivity. Exceedingly high leaf temperatures can have a negative impact on $V_{\max}$ due to the thermal breakdown of the biochemical reactions. However, the simulated leaf temperatures are still within the optimal temperature range for $C_4$ grasses (20 to $45°$C in LPJ-GUESS). The temperature dependence of $V_{\max}$, including the effect on nitrogen limitation, is illustrated in Fig. 15.

At AU-Stp, all three simulations predict much lower productivities. In this case, water availability is the limiting factor. This site receives, on average, considerably less rain water than the other three $C_4$ grasslands (Table 10), which leads to lower values of the $\beta$ factor (Eq. 48), and brings photosynthetic rates down.

As pointed out in Section 3.2, the simulated PFTs were restricted to grassy types at these sites. Table 11 shows a summary of LAI values predicted by standard LPJ-GUESS and two representative LSM simulations when establishment is not restricted 645 to grassy PFTs. All three experiments predict a mixture of trees and grasses, but total LAI in the LSM runs is much lower than in the simulations where only grasses were allowed to establish. In these runs, competition with trees limits the resources available to grasses, and shading from the taller trees helps lowering the average leaf temperature of the grassy understory, all of which helps counteract the effect described above.

### 4.3 Conclusion and outlook

The developments presented in this paper will enable to study feedbacks between the climate and the biosphere using the state-of-the-art DGVM LPJ-GUESS coupled to an atmospheric model directly. Work is in progress regarding the development of a flexible interface to enable such coupling, as well as extending the model's ability to simulate cold-climate ecosystems. More work is also needed to characterize and fully understand the model's response to the switch from using air temperature as a proxy for leaf temperature to simulating leaf temperature explicitly, particularly as these concern the productivity of $C_4$ plants 655 in well watered, no-competition situations (e.g., monoculture crops or managed pastures). These developments will allow to use LPJ-GUESS/LSM in regional as well as global studies. Given the capacity of LPJ-GUESS to represent land use change and management (Lindeskog et al., 2013, 2021; Olin et al., 2015), the range of applications includes exploring impacts of management on regional climate, which can be an important tool to help devise and assess climate change mitigation policies.

*Code and data availability.* LPJ-GUESS is a worldwide developed and refined DGVM. The model code is managed and maintained by 660 the Department of Physical Geography and Ecosystem Science, Lund University, Sweden. The source code can be made available with a collaboration agreement under the acceptance of certain conditions. For this reason, a DOI for the model code cannot be provided. The code with the augmentations developed for this paper is available to the editor and reviewers via a restricted link, on the condition that the code is used only for review purposes, and is deleted after the review process. Additional details and information can be found at the LPJ-GUESS website (http://web.nateko.lu.se/lpj-guess, last access: 17 December 2021). The forcing data, evaluation data, model output and analysis 665 scripts used in this study have been uploaded to a public repository with DOI 10.5281/zenodo.5813886.





## Appendix A: Derivation of the canopy conductance for water vapor flux

During a given time step $\Delta t$, the total amount of water evapotranspired from the sunlit part of the canopy can be expressed as the sum of the contributions from the dry and wet parts:

$$E_{\text{sun}}\Delta t = (1 - f_{\text{wet}})E_{\text{sun,tr}}\Delta t$$

$$+ f_{\text{wet}}[E_{\text{sun,ev}}\Delta t_{\text{wet,sun}} + E_{\text{sun,tr}}(\Delta t - \Delta t_{\text{wet,sun}})] \tag{A1}$$

where $E_{\text{sun}}$ is the actual evapotranspiration rate, $E_{\text{sun,tr}}$ is the potential rate of transpiration, $E_{\text{sun,ev}}$ is the potential rate of evaporation, and $\Delta t_{\text{wet,sun}}$ is the time that the wet part of the sunlit canopy remains wet at the potential evaporation rate. The latter is calculated as:

$$\Delta t_{\text{wet,sun}} = \min\left(\frac{w_{\text{c}}\text{PAI}_{\text{c,sun}}/\text{PAI}_{\text{c}}}{f_{\text{wet}}E_{\text{sun,ev}}}, \Delta t\right), \tag{A2}$$

where $w_{\text{c}}$ is the current canopy water content $(\text{kg}\,\text{m}^{-2})$.

The evaporation rates in the above equations can be expressed as follows:

$$E_{\text{sun}} = -\rho g_{\text{w,sun}}[q_{\text{ca}} - q^*(T_{\text{sun}})]; \tag{A3}$$

$$E_{\text{tr,sun}} = -\rho \sum_i \text{LAI}_{\text{sun}}^{(i)} \frac{g_{\text{s,sun}}^{(i)}g_{\text{b}}}{g_{\text{s,sun}}^{(i)} + g_{\text{b}}}[q_{\text{ca}} - q^*(T_{\text{sun}})]; \tag{A4}$$

$$E_{\text{ev,sun}} = -\rho\text{PAI}_{\text{c,sun}}g_{\text{b}}[q_{\text{ca}} - q^*(T_{\text{sun}})]. \tag{A5}$$

where the index $i$ runs over cohorts. Inserting these expressions into Eq. (A1), dividing both sides by $\Delta t$, simplifying, and rearranging terms yields

$$g_{\text{w,sun}} = f_{\text{wet}}\eta_{\text{sun}}\text{PAI}_{\text{c,sun}}g_{\text{b}}$$

$$+ (1 - f_{\text{wet}}\eta_{\text{sun}}) \sum_i \text{LAI}_{\text{sun}}^{(i)} \frac{g_{\text{s,sun}}^{(i)}g_{\text{b}}}{g_{\text{s,sun}}^{(i)} + g_{\text{b}}}, \tag{A6}$$

where $\eta_{\text{sun}} = \Delta t_{\text{wet,sun}} / \Delta t$. Identical equations apply to the shaded part of the canopy.

## Appendix B: List of symbols, parameters and variables used in the model description

**Figure 1.** Flowchart of the main daily simulation loop in standard LPJ-GUESS (red branch) and the modified version (LPJ-GUESS/LSM, blue branch). The shaded area indicates the sub-daily loop in the modified version. The dashed line encloses coupled iterative calculations.




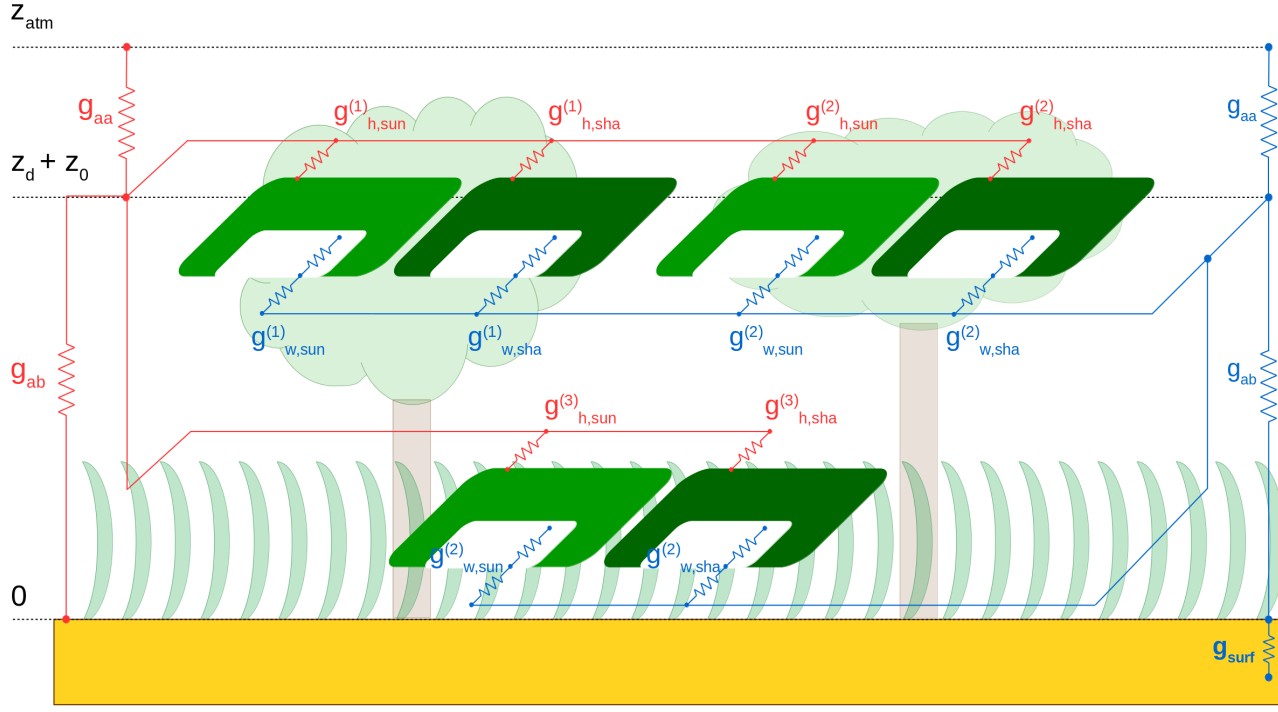

**Figure 2.** Networks of sensible (red) and latent (blue) heat exchange between the ground surface, the canopy and the atmosphere in the patch. Light green indicates the sunlit fraction of the cohorts, dark green the shaded fraction. $g_{\mathrm{aa}}$ is the aerodynamic conductance from the canopy air to the atmospheric reference level ($z_{\mathrm{atm}}$). $z_0$ and $z_{\mathrm{d}}$ are respectively roughness length and zero plane displacement. $g_{\mathrm{ab}}$ is the aerodynamic conductance for heat/moisture flux from the ground surface to the canopy air. $g_{\mathrm{surf}}$ is the surface conductance for moisture flux. $g^{(i)}_{\mathrm{h,sun[sha]}}$ is the conductance for sensible heat transport from the sunlit [shaded] part of cohort $i$ to the canopy air. $g^{(i)}_{\mathrm{w,sun[sha]}}$ is the conductance for moisture transport from the sunlit [shaded] part of cohort $i$ to the canopy air, with the contributions from stomatal conductance and leaf boundary layer conductance represented explicitly. A dry canopy ($f_{\mathrm{wet}} = 0$) is represented for clarity.







**Figure 3.** Stomatal conductance as a function of water vapor deficit at the leaf surface.



**Figure 4.** Factor limiting plant water uptake as a function of volumetric soil water content. The dashed, vertical lines represent, from left to right, the volumetric soil water content at a wilting matric potential of $-150\,\mathrm{m}$, at a wilting matric potential of $-45\,\mathrm{m}$, and at field capacity.

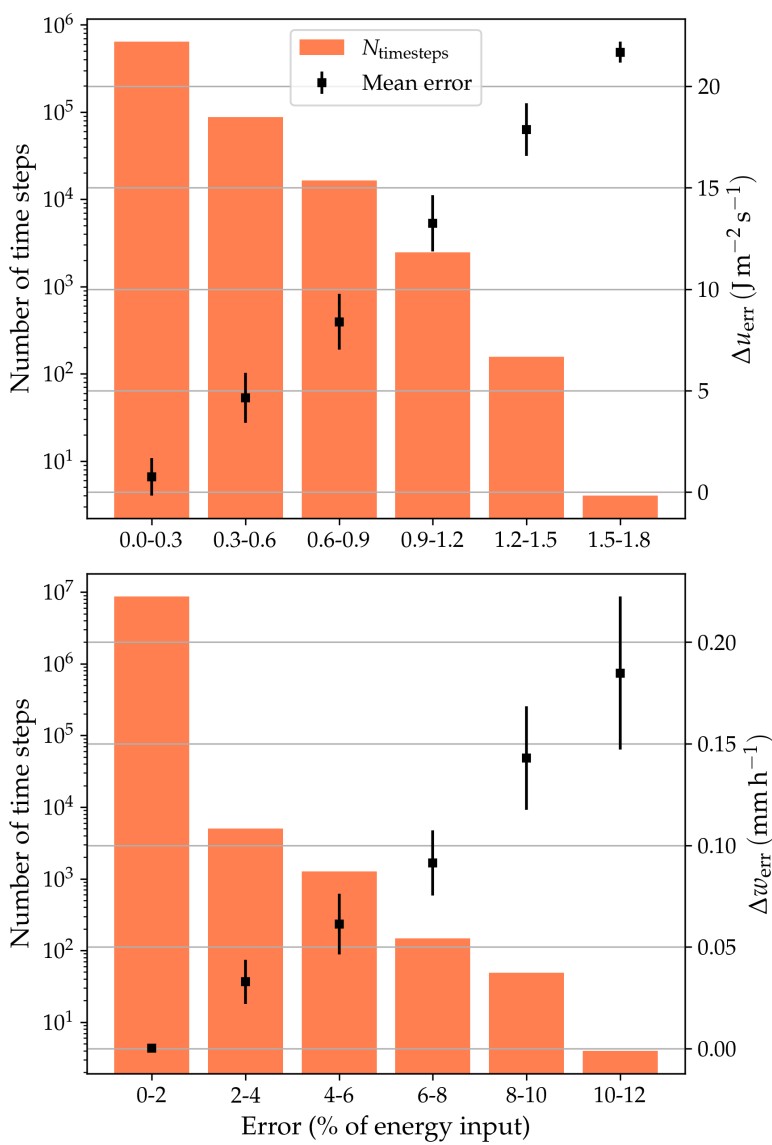

**Figure 5.** The histograms show the energy conservation error (upper panel) and the water conservation error (lower panel), as a percentage of the energy input, incurred at every time step. The symbols indicate the mean absolute error corresponding to each bin. The error bars indicate $\pm 1\sigma$ around the mean. The plots are derived from data from the historical period of all LSM simulations in this study.

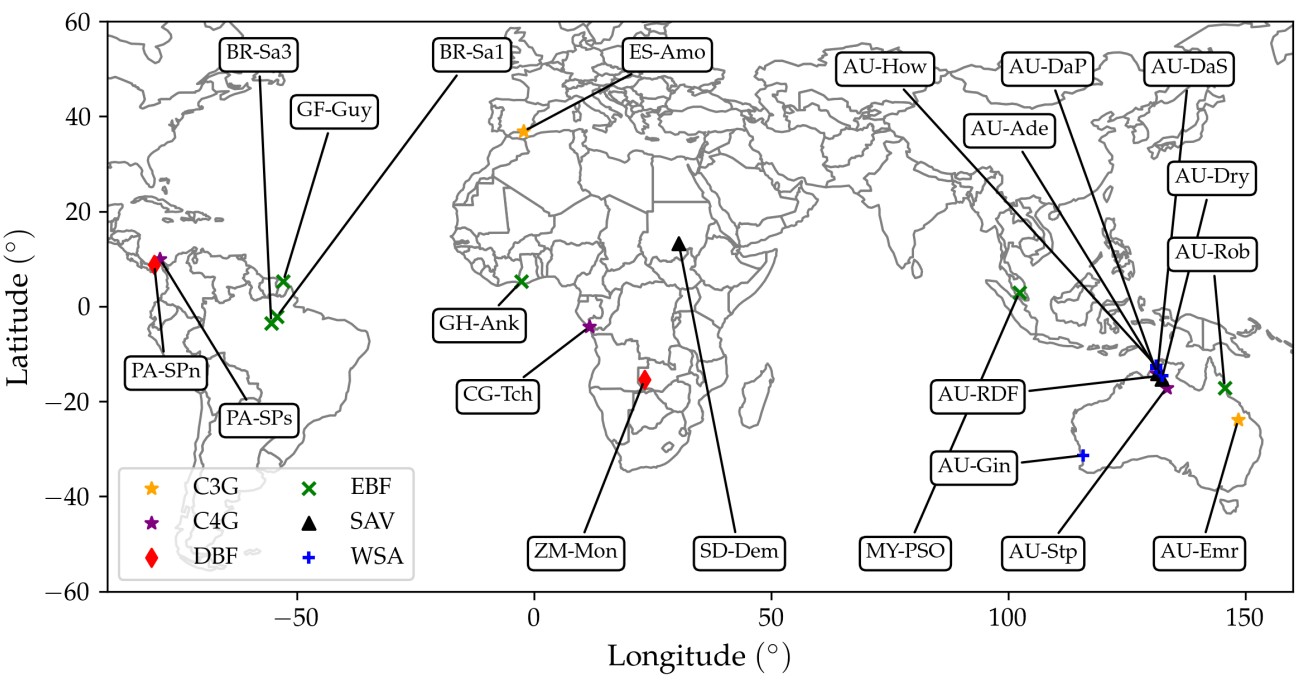

**Figure 6.** Fluxnet sites selected for model evaluation. Different symbols indicate different land cover types. The sites are labeled according to their site code (Table 1).





**Figure 7.** Observed (dashed lines) and simulated (continuous lines) annual cycles of sensible and latent heat flux at four selected sites: Gingin (AU-Gin, upper left), Daly River savanna (AU-DaS, upper right), Santarem Km. 67 (BR-Sa1, lower left), and Guyana (GF-Guy, lower right).



**Figure 8.** Monthly-averaged diurnal cycle of sensible and latent heat flux at the AU-DaP (upper panels) and AU-DaS sites (lower panels) in selected months. The red and blue lines represent simulated sensible and latent heat fluxes respectively. The shaded areas around each curve delimit one standard deviation above and below it. The symbols represent monthly averaged fluxes. The error bars indicate a $\pm 1\sigma$ deviation from the observed mean.





**Figure 9.** Monthly-averaged diurnal cycle of sensible and latent heat flux at the BR-Sa1 (upper panels) and GF-Guy sites (lower panels) in selected months. The red and blue lines represent simulated sensible and latent heat fluxes respectively. The shaded areas around each curve delimit one standard deviation above and below it. The symbols represent monthly averaged fluxes. The error bars indicate a ±1σ deviation from the observed mean.

**Figure 10.** Performance of the CLM/BB (left) and the CLM/Med (right) runs for sensible heat flux. The Taylor diagrams (Taylor, 2001) on the upper panels summarize the degree of agreement between observed and simulated hourly fluxes by relying on the geometrical relationship between the "centered pattern" root mean square difference (defined as $E'^2 = \mathrm{RMSE}^2 - \mathrm{Bias}^2$), the correlation coefficient, and the standard deviation of observed and modeled data: $E'^2 = \sigma_o^2 + \sigma_m^2 - 2\sigma_o\sigma_m r$. Each point in the polar diagram represents a simulation. The radial coordinate indicates the ratio between modeled and observed standard deviations. The correlation between observed and modeled values is encoded by the polar angle; it decreases counterclockwise from $r = 1$ (perfect correlation) for points situated on the $x$- axis to $r = 0$ for points situated on the $y$- axis. The distance between a point in the diagram and the reference value 1 on the $x$- axis equals the centered pattern RMSE normalized by the standard deviation of the observed values, $E'/\sigma_o$, and is therefore a measure of the agreement between observed and simulated data. The scatter plots (lower panels) show a direct comparison of observed and modeled monthly averaged fluxes. The different symbols refer to different land cover types: savanna (SAV), woody savanna (WSA), $C_3$ grasslands (C3G), $C_4$ grasslands (C4G), evergreen broadleaf forest (EBF), and deciduous broadleaf forest (DBF).



**Figure 11.** Performance of the CLM/BB (left) and the CLM/Med (right) runs for latent heat flux. The Taylor diagram shows statistical metrics calculated from hourly observed and simulated fluxes. The scatter plots show a direct comparison of observed and modeled monthly averaged fluxes. The different symbols refer to different land cover types: savanna (SAV), woody savanna (WSA), $C_3$ grasslands (C3G), $C_4$ grasslands (C4G), evergreen broadleaf forest (EBF), and deciduous broadleaf forest (DBF).





**Figure 12.** Average performance statistics for each model configuration, obtained from modeled and measured hourly (top row) and monthly (bottom row) fluxes of sensible and latent heat flux. The different symbol shapes represent different water uptake functions (squares: NOAH type; circles CLM type; crosses: modified CLM type; triangles: SSiB type), and the different colors represent different stomatal conductance schemes (green: Ball-Berry type; black: Medlyn). The red star represents average performance statistics for monthly latent heat fluxes derived from the standard LPJ-GUESS simulation.



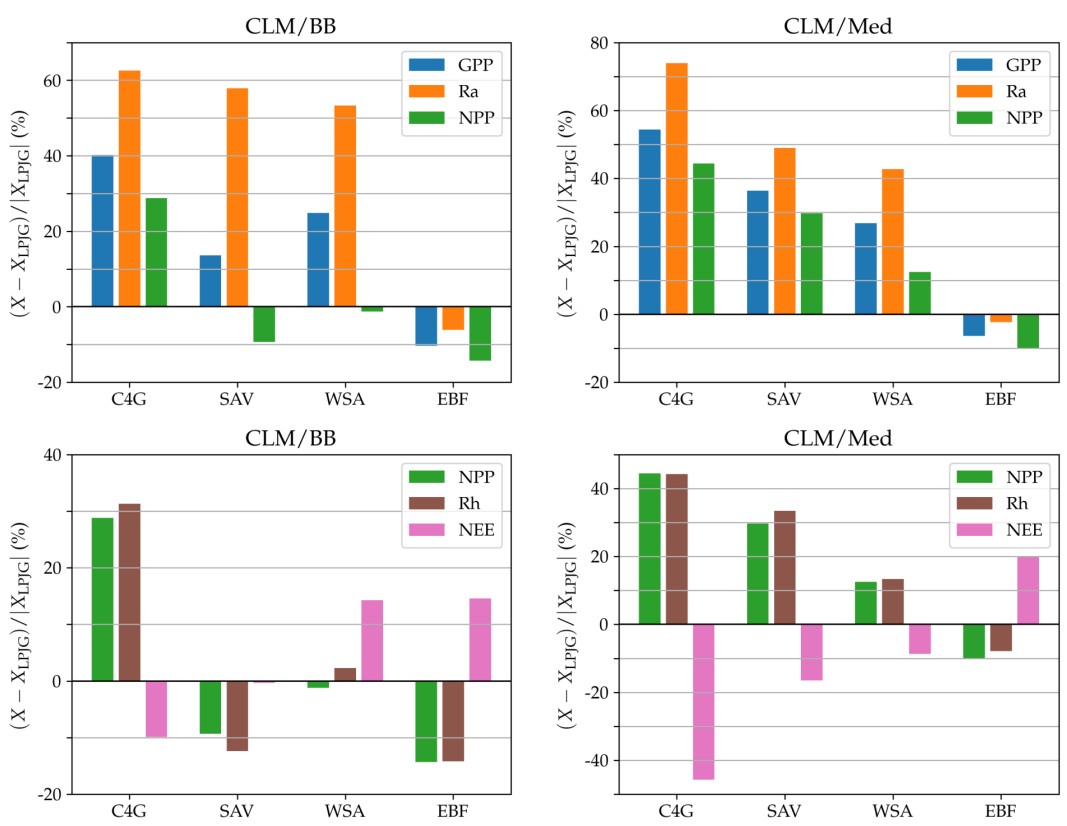

**Figure 13.** Top panels: percent change in average gross primary production (blue), autotrophic respiration (orange), and net primary production (green), simulated by the LSM version, with respect to standard LPJ-GUESS. Bottom panels: percent change in predicted average net primary production (green), heterotrophic respiration (brown) and net ecosystem exchange (pink).

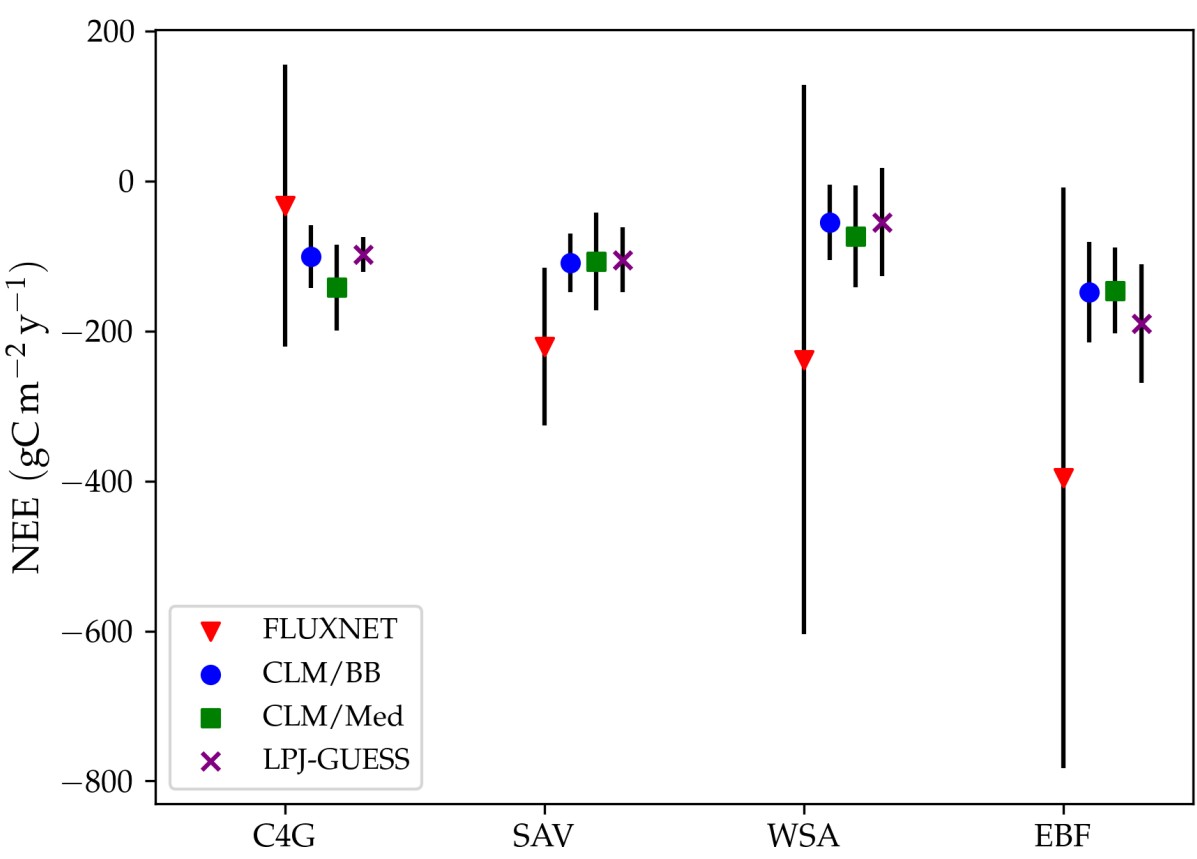

**Figure 14.** Comparison between observed and modeled annual NEE. The symbols indicate averages over sites the same land cover type. Red triangles correspond to flux tower $CO_2$ measurements. Blue dots, green squares and purple crosses correspond, respectively, to the CLM/BB, CLM/Med and standard LPJ-GUESS simulations. The bars represent one standard deviation above and below the average.



**Figure 15.** Effect of temperature (T) on modeled $V_{\mathrm{max}}$; $N_{\mathrm{leaf,opt}}$: leaf nitrogen content necessary to attain the maximum carboxylation rate; $N_{\mathrm{leaf}}$: representative leaf nitrogen concentration; $\hat{V}_{\mathrm{max}}$: normalized maximum carboxylation rate without nitrogen limitation; $\hat{V}_{\mathrm{max,lim}}$: normalized maximum carboxylation rate with nitrogen limitation. The histograms show the frequency of temperatures in the AU-DaP simulation; $T_{\mathrm{atm,day}}$: daily average of air temperature; $T_{\mathrm{leaf,dt}}$: daytime average of leaf temperature. The shaded area indicates the temperature range where $V_{\mathrm{max}}$ is nitrogen-limited.





**Table A1.** Ecosystem structure

| Parameter | Description | Units |
|---|---|---|
| $\text{LAI}_\text{c}$ | Patch canopy leaf area index | $\text{m}^2/\text{m}^2$ |
| $\text{PAI}_\text{c}$ | Patch canopy plant matter area index | $\text{m}^2/\text{m}^2$ |
| $\text{PAI}^{(l)}$ | Plant matter area index of patch canopy layer $l$ | $\text{m}^2/\text{m}^2$ |
| $\text{LAI}^{(i)}$ | Leaf area index of cohort $i$ in the patch | $\text{m}^2/\text{m}^2$ |
| $\text{SAI}^{(i)}$ | Stem area index of cohort $i$ in the patch | $\text{m}^2/\text{m}^2$ |
| $\text{LAI}^{(i,l)}$ | Leaf area index of cohort $i$ in patch canopy layer $l$ | $\text{m}^2/\text{m}^2$ |
| $\text{SAI}^{(i,l)}$ | Stem area index of cohort $i$ in patch canopy layer $l$ | $\text{m}^2/\text{m}^2$ |
| $h_\text{c}$ | Patch canopy height | m |





**Table A2.** Energy balance

| Parameter | Description | Units |
|---|---|---|
| $S_{[\text{sun,sha}]}$ | Shortwave radiation absorbed by the [sunlit, shaded] part of the canopy | $\mathrm{Wm^{-2}}$ |
| $L_{[\text{sun,sha}]}$ | Net longwave radiation emitted by the [sunlit, shaded] part of the canopy | $\mathrm{Wm^{-2}}$ |
| $H_{[\text{sun,sha}]}$ | Sensible heat flux from the [sunlit, shaded] canopy to the canopy air space | $\mathrm{Wm^{-2}}$ |
| $\lambda E_{[\text{sun,sha}]}$ | Sensible heat flux from the [sunlit, shaded] canopy to the canopy air space | $\mathrm{Wm^{-2}}$ |
| $S_{\mathrm{g}}$ | Shortwave radiation absorbed by the ground surface | $\mathrm{Wm^{-2}}$ |
| $L_{\mathrm{g}}$ | Net longwave radiation emitted by the ground surface | $\mathrm{Wm^{-2}}$ |
| $H_{\mathrm{g}}$ | Sensible heat flux from the ground surface to the canopy air space | $\mathrm{Wm^{-2}}$ |
| $\lambda E_{\mathrm{g}}$ | Latent heat flux from the ground surface to the canopy air space | $\mathrm{Wm^{-2}}$ |
| $G$ | Heat flux conducted into the soil | $\mathrm{Wm^{-2}}$ |
| $H^{\uparrow}$ | Sensible heat flux from the canopy air space to the atmosphere | $\mathrm{Wm^{-2}}$ |
| $\lambda E^{\uparrow}$ | Sensible heat flux from the canopy air space to the atmosphere | $\mathrm{Wm^{-2}}$ |
| $T_{[\text{sun,sha}]}$ | Temperature of the [sunlit,shaded] part of the canopy | $^{\circ}\mathrm{C}$, K |
| $T_{\mathrm{g}}$ | Temperature of the ground surface | $^{\circ}\mathrm{C}$, K |
| $T_{\mathrm{ca}}$ | Temperature of the canopy air | $^{\circ}\mathrm{C}$ |
| $T_{\mathrm{atm}}$ | Temperature of the atmosphere at the reference level | $^{\circ}\mathrm{C}$ |
| $q_{[\text{sun,sha}]}$ | Specific humidity of the stomatal cavity air for [sunlit,shaded] leaves | kg/kg |
| $q^{*}(T_{\mathrm{g}})$ | Saturated specific humidity at the ground surface temperature | kg/kg |
| $\alpha$ | Ground surface specific humidity as a fraction of $q^{*}(T_{\mathrm{g}})$ | - |
| $q_{\mathrm{ca}}$ | Specific humidity of the canopy air | kg/kg |
| $q_{\mathrm{atm}}$ | Specific humidity of the atmosphere at the reference level | kg/kg |
| $g_{\mathrm{b}}$ | Leaf boundary layer conductance | $\mathrm{ms^{-1}}$ |
| $g_{\mathrm{w},[\text{sun,sha}]}$ | Conductance to water vapor between the [sunlit,shaded] canopy and the canopy air | $\mathrm{ms^{-1}}$ |
| $g_{\mathrm{surf}}$ | Conductance to water vapor from near-surface soil pores to the ground surface | $\mathrm{ms^{-1}}$ |
| $g_{\mathrm{ab}}$ | Aerodynamic conductance from the ground surface to the canopy air | $\mathrm{ms^{-1}}$ |
| $g_{\mathrm{aa}}$ | Aerodynamic conductance from the canopy air to the atmosphere reference level | $\mathrm{ms^{-1}}$ |
| $f_{\mathrm{wet}}$ | Wet fraction of the canopy | - |
| $\eta_{[\text{sun,sha}]}$ | Factor limiting evaporation from the [sunlit,shaded] canopy | - |
| $z_{0}$ | Canopy roughness length | m |
| $z_{\mathrm{d}}$ | Zero plane displacement height | m |
| $\Delta z^{(1)}$ | Top soil layer thickness | m |
| $T_{\mathrm{s}}^{(1)}$ | Temperature of top soil layer | $^{\circ}\mathrm{C}$ |
| $\kappa_{\mathrm{s}}^{(1)}$ | Thermal conductivity of the top soil layer | $\mathrm{Wm^{-1}K^{-1}}$ |



**Table A3.** Radiative transfer

| Parameter | Description | Units |
|---|---|---|
| $\mathrm{PAI}_{\mathrm{c},[\mathrm{sun},\mathrm{sha}]}$ | Plant matter area index of the [sunlit, shaded] part of the canopy | $\mathrm{m}^2/\mathrm{m}^2$ |
| $\mathrm{PAI}_{[\mathrm{sun},\mathrm{sha}]}^{(i)}$ | Plant matter area index of the [sunlit, shaded] part of cohort $i$ | $\mathrm{m}^2/\mathrm{m}^2$ |
| $\mathrm{LAI}_{[\mathrm{sun},\mathrm{sha}]}^{(i)}$ | Leaf area index of the [sunlit, shaded] part of cohort $i$ | $\mathrm{m}^2/\mathrm{m}^2$ |
| $I_{[\mathrm{D},\mathrm{d}]0}^{\downarrow}$ | Incoming [direct beam, diffuse] radiation | $\mathrm{Wm}^{-2}$ |
| $I_{\mathrm{D}}^{\downarrow}$ | Direct beam profile in the canopy | $\mathrm{Wm}^{-2}$ |
| $\hat{I}_{\mathrm{b}}^{[\downarrow,\uparrow]}$ | Normalized profile of [downwards, upwards] scattered direct beam in the canopy | - |
| $\hat{I}_{\mathrm{a}}^{[\downarrow,\uparrow]}$ | Normalized profile of [downwards, upwards] scattered diffuse atmospheric radiation in the canopy | - |
| $I^{\uparrow}$ | Outgoing shortwave radiation | $\mathrm{Wm}^{-2}$ |
| $k$ | Direct beam extinction coefficient | - |
| $\omega$ | Direct beam scattering coefficient | - |
| $S_{[\mathrm{D},\mathrm{d}]}^{(l)}$ | [Direct, diffuse] shortwave radiation absorbed by canopy layer $l$ | $\mathrm{Wm}^{-2}$ |
| $S_{[\mathrm{sun},\mathrm{sha}]}^{(l)}$ | Shortwave radiation absorbed by the [sunlit, shaded] part of canopy layer $l$ | $\mathrm{Wm}^{-2}$ |
| $f_{[\mathrm{sun},\mathrm{sha}]}^{(l)}$ | Fraction of canopy layer $l$ that is [sunlit, shaded] | $\mathrm{Wm}^{-2}$ |
| $S_{[\mathrm{sun},\mathrm{sha}],\mathrm{vis}}^{(l)}$ | Visible radiation absorbed by the [sunlit,shaded] part of canopy layer $l$ | $\mathrm{Wm}^{-2}$ |
| $\mathrm{PAR}_{[\mathrm{sun},\mathrm{sha}]}^{(i)}$ | Photosynthetically active radiation absorbed by the [sunlit, shaded] part of cohort $i$ | $\mathrm{Wm}^{-2}$ |
| $\alpha_{[\mathrm{sun},\mathrm{sha}],[\mathrm{vis},\mathrm{nir}]}$ | Reflectivity of [sunlit, shaded] leaves in the [visible, infrared] waveband | - |
| $\tau_{[\mathrm{sun},\mathrm{sha}],[\mathrm{vis},\mathrm{nir}]}$ | Transmissivity of [sunlit, shaded] leaves in the [visible, infrared] waveband | - |
| $L^{\downarrow}$ | Incoming (atmospheric) longwave radiation | $\mathrm{Wm}^2$ |
| $L^{\uparrow}$ | Outgoing longwave radiation | $\mathrm{Wm}^2$ |
| $\gamma_{[\mathrm{sun},\mathrm{sha}]}$ | Effective thermal emissivity of the [sunlit, shaded] part of the canopy | - |





**Table A4.** Assimilation and stomatal conductance

| Parameter | Description | Units |
|---|---|---|
| $g_{s,[sun,sha]}^{(i)}$ | Stomatal conductance of the [sunlit,shaded] leaves of cohort $i$ | $ms^{-1}$ |
| $g_{min}$ | Minimum stomatal conductance | $ms^{-1}$ |
| $g_{1,BB}$ | Stomatal conductance parameter for the Ball-Berry model | - |
| $g_{1,Med}$ | Stomatal conductance parameter for the Medlyn model | $kPa^{0.5}$ |
| $A_{n,[sun,sha]}^{(i)}$ | Net photosynthetic assimilation rate of [sunlit,shaded] leaves of cohort $i$ | $\mu molC\,m^{-2}s^{-1}$ |
| $h_{s,[sun,sha]}$ | Fractional humidity at the surface of [sunlit, shaded] leaves | $kPa\,kPa^{-1}$ |
| $D_{s,[sun,sha]}$ | Water vapor deficit at the surface of [sunlit, shaded] leaves | $kPa$ |
| $c_{s,[sun,sha]}$ | Carbon dioxide concentration at the surface of [sunlit, shaded] leaves | $\mu mol\,mol^{-1}$ |
| $PAR_{day}^{(i)}$ | Daily photosynthetically active radiation absorbed by cohort $i$ | $J\,day^{-1}m^{-2}$ |
| $PAR_{[sun,sha],day}^{(i)}$ | Daily photosynthetically active radiation absorbed by the [sunlit, shaded] parts of cohort $i$ | $J\,day^{-1}m^{-2}$ |
| $V_{max,day}^{(i)}$ | Maximum carboxylation rate of cohort $i$ | $\mu molC\,m^{-2}day^{-1}$ |
| $f_v$ | Slope of the relationship between $PAR^{(i)}$ and $V_{max}^{(i)}$ | $\mu molC\,J^{-1}$ |
| $T_{leaf,dt}^{(i)}$ | Daytime average temperature of cohort $i$ | $°C$ |
| $n_{dt}$ | Number of subdaily periods at the end of the simulation day | - |
| $V_{max,[sun,sha],day}^{(i)}$ | Maximum carboxylation rate of the [sunlit, shaded] part of cohort $i$, per unit patch area | $\mu molC\,m^{-2}day^{-1}$ |
| $V_{max,[sun,sha],leaf}^{(i)}$ | Maximum carboxylation rate of the [sunlit, shaded] part of cohort $i$, per unit leaf area | $\mu molC\,m^{-2}s^{-1}$ |
| $LAI_{[sun,sha],dt}^{(i)}$ | Daytime average of the [sunlit,shaded] LAI for cohort $i$. | $m^2m^{-2}$ |
| $\beta$ | Water stress factor limiting the assimilation rate | - |
| $r^{(j)}$ | Fraction of roots in soil layer $j$ | - |
| $W_{av}^{(j)}$ | Soil water uptake function (layer $j$) | - |





**Table A5.** Soil physics

| Parameter | Description | Units |
|---|---|---|
| $T_\mathrm{s}$ | Soil temperature | $^\circ\mathrm{C}$ |
| $c_\mathrm{h}$ | Volumetric heat capacity | $\mathrm{Jm^{-3}{}^\circ C^{-1}}$ |
| $\kappa_\mathrm{s}$ | Thermal conductivity | $\mathrm{Wm^{-1}{}^\circ C^{-1}}$ |
| $\theta^{(l)}$ | Volumetric water content of soil layer $l$ | $\mathrm{m^3 m^{-3}}$ |
| $\theta_\mathrm{wilt}$ | Volumetric soil water content at wilting point | $\mathrm{m^3 m^{-3}}$ |
| $\theta_\mathrm{fc}$ | Volumetric soil water content at field capacity | $\mathrm{m^3 m^{-3}}$ |
| $\psi^{(l)}$ | Matric potential of soil layer $l$ | $\mathrm{m}$ |
| $\psi_\mathrm{wilt}$ | Matric potential of soil water at wilting point | $\mathrm{m}$ |
| $\psi_\mathrm{fc}$ | Matric potential of soil water at saturation point | $\mathrm{m}$ |
| $\gamma_\mathrm{w}$ | Hydraulic conductivity | $\mathrm{m\,s^{-1}}$ |
| $\lambda_\mathrm{w}$ | Hydraulic diffusivity | $\mathrm{m^2 s^{-1}}$ |
| $S_\theta$ | Volumetric soil water uptake sink term | $\mathrm{m^3 m^{-3}}$ |





*Author contributions.* AA had the original idea and motivated the development. DMB designed the model augmentations described in this work with input from all the coauthors. DMB implemented the code, ran the experiments and performed the model evaluation analysis. All coauthors provided input regarding the analysis of the results and the discussion, and helped shape the final form of the manuscript.

*Competing interests.* The author declare no competing interests.





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
