# Peer review of "LPJ-GUESS/LSMv1.0: A next generation Land Surface Model with high ecological realism"

_Geoscientific Model Development, 2022_

## Author Comment (AC1)

We thank the anonymous reviewer for the kind words about our work and for their thorough revision of our manuscript. We will address the *reviewer's comments* in order.

As for the water (and latent heat) fluxes issue, the authors invested a lot of efforts to look for reasons with different stomatal conductance models and water uptake functions. I think an expanded discussion about the interactions between vegetation settings from the LPJ-GUESS model and physical environments derived from the new biophysical processes would be helpful. This would also be helpful for understanding the differences in productivity between the original LPJ-GUESS and it coupled with LSM.

LPJ-GUESS calculates many vegetation properties dynamically, including key photosynthetic quantities such as Vmax. The new scheme interacts with these processes, and changes in simulated fluxes and productivity arise as a result of the interplay of many factors. We found that the biggest discrepancies between the standard model and the LSM version are due to the latter using leaf temperature, rather than daily averaged air temperature, for photosynthetic calculations. Leaf temperature can be several degrees above air temperature, affecting the photosynthetic rate directly, via the temperature inhibition factor, but also (and most dramatically in the case of C4 grasses) indirectly, by mitigating nitrogen limitation of photosynthesis. This is seen in simulations where we restrict establishment to grassy PFTs and C4 grasses emerge as the dominant grass type. While this effect seems to be far less pronounced in a potential natural vegetation simulation, due to trees entering the competition, it may be important when simulating C4 crops (since these grow without competition in well-watered conditions). The effect of leaf temperature on the biochemical nitrogen limitation of photosynthesis, and how it affects C4 grass productivity in competition and no-competition situations, is discussed in detail in section 4.2.

We now discuss in the supplement how the new simulated physical environments affect soil carbon and nitrogen build-up and how the PAR absorption calculations compare for the old and the new schemes (see attached document). We also discuss how the different stomatal conductance and soil water uptake formulations lead to differences in simulated soil water content that can alter the PFT composition (new section 3.4.4, attached).

We will now address the minor comments.

1. Line 146: the unit of lambda. I think "C-1" is not necessary. The number of 2.44x106 has included an assumption of normal water temperature

We thank the reviewer for spotting the mistake in the units. Lambda represents the amount of energy that it takes to evaporate a unit mass of liquid water, so the correct units should be of energy per unit mass (J/kg). We will correct the text accordingly.

2. Lines 380 and 388: How do the vegetation conditions vary with the actual vegetation? I think the vegetation and soil states (equilibrium or not) may affect NEE, as shown in Fig. 14 that the simulations are close to equilibrium state.

4. Lines 572: measured NEE is more negative than those simulated. I think it is related to how far the vegetation is from its theoretical equilibrium state. Disturbances also play a role here. For example, at equilibrium state, an ecosystem will have a zero NEE (or fluctuated around zero) if the system has no disturbances. However, if it is equilibrated with a particular disturbance regime (e.g., a given fire frequency distribution), the system must have a negative NEE that is to counter the carbon release at

disturbance events. In long-term, it is still carbon neutral. This is the pattern this paper showed in this section that observations have higher carbon sink (more negative NEE) than the simulated (with 500 years of model run).

We agree with the reviewer that the discrepancy in NEE probably reflects the fact that the simulated carbon pools are close to equilibrium with the synthetic climate data used to spin up the model, and that this set up differs from the actual situation at the different sites (this is seen also in the difference between the variabilities of observed and simulated fluxes). In a standard LPJ-GUESS simulation, the spinup procedure is designed to bring carbon pools close to equilibrium with pre-industrial conditions, before simulating the historical period of rising temperatures and CO2 concentrations. Here, our primary goal was to evaluate the sub-daily latent and sensible heat fluxes calculated by the augmented model against observations, and then compare the ecosystem-related predictions (structure, composition and fluxes) of the new LSM model with those made by the standard (non-LSM) version. Therefore, differences between simulated and measured NEE are to be expected because we did not attempt to fully reproduce or account for site history, including age, disturbance, and legacies arising from historical trends in CO2 concentration. Evaluating simulated ecosystem productivity and carbon fluxes against observations, as well as the differences between the standard and the augmented model in regional and global scales, is the object of future work, but was out of the scope of this paper.

We suggest adding the following sentence in section 3.4.4:

"These discrepancies between observed and simulated NEE magnitude and variability reflect the fact that, in the simulations, the carbon pools are all close to equilibrium with the climate and atmospheric CO2 concentration as a result of the spinup procedure described in section 3.2 and the supplement. Differences are to be expected because we did not attempt to fully reproduce or account for site history, including age, disturbance, and legacies arising from historical trends in CO2 concentration."

Additionally, we added a discussion of the build-up of soil organic matter pools during the spinup phase in the supplement.

3. Line 506 section Ecosystem structure and function: They are related to the settings of the vegetation model since the structure is highly dynamic. How to make them consistent with each case and the measurement data?

For this work, we used the standard set of PFTs provided with LPJ-GUESS, which aims to capture the main functional traits of different vegetation classes in global simulations. Our aim in this paper was to evaluate how well the model (including the standard set of PFTs) simulated energy fluxes, rather than fitting the PFT parameters at each site specifically to achieve a best fit between model and observations. We chose this approach since the end goal is to perform regional and global runs. Also, see response to previous comment.

5. Time steps of and growth (yearly) and SOM (daily): how LAI dynamics and heterotrophic respiration are calculated? Usually, LAI should be updated daily and Rh hourly (or half hourly). Are they connected with plant growth and SOM dynamics at each step, respectively? This just need to clarify. I may miss the description.

- In this work we did not modify the original model's growth and phenology routines. At present, the allocation of carbon to the different plant structures (what we call growth) is simplified in the model, happening annually. The phenological status is calculated and updated daily according to the phenology

of the different PFTs (Raingreen, Summergreen, Evergreen, as described in appendix B4 of Smith et al. (2014), https://doi.org/10.5194/bg-11-2027-2014). This is stated at the beginning of section 2.2 of the model description and in Figure 1.

- Rh is updated daily in the current version of the model. Even though Rh can vary on diurnal timescales, we focused on averages of yearly NEE over the measuring period, so we assumed calculating Rh on a daily basis was sufficient for our purposes. We stress, however, that photosynthesis and autotrophic respiration are calculated on a subdaily basis, consistent with the canopy energy balance, and accumulated to calculate daily NPP, and then subtracted from daily RH to calculate daily NEE at the end of the day. To clarify this, we added the following text at the beginning of section 2.2.4:

"The net photosynthetic assimilation is accumulated over the diurnal cycle and subtracted from heterotrophic respiration (Rh, computed daily) to calculate daily net ecosystem exchange (NEE)."

6. In discussion, for the water uptake functions and C4 grass carbon assimilation simulations, they are phenomenological equations in the model that directly link soil water availability to leaf functions. A discussion of actual plant-soil hydraulics would be helpful for understanding why they happen and why we don't have to spend much time to tune these functions.

We agree with the reviewer that these expressions are oversimplifications of the very complex process of stomatal regulation. To stress this fact in a concise way, we suggest adding the following text when introducing the soil moisture stress factor (end of section 2.2.4):

"This type of formulations, which are widely use in LSMs (see Damour et al, 2010, for an overview), are phenomenological relationships that attempt to capture the response of plants to water stress in a rather simplified way (Egea et al., 2011; De Kauwe et al., 2013). Transpiration of soil water by plants is primarily driven by the water potential gradient along the soil-plant-atmosphere continuum. Plants regulate this gradient by opening and closing their stomata in response to environmental factors, including leaf water potential, vapor pressure deficit, and soil water availability, in a way that depends on their hydraulic strategy (a detailed discussion can be found in Lambers et al, 2008). Including a more explicit representation of soil-plant-air hydraulics as well as physiological constraints in a stomatal conductance model has been shown to perform better than the above formulations under soil water stress conditions (Bonan et al., 2014). However, implementing these more complex models in ESMs remains a challenge due to a lack of data for broader applicability and computational efficiency tradeoffs (Clark et al., 2015)."

**7. I just realized the codes of LPJ-GUESS are still not publicly available. Maybe, this question should be asked by the handling editor. Does it comply with the journal's policy?**

As specified in the "Code availability section" of the manuscript, "The [LPJ-GUESS] source code can be made available with a collaboration agreement under the acceptance of certain conditions. For this reason, a DOI for the model code cannot be provided. The code with the augmentations developed for this paper is available to the editor and reviewers via a restricted link, on the condition that the code is used only for review purposes, and is deleted after the review process." In any case, we are happy to provide the code upon request.

This is consistent with point #2 in the core principles of the journal's code and data policy (https://www.geoscientific-model-development.net/policies/code\_and\_data\_policy.html):

"Where the authors cannot, for reasons beyond their control, publicly archive part or all of the code and data associated with a paper, they must clearly state the restrictions. They must also provide confidential access to the code and data for the editor and reviewers in order to enable peer review. The arrangements for this access must not compromise the anonymity of the reviewers. All manuscripts which do not make code and data available at this level are to be rejected. Where only part of the code or data is subject to these restrictions, the remaining code and/or data must still be publicly archived. In particular, authors must make every endeavour to publish any code whose development is described in the manuscript."

**1 Ecosystem composition and function**

We compared the predictions of the LSM simulations to standard LPJ-GUESS for species composition and a number of ecosystem structure and function variables.

The emerging ecosystem composition in both LSM runs is similar to the standard LPJ-GUESS prediction over forests and grasslands, but it is sensitive to the choice of stomatal conductance scheme at savanna and woody savanna sites (Table 1). Figure 1 shows the LAI evolution of the established PFTs over the spinup period for the CLM/BB, CLM/Med and standard LPJ-GUESS simulations at three selected sites. All three simulations predict a  $C_4$  grassland at PA-SPs, but LAI values are much higher in the LSM simulations ( $\sim 11$ ) than the LPJ-GUESS prediction ( $\sim 6.5$ ). At BR-Sa1 (a tropical rainforest), the species composition is similar for the three simulations, but LAI values are lower in the LSM runs ( $\sim 5.5 \text{ vs} \sim 6.2$ ). At AU-Dry, the use of different stomatal conductance schemes causes a shift in PFT composition. The BB simulation favors evergreen trees, while the PFT mix is dominated by raingreen trees in the Med simulation, a prediction closer to standard LPJ-GUESS. We found this behaviour to be representative of how the soil water uptake factor and the stomatal conductance scheme determine the PFT composition at most savanna and woody savanna sites in the LSM simulations. A stronger limitation on transpiration (e.g. the NOAH-type water uptake factor or the Ball-Berry stomatal conductance model) results in higher soil water content throughout the year, which promotes stronger growth of every even trees (graphs of the average soil water content over the last 100 years of spinup are provided in the supplement).

Model predictions for the rest of the selected variables are shown in Table 1. The two C3 grassland sites show different behaviour with respect to ecosystem productivity and respiration. At AU-Emr, LSM simulations predict substantially lower gross primary production (GPP) and autotrophic respiration ( $R_a$ ) than standard LPJ-GUESS, which results in lower estimates of net primary production (NPP). This site is a net carbon source (positive NEE) in all three simulations, which agrees with observations. At ES-Amo, the NPP increase in the LSM runs does not overcome the decrease in heterotrophic respiration ( $R_h$ ), resulting in a slightly enhanced carbon sink compared to standard LPJ-GUESS.

The two deciduous broadleaf forest sites show slight differences between runs, but the fluxes are similar in all three simulations, resulting in carbon sinks of  $-58 \,\mathrm{gC}^{-2} \mathrm{y}^{-1} \mathrm{m}^{-2}$  (standard LPJ-GUESS and CLM/Med), and  $-71 \,\mathrm{gC}^{-2} \mathrm{y}^{-1} \mathrm{m}^{-2}$  (CLM/BB). This result is inconsistent with measurements at the site, which indicate a carbon source of  $143 \,\mathrm{gC}^{-2} \mathrm{y}^{-1} \mathrm{m}^{-2}$ .

Differences in simulated carbon fluxes between standard LPJ-GUESS and the CLM/BB and CLM/Med runs for the reamaining land cover types are summarized in Fig. 2. Both LSM runs predict, on average, higher GPP and  $R_a$ values than the non-LSM simulation over C4 grasslands, savanna and woody savanna sites. This results in an increased average NPP in C4 grasslands (~ 18% in the CLM/BB run and ~ 31% in the CLM/Med run), and a decreased average NPP at woody savanna sites (~ -11% and ~ -7% in the CLM/BB and the CLM/Med runs, respectively). At savanna sites, the increase in NPP in both LSM simulations is similar (~ 10%), but the increase in  $R_a$  is much higher for CLM/BB, which leads to changes in NPP of ~ -8% in the CLM/BB run and  $\sim 5\%$  in the CLM/Med run. At forest sites, the balance between decreased values of GPP and  $R_{\rm a}$  results in lower NPP values in the LSM simulations. Average values of  $R_{\rm h}$  in the CLM/BB simulation increase over C4 grasslands, and decrease for all the other three land cover types. The CLM/Med simulation shows the same pattern except over savanna sites, where  $R_{\rm a}$  increases by  $\sim 6\%$  with respect to standard LPJ-GUESS. This causes an average NEE change of  $\sim 116\%$  for the CLM/BB run, turning savanna into an average net source, and  $\sim -200\%$  for the CLM/Med run, an enhanced carbon sink.

The above-described discrepancies between standard LPJ-GUESS and the LSM versions stem from the different physical environments simulated in the models. Calculating assimilation at the newly simulated canopy temperature, rather than the air temperature, can lead to either higher or lower productivity, depending on the optimal photosynthetic temperature ranges of each PFT. Canopy temperature also affects autotrophic respiration, while differences in the simulated soil humidity and temperature impact organic matter decomposition rates and heterotrophic respiration. The combination of these effects results in differences in simulated carbon and nitrogen pools and NEE (we have included a comparison between soil carbon and nitrogen pools simulated by standard LPJ-GUESS and LPJ-GUESS/LSM in the supplement).

The large relative changes in NEE between simulations result from small discrepancies in magnitude. Figure 3 shows a comparison between land-cover averages of measured and modeled NEE for  $C_4$  grasslands, savanna, woody savanna and evergreen forests. Average measured NEE is negative for all land cover types, and substantially more negative than in the simulations for savanna, woody savanna and evergreen broadleaf forests, implying an average underestimation of the C sink by the models at these sites. At C4 sites simulations predict NEE values between  $-88 \,\mathrm{gCm^{-2}y^{-1}}$  and  $-111 \,\mathrm{gCm^{-2}y^{-1}}$ , while observations indicate a less negative value of  $-33 \,\mathrm{gCm^{-2}y^{-1}}$ . For savanna, measured NEE is  $-221 \text{ gCm}^{-2}\text{y}^{-1}$ , while simulations predict an average between  $-34 \text{ gCm}^{-2}\text{y}^{-1}$  and  $-48 \text{ gCm}^{-2}\text{y}^{-1}$ . For woody savanna, measured NEE averages to  $-238 \,\mathrm{gCm^{-2}y^{-1}}$ , while simulated fluxes range between  $-36 \,\mathrm{gCm^{-2}y^{-1}}$ and  $2 \,\mathrm{gCm}^{-2} \mathrm{y}^{-1}$ . Measured fluxes at every even broadleaf forests are, on average,  $-396 \,\mathrm{gCm^{-2}y^{-1}}$ , while simulations predict average fluxes between -98 and  $-130 \,\mathrm{gCm^{-2}y^{-1}}$ . However, this is the result of very large negative values measured at AU-Rob and MY-PSO (Table 1). In general, differences in simulated fluxes between standard LPJ-GUESS and the two LPJ-GUESS/LSM simulations are small compared to the magnitude of observed fluxes, and the interannual and cross-site variability of the measured fluxes is much greater than in the simulations. These discrepancies between observed and simulated NEE magnitude and variability reflect the fact that, in the simulations, the carbon pools are all close to equilibrium with the climate and atmospheric  $CO_2$  concentration as a result of the spinup procedure described in section 3.2. Differences between observed and simulated NEE values are to be expected because we did not attempt to reproduce site history, including age, disturbance, and legacies arising from historical trends in  $CO_2$  concentration.

Figure 1: LAI values for the spinup period at three selected sites: PA-SPs (panels a-c), BR-Sa1 (panels d-f), and AU-Dry (panels g-i). The columns correspond to standard LPJ-GUESS (right), CLM/BB (center) and CLM/Med (right) simulations. The time series were smoothed for better visualization by applying a 15-year running average.

|       | Med   | I         | Ι      | 96 | 61     | 98     | 98     | 12        | 20     | 40     | 2      | Ι         | ъ      | 7      | 4      | 9      | 4      | 5
C | က      | 7      | e
S | 14     |                   |             |
|-------|-------|-----------|--------|-----------|--------|--------|--------|-----------|--------|--------|--------|-----------|--------|--------|--------|--------|--------|--------|--------|--------|--------|--------|-------------------|-------------|
| C4G   | BB    | I         | Ι      | 96 | 56     | 98     | 98     | 7         | 11     | 13     | 2      | I         | 8      | 9      | 4      | 2      | 9      | 7      | S      | ъ      | e
C | ×      | en as             | font.       |
|       | LPJ-G | I         | Ι      | 93        | 77     | 89     | 95     | 18        | 26     | 27     | 19     | I         | 17     | 21     | 3      | ъ      | ъ      | 2      | 2      | 4      | 33     | 12     | site. e iv | in bold     |
|       | Med   | 24 | 63     | 1         | I      | I      | I      | I         | I      | I      | I      | ъ         | I      | I      | I      | I      | I      | I      | I      | I      | I      | I      | r each            | lighted     |
| C3G   | BB    | 23        | 68     | I         | I      | I      | I      | I         | I      | I      | I      | c,        | I      | I      | I      | Ι      | Í      | Í      | Ι      | I      | I      | I      | cted fo           | s high      |
|       | LPJ-G | 61        | 62     | I         | I      | -1     | I      | I         | I      | I      | I      | 9         | I      | I      | I      | I      | Ι      | Ι      | I      | I      | I      | I      | bes predi         | ach site i  |
|       | Med   | I         | I      | I         | I      | I      | I      | 36        | 43     | x      | 42     | I         | 19     | 39     | 2      | 9      | 7      | ъ      | 7      | 6      | ъ      | 48     | nal tvr           | L for e     |
| TrBE  | BB    | I         | I      | I         | I      | I      | I      | 9         | 4      | 16     | 2      | I         | 4      | 5
2 | 4      | ъ      | 4      | 4      | 1      | 4      | 4      | 19     | functio           | nt PF       |
|       | LPJ-G | I         | I      | 1         | I      | l      | I      | 58        | 45     | 18     | 52     | I         | 57     | 52     | 2      | 2      | 5      | 2      | 2      | 33     | n      | 60     | he plant :        | e domina    |
|       | Med   | I         | I      | I         | I      | I      | I      | 11        | 9      | I      | 13     | I         | 16     | ×      | 18     | 22     | 20     | 22     | 20     | 19     | 20     | 4      | d. of t           | or. Th      |
| Tribe | BB    | I         | I      | I         | I      | I      | I      | 19        | 13     | 4      | 14     | I         | 16     | 16     | 21     | 27     | 21     | 23     | 14     | 18     | 15     | 14     | eric              | ake fact    |
|       | LPJ-G | ļ         | I      | 1         | I      | I      | I      | 2         | 2      | I      | 5
L | I         | ы      | 1      | 27     | 22     | 28     | 22     | 19     | 20     | 28     | 2      | e simulate        | vater upt   |
|       | Med   | I         | I      | 1         | I      | I      | I      | 26        | 12     | I      | 25     | Ι         | 47     | 30     | 65     | 63     | 62     | 61     | 64     | 60     | 63     | 11     | e whol            | type v      |
| TrBR  | BB    | I         | Ι      | 1         | I      | I      | I      | 48 | 45     | 6      | 56     | Ι         | 53     | 53     | 60     | 62     | 61     | 63     | 68     | 64     | 68     | 37     | over th           | e CLM       |
|       | LPJ-G | I         | I      | I         | I      | I      | I      | ъ         | 4      | 1      | ы      | Ι         | 9      | 9      | 60     | 68     | 61     | 69     | 73     | 74     | 67     | 3      | veraged o         | ns use th   |
|       | Med   | I         | I      | 1         | I      | I      | I      | I         | I      | I      | I      | 39        | I      | I      | I      | I      | I      | I      | I      | I      | 1      | I      | cover.            | nulatic     |
| TeBE  | BB    | I         | Ι      | I         | Ι      | I      | I      | I         | I      | I      | I      | 48 | I      | I      | I      | I      | I      | I      | Ι      | I      | I      | I      | ective (          | SM sir      |
|       | LPJ-G | 1         | I      | 1         | I      | I      | Ι      | I         | I      | I      | I      | 51        | I      | I      | 1      | I      | Ι      | Ι      | I      | I      | I      |        | oliar proie       | re. The L   |
| Site  |       | AU-Emr    | ES-Amo | AU-DaP    | AU-Stp | CG-Tch | PA-SPs | AU-DaS    | AU-Dry | SD-Dem | AU-Ade | AU-Gin    | AU-How | AU-RDF | AU-Rob | BR-Sa1 | BR-Sa3 | GF-Guy | GH-Ank | MY-PSO | PA-SPn | ZM-Mon | Table 1: Fo       | a percentas |

| nt functional types predicted for each site, given a | inant PFT for each site is highlighted in bold font |
|------------------------------------------------------|-----------------------------------------------------|
| l, of the plan                                       | or. The dom                                         |
| over the whole simulated period                      | ne CLM type water uptake facto                      |
| r projective cover, averaged                         | The LSM simulations use the                         |
| Table 1: Foliar                                      | a percentage.                                       |

---

## Author Comment (AC2)

We thank the anonymous reviewer for their thorough feedback and useful suggestions, which helped improve our manuscript. Our replies to the reviewer's comments are written in blue italics. Proposed changes to the manuscript are highlighted in red.

**Major comments.**

The benefits of implementing a new radiative transfer scheme and soil physics are not sufficiently analyzed or discussed. The implementation of sunlit and shaded leaves should have some impact on the radiative budgets and the simulation of PAR. It would be good to show a comparison of PAR between old and new versions of LPJ-GUESS to just showcase the differences.

The main motivation for the developments described in the paper is to be able to use LPJ-GUESS to sutdy feedbacks between the land vegetation ecosystems and the climate. In order to capture feedbacks between the climate and the surface radiation budget, upwelling short wave radiation (or, equivalently, surface albedo) must be calculated dynamically by the model. This quantity depends on surface vegetation cover, soil colour and water content and, importantly in the case of subdaily calculations, the solar zenith angle. Additionally, the optical properties of the canopy and the soil also have a spectral dependency, so it is necessary account for visible and near-infrared radiation separately. The radiative scheme in LPJ-GUESS deals only with the visible part of the spectrum (PAR), and assumes an average global, daily albedo of 17%. Therefore, it is unfit to calculate upwelling shortwave radiation on subdaily time steps. The sunlit/shaded leaves approach was chosen because it has been shown to be a reasonable compromise between accuracy and computational efficiency (Leuning et al., 1998).

To highlight the main motivation to change the radiative transfer scheme early in the manuscript, we suggest modifying lines 61-63 of the manuscript (at the end of the introduction) as follows:

"To achieve [calculating diurnal energy fluxes to make coupling with an atmospheric model possible], we introduced several major modifications to LPJ-GUESS v4.0, namely: (a) a new radiative transfer scheme, capable of representing direct and diffuse light, as well as treating sunlit and shaded leaves separately; capable of calculating upwelling short wave radiation dynamically on a sub-daily time step, as well as accounting for direct and diffuse solar radiation separately; [...]"

*Lines* 129-135 of the manuscript explain in more detail the advantages of introducing a sunlit/shaded partition, as well as the necessity to account for near-infrared radiation.

"The vertical layering of the canopy is kept in the radiation calculations, but the new scheme distinguishes direct and diffuse radiation and two separate wavebands (visible and near infrared). Infrared radiation does not contribute to photosynthetic assimilation, but needs to be accounted for in the energy balance calculations. A separate treatment of diffuse and direct radiation allows to resolve sunlit and shaded leaves. This approach has been shown to lead to predictions of fluxes of energy, water and CO2 that are comparable in accuracy to those made by more complex, and considerably more computationally expensive, multi-layered canopy models (Wang and Leuning, 1998)."

To further stress the above points, we propose to modify lines 581-585 in the summary section as follows:

"The newly incorporated energy balance module resolves the diurnal cycle of energy and water fluxes between the canopy and the atmosphere, as opposed to LPJ-GUESS's daily calculations. This enablesthe shorter time step used by atmospheric models to be matched. The simple, Beer's law-based PAR absorption calculations were replaced with a more sophisticated two-stream radiative transfer scheme (Sellers, 1985; Dai et al., 2004), which allows for separate treatment of sunlit and shaded leaves in the canopy Calculating these fluxes on a sub-daily basis is necessary to match the shorter time steps at which atmospheric models operate (typically one hour or shorter, depending on resolution). The original daily PAR absorption calculations were replaced with a more sophisticated radiative transfer scheme by adapting the models of Sellers (1985) and Dai et al. (2004) to LPJ-GUESS's multi-cohort, multi-layer canopy (some differences in PAR absorption calculated by both schemes are shown in the supplement). This enables the model to simulate the upwelling shortwave radiation flux on sub-daily time scales. Direct and diffuse radiation are treated separately, which allows to resolve sunlit and shaded leaves in the canopy. This approach offers a reasonable compromise between accuracy of the modeled fluxes and computational efficiency (Wang and Leuning, 1998)."

A comparison between PAR absorption calculated by the new radiative transfer scheme and standard LPJ-GUESS's algorithm is now shown in the supplement (see below).

The implementation of more soil layers and sub-daily calculation of soil temperature and moisture may have an impact on soil C and N cycle and thus influence NEE or Rh, which might be better discussed in the light of Table 8.

To address this point, we propose adding the following text to the summary section (line 589):

"These formulations are better fit to resolve near-surface heat and water fluxes on the sub-daily time scales introduced in the model. They lead, however, to discrepancies between the LSM and standard LPJ-GUESS, which stem from the different physical environments simulated in the models. Calculating assimilation at the newly simulated canopy temperature, rather than the air temperature, can lead to either higher or lower productivity, depending on the optimal photosynthetic temperature ranges of each PFT and the effect of temperature on N limitation (see Sec. 4.2). Canopy temperature also affects autothrophic respiration, while differences in the simulated soil humidity and temperature impact organic matter decomposition rates and heterotrophic respiration. The combination of these effects results in differences in the simulated equilibrium carbon and nitrogen pools and ecosystem-atmosphere carbon fluxes. Plots of the carbon and nitrogen pools and the ecosystem fluxes for selected sites are included in the supplement."

Additionally, selected spinup plots are now included in a supplement.

\*\*\*

The choice of leaving Vcmax outside of the sub-daily loop, needs to be explained and discussed. Choosing to update Vcmax on daily scale (not subdaily or even longer time scale) should be justified.

How does nitrogen limitation affect An? Does nitrogen limitation operate on a daily or sub-daily scale? These calculations might be similar to the old model version of LPJ-GUESS, but still it is useful to explain here.

Also in Section 2.2.4, it is not clearly explained how the net photosynthetic rate (An) is derived in the model.

Since there is a mixed use of daily and sub-daily variables in Section 2.2.4, it would be clearer if all the variables in the equations are clearly denoted whether they are daily or hourly variables to avoid mis-understanding.

In both the original and the augmented model, Vmax is calculated daily, and is limited by the available nitrogen. The impact of N limitation on An is therefore through reduced Vmax values, and is updated on a daily time step. The time scale of readjustment of leaf nitrogen and its impact on photosynthetic rates is days to weeks (e.g. Reich et al., 1991; Irvin and Robinson, 2006), which is too slow to follow diurnal environmental variations. We therefore did not consider moving the Vmax calculation into the subdaily loop. Longer time scales were not considered because the original model updates this quantity daily. Assessing the impact of updating this quantity on longer time scales in both the original and the LSM versions of LPJ-GUESS was out of the scope of this work. In order to clarify these points we propose the following changes to the text:

- The description of the stomatal conductance models will be moved to the end of the section.
- We will reference the original works upon which LPJ-GUESS's photosynthetic scheme is based.

**Lines 262 and following:**

"In what follows, variables that are updated daily are denoted with the subscript 'day'. Daytime averages are denoted with the subscript 'dt'. All the other variables are computed on a subdaily basis. Photosynthetic assimilation is now calculated within the subdaily energy balance routine (Fig. 1). A net photosynthesis rate is computed for the sunlit and shaded leaves of each cohort separately: by calling the photosynthesis routine built in LPJ-GUESS. This calculation is based on the biochemical model of Collatz et al. (1991, 1992), the strong-optimality model of light use efficiency of Haxeltine and Prentice (1996), and the nitrogen limitation of the maximum carboxylation rate,  $V_{max, day}$ , introduced in Smith et al. (2014). For a given cohort i,  $V^{(i)}_{max, day}$  is recalculated at the end of each simulation day and depends linearly on the total amount of daily absorbed photosynthetic active radiation, PAR(i)day (Haxeltine and Prentice, 1996):

**(...)**

In this equation,  $V^{(i)}_{max, day}$ , is expressed per unit patch area. This potential rate is calculated by LPJ-GUESS for every cohort daily (Fig 1). The slope of the relationship,  $f_v$ , depends on environmentalfactors, including temperature and leaf nitrogen content. encodes the influence of temperature and nitrogen limitation. Updating  $V^{(i)}_{max, day}$  on sub-daily time scales is not necessary because readjustment of leaf nitrogen content and photosynthetic traits occurs on time scales of days to weeks (e.g. Reich et al., 1991; Irvin and Robinson, 2006), and therefore cannot follow diurnal environmental variations."

We will also add the subscript 'day' to  $V^{(i)}_{max,sun,day,leaf}$

Are stomatal conductance and An co-determining each other? How CO2 impacts An?

To clarify this, we suggest adding the following text after the stomatal conductance descriptions (currently line 275):

"The photosynthesis rate depends on the CO2 concentration inside the stomatal cavity. This concentration is related to the atmospheric CO2 concentration through a diffusion process across the stomatal opening and the leaf boundary layer, and therefore depends upon stomatal conductance,

which in turn depends on the photosynthetic rate (Eqs. 41 and 42). Hence, photosynthetic rates and stomatal conductance are calculated simultaneously by iteration. A detailed description of the algorithm can be found in Bonan (2020)."

\*\*\*

The comparison between the new and old versions of LPJ-GUESS is not very clear to me, because the differences between the two experiments can be due to either the newly implemented codes in this manuscript (direct effect) or the differences in the PFT cover fractions of the two experiments as shown in Table 6 (an indirect effect of the newly implemented codes). It would be interesting to tease out the direct and indirect effects of the newly implemented codes on heat fluxes. I am wondering if it is possible to do a set of sensitivity experiments using the new version of LPJ-GUESS but with prescribed PFT cover fractions from the old version of LPJ-GUESS?

We agree with the reviewer that the difference in model predictions between the standard and the LSM versions of LPJ-GUESS owes to both direct and indirect effects. However, we think these are not easy to disentangle. In LPJ-GUESS, relative PFT coverage results from both productivity (since it depends on LAI) and competition, and cannot be prescribed within the area where competition occurs (the patch). An experiment similar to the one suggested by the reviewer could be set up by dividing the gridcell in LSM simulations into several tiles, whose relative size would be determined by the fractional FPC predicted by the standard LPJ-GUESS simulation. However, this would not help to tell direct effects from indirect effects:

- Since PFTs do not compete with each other in the tile setup, such experiment would also lead to different predictions in the standard version of LPJ-GUESS.

- FPC coverage within each tile would depend on the productivity of the PFT growing in a competition-free environment, so the resulting total gridcell coverage would still turn out different than the one predicted by the standard LPJ-GUESS simulation. This can be seen in the simulations where C4 grasses grow without competition. In three out of four sites, productivity per unit leaf area is higher in the LSM version, which is a direct effect of using leaf rather than air temperature to calculate Vmax. This results on a higher plant coverage (e.g. it grows from 82 to 94-95% in Au-DaP, Table 6), which will result in even higher productivity per unit patch area (an indirect effect).

The full impact of the new schemes on the model output will be better evaluated in combination with regional and global experiments (these are the spatial scales that LPJ-GUESS is designed to run at; a site-based study was chosen for this paper because of the focus on the evaluation of the energy fluxes against flux tower measurements).

**Other comments:**

Line 56-57: Please shortly explain the deficiencies of the existing "coupled biosphere-atmosphere regional and global studies" using LPJ-GUESS. This will help the reader to better understand the importance of this work.

Since some of the deficiencies arising from an indirect coupling approach have been already listed above these lines, we propose modifying the text as follows to stress the point:

**Lines 49 and following:**

"DGVMs are frequently coupled to integrated into ESMs through an intermediary Land Surface Model (LSM), which facilitates the sub-daily energy, water and gas exchange calculations (e.g. Bonan et al., 2003; Krinner et al., 2005; Smith et al., 2011; Döscher et al., 2021). This is necessary because DGVMs run normally on a daily or longer time step, while atmospheric models may use time steps ranging from seconds to tens of minutes, depending on the required resolution. This indirect approach to coupling can, however, entail inconsistencies between the DGVM and the LSM, such as the use of different time steps and temperatures in photosynthetic calculations, duplicated or inconsistent soil water tracking, or different characterization of vegetation types. One possible important consequence of these inconsistencies is the failure to conserve carbon mass. In this work we modify the LPJ-GUESS DGVM (Smith et al., 2001, 2014) to enable coupling with an atmospheric model without the need for a mediating LSM, LPJ-GUESS simulates a wide range of land-biosphere processes, including vegetation growth, establishment and mortality, plant functional type (PFT) competition, disturbances, wildfires, and land use change. This model has been used in a broad range of applications, including coupled biosphere-atmosphere regional (Wramneby et al., 2010; Smith et al., 2011; Zhang et al., 2014, 2018; *Wu et al.*, 2016, 2021) and global (*Weiss et al.*, 2014; *Alessandri et al.*, 2017; *Forrest et al.*, 2020; Döscher et al., 2021) studies, although these suffer from the above-mentioned limitations of the indirect coupling approach. LPJ-GUESS is maintained by an international developer community and undergoes active development and evaluation, which makes it a suitable choice to study climatebiosphere interactions."

Line 230: Why set the same optical properties for all the PFTs? There should be data available to help parameterize these parameters for different PFTs.

We used the same optical properties for all PFTs in order to limit the degrees of freedom and keep the development tractable. In this way we can introduce them at a later stage and assess the impact of this change more easily (we note that in standard LPJ-GUESS PFTs do not have different optical properties).

Line 236-237: Have you considered the effect of soil moisture on soil optical properties?

*Yes. In order to clarify this point, we suggest replacing the sentence (line 236)*

"Soil optical properties are from the dataset prepared by Lawrence and Chase (2007)."

with

"Soil albedo is calculated from the soil dry and moisture-saturated reflectances and the water content of the top soil layer following Oleson et al. (2010). Soil color classes are from Lawrence and Chase (2007), and were obtained from the dataset included in the CLM4.0 code (Lawrence et al., 2011)."

Equation 47: "Vsun,day" should be "V max, sun, day"?

We thank the reviewer for spotting the typo in the equation. We will correct it in the revised manuscript.

Line 328: Does heat capacity also depend on organic matter content?

Soil heat capacity does not depend on organic matter content in this version of the model. We plan to introduce this dependency at a later stage in the development. In order to clarify this, we suggest adding the following text (line 327):

"Soil heat capacity is computed as a weighted sum of the heat capacities of the dry soil, which depends on texture, and water (de Vries, 1963). Soil organic matter does not contribute to soil heat capacity in the current version of the model."

Line 359-360: Please explain in more details why such overestimation happens?

The overestimation happens because the potential evaporation rates are calculated on the basis of the current canopy temperature in a given iteration. At the end of the iteration, with the updated canopy temperature, the energy fluxes are recalculated. We found that this recalculation before proceeding to the next iteration gives stability to the numerical scheme and reduces the number of iterations. Since fluxes are recalculated at the updated temperature, but evapotranspiration is partitioned between canopy evaporation and transpiration on the basis of the conductances derived from the potential evaporation rates before updating the temperatures, a mismatch can happen if the canopy does not hold enough water to meet the newly calculated evaporation demand. The larger the new potential evaporation rate, the larger this error will be. This error could be compensated by supplying an equivalent amount of water from the soil.

We suggest changing the text as follows:

"We found that the bulk of the water conservation error is due to a generally small overestimation of canopy evaporation when the potential evaporation at a given time step is substantially larger than the available canopy water arising from a recalculation of the energy fluxes at the end of every iteration. This recalculation leads to energy fluxes that are slightly inconsistent with the potential evaporation rates calculated at a different temperature at the beginning of the iteration, which are used to calculate the partitioning between canopy evaporation and transpiration after the iteration is completed. A possible solution would be to assign the excess canopy evaporation to transpiration, and subtract the corresponding amount of water from the soil."

Line 375-377: What time step of LPJ-GUESS is used in the simulation? Half hour or 1 hour or 3 hour? Why do hourly averaging for the forcing data instead of using the original half-hourly forcing data which might be more physically consistent for different forcing variables.

We used a time step of 1h in the LSM runs. Since some of the sites provide only hourly averages of the climatology, we decided to average half-hourly data in order to have a consistent input time step across sites. We believe, however, that this choice should not affect the results significantly.

We suggest adding the following text to clarify the time step used in the simulations (line 375):

"We used the climate data collected at the tower sites to force the model. Half-hourly forcing data was converted to hourly averages to use a fixed time step of 1 hour in all simulations."

Section 3.2: It is not clearly explained how soil properties were set for each site simulation.

In section 2.2.5 (Soil Physics) we state:

(*Line 327*): "Soil heat capacity is computed as a weighted sum of the heat capacities of the dry soil, which depends on **soil texture**, and water (de Vries, 1963)."

(Line 333): "Hydraulic diffusivity and conductivity are calculated as a function of **soil texture** and soil water content by using the expressions derived by Clapp and Hornberger (1978) and Cosby et al (1984)."

To clarify where the texture data come from, we suggest inserting the following text in line 378:

"Nitrogen deposition data is from Lamarque et al. (2013). Atmospheric CO2 concentration data is from McGuire et al. (2001). The soil texture data used to calculate soil hydraulic and thermal properties (as described in Sec. 2.2.5) at each site were as in Sitch et al. (2003), based on The Digitized Soil Map of the World (Zobler, 1986; FAO, 1991)."

Figure 7-13: please add (a, b, c, d, e ....) for each subplot.

The plots were corrected as suggested and will be included in the revised version of the manuscript.

Line 422: "dry season": please specify which months.

The text will be changed in the revised manuscript as follows:

"At the AU-DaS site (Fig. 7, upper right panel), the shapes of measured and simulated annual cycles match relatively well at the beginning and the end of the year, but diverge substantially during the dry season (July-November)."

Additionally, we suggest to include, for reference (to also address the next comment about the 'systematic overestimation of turbulent fluxes'), the average precipitation and the net radiation in the plot:

Line 434-435: Where do "the systematic overestimation of sensible and latent heat fluxes" (i.e., excess energy) come from in the model?

At this site, the model overestimates net (absorbed) radiation between March and May, but between June and November modeled absorbed radiation is very close to measurements. Since energy is balanced in the model, the excess energy in turbulent fluxes must be compensated by an underestimation of heat conducted into the ground. One possibility is that simulated upper soil moisture is lower than actual soil moisture at the site, which would lead to an underestimation of upper soil thermal conductivity in the model. Unfortunately, soil moisture data are not available for this site, so we could not to test this hypothesis. We suggest adding the following text to address this point in the paper:

Lines 434-435: "Sensible and latent heat fluxes are systematically overestimated by the model by ~  $10-20 \text{ Wm}^{-2}$ . This overestimation takes place even when simulated net radiation is very close to observations (June to November), so it must be compensated by an underestimation of ground heat, possibly caused by an underestimation of upper soil moisture. Unfortunately, soil moisture measurements are not available for this site, so we were not able to test this hypothesis."

---

## Author Comment (AC4)

*We thank the anonymous reviewer for their kind words, as well as their thorough feedback and useful suggestions, which helped improve our manuscript. Our replies to the* reviewer's comments *are written in blue italics. Proposed changes to the manuscript are highlighted in red.*

As for the water (and latent heat) fluxes issue, the authors invested a lot of efforts to look for reasons with different stomatal conductance models and water uptake functions. I think an expanded discussion about the interactions between vegetation settings from the LPJ-GUESS model and physical environments derived from the new biophysical processes would be helpful. This would also be helpful for understanding the differences in productivity between the original LPJ-GUESS and it coupled with LSM.

*LPJ-GUESS calculates many vegetation properties dynamically, including key photosynthetic quantities such as Vmax. The new scheme interacts with these processes, and changes in simulated fluxes and productivity arise as a result of the interplay of many factors. We found that the biggest discrepancies between the standard model and the LSM version are due to the latter using leaf temperature, rather than daily averaged air temperature, for photosynthetic calculations. Leaf temperature can be several degrees above air temperature, affecting the photosynthetic rate directly, via the temperature inhibition factor, but also (and most dramatically in the case of C4 grasses) indirectly, by mitigating nitrogen limitation of photosynthesis. This is seen in simulations where we restrict establishment to grassy PFTs and C4 grasses emerge as the dominant grass type. While this effect seems to be far less pronounced in a potential natural vegetation simulation, due to trees entering the competition, it may be important when simulating C4 crops (since these grow without competition in well-watered conditions). The effect of leaf temperature on the biochemical nitrogen limitation of photosynthesis, and how it affects C4 grass productivity in competition and no-competition situations, is discussed in detail in section 4.2.*

*We now discuss in the supplement how the new simulated physical environments affect soil carbon and nitrogen build-up and how the PAR absorption calculations compare for the old and the new schemes (see attached document). We also discuss how the different stomatal conductance and soil water uptake formulations lead to differences in simulated soil water content that can alter the PFT composition (new section 3.4.4, attached).*

*We also suggest adding the following text to sect. 3.4.2:*

"*The above-described discrepancies between standard LPJ-GUESS and the LSM versions stem from the different physical environments simulated in the models. Calculating assimilation at the newly simulated canopy temperature, rather than the air temperature, can lead to either higher or lower productivity, depending on the optimal photosynthetic temperature ranges of each PFT and the impact of temperature on nitrogen limitation (Sec. 4.2). Canopy temperature also affects autotrophic respiration, while differences in the simulated soil humidity and temperature impact organic matter decomposition rates and heterotrophic respiration. The combination of these effects results in differences in simulated carbon and nitrogen pools and NEE (we have included a comparison between soil carbon and nitrogen pools simulated by standard LPJ-GUESS and LPJ-GUESS/LSM in the supplement).*"

*We will now address the minor comments.*

1. Line 146: the unit of lambda. I think "C-1" is not necessary. The number of 2.44x10^6 has included an assumption of normal water temperature

*We thank the reviewer for spotting the mistake in the units. Lambda represents the amount of energy that it takes to evaporate a unit mass of liquid water, so the correct units should be of energy per unit mass (J/kg). We will correct the text accordingly.*

2. Lines 380 and 388: How do the vegetation conditions vary with the actual vegetation? I think the vegetation and soil states (equilibrium or not) may affect NEE, as shown in Fig. 14 that the simulations are close to equilibrium state.

4. Lines 572: measured NEE is more negative than those simulated. I think it is related to how far the vegetation is from its theoretical equilibrium state. Disturbances also play a role here. For example, at equilibrium state, an ecosystem will have a zero NEE (or fluctuated around zero) if the system has no disturbances. However, if it is equilibrated with a particular disturbance regime (e.g., a given fire frequency distribution), the system must have a negative NEE that is to counter the carbon release at disturbance events. In long-term, it is still carbon neutral. This is the pattern this paper showed in this section that observations have higher carbon sink (more negative NEE) than the simulated (with 500 years of model run).

*We agree with the reviewer that the discrepancy in NEE probably reflects the fact that the simulated carbon pools are close to equilibrium with the synthetic climate data used to spin up the model, and that this set up differs from the actual situation at the different sites (this is seen also in the difference between the variabilities of observed and simulated fluxes). In a standard LPJ-GUESS simulation, the spinup procedure is designed to bring carbon pools close to equilibrium with pre-industrial conditions, before simulating the historical period of rising temperatures and $CO_2$ concentrations. Here, our primary goal was to evaluate the sub-daily latent and sensible heat fluxes calculated by the augmented model against observations, and then compare the ecosystem-related predictions (structure, composition and fluxes) of the new LSM model with those made by the standard (non-LSM) version. Therefore, differences between simulated and measured NEE are to be expected because we did not attempt to fully reproduce or account for site history, including age, disturbance, and legacies arising from historical trends in $CO_2$ concentration. Evaluating simulated ecosystem productivity and carbon fluxes against observations, as well as the differences between the standard and the augmented model in regional and global scales, is the object of future work, but was out of the scope of this paper.*

*We suggest adding the following sentence in section 3.4.4:*

*"These discrepancies between observed and simulated NEE magnitude and variability reflect the fact that, in the simulations, the carbon pools are all close to equilibrium with the climate and atmospheric $CO_2$ concentration as a result of the spinup procedure described in section 3.2. Differences are to be expected because we did not attempt to fully reproduce or account for site history, including age, disturbance, and legacies arising from historical trends in $CO_2$ concentration."*

*Additionally, we added a discussion of the build-up of soil organic matter pools during the spinup phase in the supplement.*

3. Line 506 section Ecosystem structure and function: They are related to the settings of the vegetation model since the structure is highly dynamic. How to make them consistent with each case and the measurement data?

*For this work, we used the standard set of PFTs provided with LPJ-GUESS, which aims to capture the main functional traits of different vegetation classes in global simulations. Our aim in this paper was to evaluate how well the model (including the standard set of PFTs) simulated energy fluxes, rather than fitting the PFT parameters at each site specifically to achieve a best fit between model and observations. We chose this approach since the end goal is to perform regional and global runs. Also, see response to previous comment.*

5. Time steps of and growth (yearly) and SOM (daily): how LAI dynamics and heterotrophic respiration are calculated? Usually, LAI should be updated daily and Rh hourly (or half hourly). Are they connected with plant growth and SOM dynamics at each step, respectively? This just need to clarify. I may miss the description.

*- In this work we did not modify the original model's growth and phenology routines. At present, the allocation of carbon to the different plant structures (what we call growth) is simplified in the model, happening annually. The phenological status is calculated and updated daily according to the phenology of the different PFTs (Raingreen, Summergreen, Evergreen, as described in appendix B4 of Smith et al. (2014), https://doi.org/10.5194/bg-11-2027-2014). This is stated at the beginning of section 2.2 of the model description and in Figure 1.*

*- Rh is updated daily in the current version of the model. Even though Rh can vary on diurnal timescales, we focused on averages of yearly NEE over the measuring period, so we assumed calculating Rh on a daily basis was sufficient for our purposes. We stress, however, that photosynthesis and autotrophic respiration are calculated on a subdaily basis, consistent with the canopy energy balance, and accumulated to calculate daily NPP, and then subtracted from daily RH to calculate daily NEE at the end of the day. To clarify this, we added the following text at the beginning of section 2.2.4:*

*"The net photosynthetic assimilation is accumulated over the diurnal cycle and subtracted from heterotrophic respiration (Rh, computed daily) to calculate daily net ecosystem exchange (NEE)."*

6. In discussion, for the water uptake functions and C4 grass carbon assimilation simulations, they are phenomenological equations in the model that directly link soil water availability to leaf functions. A discussion of actual plant-soil hydraulics would be helpful for understanding why they happen and why we don't have to spend much time to tune these functions.

*We agree with the reviewer that these expressions are oversimplifications of the very complex process of stomatal regulation. To stress this fact in a concise way, we suggest adding the following text when introducing the soil moisture stress factor (end of section 2.2.4):*

*"This type of formulations, which are widely use in LSMs (see Damour et al, 2010, for an overview), are phenomenological relationships that attempt to capture the response of plants to water stress in a rather simplified way (Egea et al., 2011; De Kauwe et al., 2013). Transpiration of soil water by plants is primarily driven by the water potential gradient along the soil-plant-atmosphere continuum. Plants regulate this gradient by opening and closing their stomata in response to environmental factors, including leaf water potential, vapor pressure deficit, and soil water availability, in a way that depends on their hydraulic strategy (a detailed discussion can be found in Lambers et al, 2008). Including a more explicit representation of soil-plant-air hydraulics as well as physiological constraints in a stomatal conductance model has been shown to perform better than the above formulations under soil water stress conditions (Bonan et al., 2014). However, implementing these more complex models in*

*ESMs remains a challenge due to a lack of data for broader applicability and computational efficiency tradeoffs (Clark et al., 2015)."*

NEW SECTION 3.4.1

[revised manuscript text omitted]
 PAR absorbed by the vegetation, calculated using the new radiative transfer scheme and the PAR absorption scheme in standard LPJ-GUESS. Data are from the CLM/Med simulations described in the paper. PAR values are averages over the measurement period of the simulations, in MJ/year/m$^2$. The percent change is relative to the standard LPJ-GUESS run.

**1 Differences in PAR absorption between LPJ-GUESS and LPJ-GUESS/LSM**

Table 1 shows a comparison of average PAR absorption per unit LAI calculated by the new radiative transfer scheme and the PAR absorption algorithm in stadard LPJ-GUESS. The calculations were made in the CLM/Med simulation, i.e., PAR absorption is calculated with both schemes in the same modeled areas for the purpose of this comparison. In general, the new radiative transfer calculates higher absorbed PAR values than standard LPJ-GUESS at sites with low modeled LAI values, while both calculations yield similar results at sites with high LAI values. This behaviour can be understood by examining PAR absorption by individual cohorts. Figure 1 shows PAR absorption by the vegetation over 60 years during the spinup period at BR-Sa1, starting after a disturbance. Three tree cohorts (0, 1 and 2) and a grass individual (4) establish. Initially, grass has a high LAI, but, as trees grow and the canopy thickens, the grass LAI declines (panels $c$ and $d$). Calculated tree PAR absorption per leaf area is initially similar for both schemes (panel $a$), but as cohort 0 grows it shadows cohorts 1 and 2. The new radiative transfer scheme calculates lower PAR values for these two cohorts, but since their leaf area index is also declining, this does not contribute substantially to the overall difference, which is small and dominated by cohort 0 (panel $b$).

Figure 2 shows the same comparison for a patch at AU-Gin. In this case,

[Figure]

Figure 1: Comparison of PAR absorbed by the cohorts in a patch at BR-Sa1, calculated using the new radiative transfer scheme and the standard LPJ-GUESS PAR absorption scheme. *(a):* Annual absorbed PAR per leaf area; *(b):* Percent change in PAR absorption relative to standard LPJ-GUESS; *(c):* LAI; *(d):* Cohort height.

the tree cohorts have a lower leaf area index, so their leaves receive, on average, more direct sunlight than in the case of a thicker canopy. The new radiative scheme calculates higher values of absorbed PAR for these cohorts (panels *a* and *b*), and this feature dominates the overall difference between the two schemes in this site.

**2   Spinup information**

In a standard LPJ-GUESS simulation the 500-year spinup process proceeds as follows: the first 100 years, the model runs without nitrogen uptake to allow build up of soil nitrogen pools. All vegetation in the patch is then reset, and plant nitrogen uptake is turned on. Between years 140 and 220, information on the rates of change of C and N pools is collected. This information is then used

[Figure]

Figure 2: Comparison of PAR absorbed by the cohorts in a patch at BR-Sa1, calculated using the new radiative transfer scheme and the standard LPJ-GUESS PAR absorption scheme. *(a):* Annual absorbed PAR per leaf area; *(b):* Percent change in PAR absorption relative to standard LPJ-GUESS; *(c):* LAI; *(d):* Cohort height.

to calculate carbon and nitrogen steady-state pool sizes analitically, assuming an equilibration time of 40000 years for the soil organic matter pools. The model then runs for another 280 years, a period considered long enough for the vegetation C and N pools to reach steady state.

In general, the steady-state size of the carbon and nitrogen pools is determined by the balance between the rate of carbon input to the system (NPP) and the turnover rates of the soil organic matter pools. The LSM implementation changes the physical environment at which these processes take place in the model. Calculating photosynthesis rates at the newly simulated leaf temperature can lead to higher or lower carbon assimilation, depending on the PFT's optimal photosynthetic temperature range. It can also boost productivity by mitigating the effect of N limitation (see paper, Section 4.2). Soil organic matter decomposition is affected by soil temperature and humidity; higher (lower) tem-

| | | | BB | | | | | Med | | |
|---|---|---|---|---|---|---|---|---|---|---|
| | NPP | Temp | AWC | Soil C | Soil N | NPP | Temp | AWC | Soil C | Soil N |
| AU-Emr | -66.7 | 13.1 | -30.0 | -74.1 | -74.1 | -68.7 | 14.4 | -49.6 | -75.3 | -75.3 |
| ES-Amo | 6.1 | 3.3 | -11.6 | -12.9 | -12.9 | 0.6 | 3.7 | -29.9 | -12.9 | -12.7 |
| AU-DaP | 1.8 | 4.5 | 46.2 | -12.8 | -10.2 | 25.5 | 4.4 | 19.8 | 21.7 | 25.7 |
| AU-Stp | -42.8 | 8.5 | -24.1 | -50.5 | -50.4 | -23.0 | 9.0 | -37.0 | -33.6 | -33.4 |
| CG-Tch | 87.6 | 2.5 | 51.5 | 6.8 | 9.2 | 92.0 | 1.8 | 52.1 | 10.4 | 12.9 |
| PA-SPs | 34.7 | 1.2 | 6.7 | 22.2 | 24.3 | 38.9 | 1.2 | 3.5 | 27.2 | 29.4 |
| AU-DaS | -11.8 | 0.9 | 42.1 | -0.9 | 3.5 | -6.5 | 1.7 | 16.5 | 15.9 | 21.3 |
| AU-Dry | -7.1 | 3.8 | 47.1 | -2.0 | 1.0 | 1.4 | 3.3 | 11.8 | 4.5 | 6.5 |
| SD-Dem | -13.1 | -0.4 | 85.0 | -45.9 | -47.2 | 34.8 | -0.3 | -0.0 | 43.6 | 46.6 |
| AU-Ade | -13.4 | 0.8 | 34.9 | 8.1 | 15.6 | -8.7 | 0.8 | 16.6 | 21.4 | 29.5 |
| AU-Gin | 0.6 | 6.2 | 56.3 | -32.7 | -32.3 | -9.1 | 6.0 | 25.7 | -25.3 | -23.6 |
| AU-How | -13.8 | -0.5 | 37.9 | 2.9 | 10.2 | -10.6 | -0.0 | 20.8 | 18.6 | 27.8 |
| AU-RDF | 4.6 | 3.3 | 45.4 | 15.0 | 19.9 | 12.0 | 3.6 | 21.0 | 28.7 | 33.1 |
| AU-Rob | 4.4 | 1.1 | 18.3 | -6.4 | -6.2 | 3.3 | 1.1 | 13.0 | -2.1 | -1.4 |
| BR-Sa1 | -25.2 | -0.5 | 8.1 | -16.0 | -15.3 | -20.5 | -0.6 | 5.7 | -14.2 | -13.3 |
| BR-Sa3 | -11.1 | -2.4 | 5.6 | -8.7 | -8.4 | -6.2 | -2.5 | -3.2 | -6.5 | -6.3 |
| GF-Guy | -14.6 | 0.3 | 9.9 | -14.7 | -14.5 | -11.3 | 0.6 | 6.0 | -12.6 | -12.3 |
| GH-Ank | -23.9 | -0.3 | 13.3 | -15.6 | -13.4 | -18.8 | -0.7 | 11.4 | -13.8 | -11.5 |
| MY-PSO | -30.1 | 0.3 | 59.2 | -43.1 | -42.6 | -23.2 | 0.5 | 54.4 | -37.5 | -37.4 |
| PA-SPn | -15.8 | 0.6 | 8.5 | -20.6 | -18.5 | -12.1 | 0.7 | 5.5 | -16.0 | -13.6 |
| ZM-Mon | -1.6 | 3.9 | 68.9 | -18.1 | -12.9 | -3.8 | 1.8 | 39.5 | -20.1 | -15.3 |

Table 2: Percent change in steady-state NPP, average soil temperature over the top 50 cm of soil, average water content over the top 50 cm of soil, soil carbon content, and soil nitrogen content, relative to standard LPJ-GUESS. Steady state values are taken as the average of the last 100 years of spinup.

peratures and humidities lead to higher (lower) turnover rates. Table 2 shows a comparison of these factors in LSM and standard LPJ-GUESS simulations for all the sites considered in this study.

We show two examples of the build-up of the soil organic matter pools at BR-Sa1 (Fig. 3) and SD-Dem (Fig. 4), for the standard LPJ-GUESS, the CLM/BB, and the CLM/Med runs. At BR-Sa1 in the BB simulation, equilibrium NPP is lower than in standard LPJ-GUESS by $\sim 25\%$ (Table 2). Soil temperature is similar to standard LPJ-GUESS, but soil moisture is $\sim 8\%$ larger. This leads to lower equilibrium soil carbon ($\sim -16\%$) and nitrogen ($\sim -15\%$) content. The CLM/Med simulation behaves similarly at this site (and at most forest sites).

At SD-Dem the BB and Med simulations show very different behaviours. In the BB simulation, NPP is lower than in LPJ-GUESS, while the higher stomatal resistance given by the Ball-Berry scheme (see paper, Fig. 3) causes higher soil moisture content. This leads to lower equilibrium soil organic matter content values (a $\sim 46\%$ decrease compared to standard LPJ-GUESS). In the Med simulation, equilibrium NPP is substantially higher than in the standard LPJ-GUESS run, while lower soil moisture retention leads to slower decomposition rates, resulting in soil organic matter pools $\sim 44\%$ larger than in standard LPJ-GUESS.

[Figure]

Figure 3: Comparison of the build up of carbon and nitrogen pools in the CLM/BB *(a) and (b)* and the CLM/Med *(c) and (d)* simulations with standard LPJ-GUESS, at BR-Sa1.

[Figure]

Figure 4: Comparison of the build up of carbon and nitrogen pools in the CLM/BB *(a) and (b)* and the CLM/Med *(c) and (d)* simulations with standard LPJ-GUESS, at SD-Dem.

---

## Author Comment (AC5)

*We thank the anonymous reviewer for their thorough feedback and useful suggestions, which helped improve our manuscript. Our replies to the* reviewer's comments *are written in blue italics.* Proposed changes to the manuscript are highlighted in red.

**Major comments.**

The benefits of implementing a new radiative transfer scheme and soil physics are not sufficiently analyzed or discussed. The implementation of sunlit and shaded leaves should have some impact on the radiative budgets and the simulation of PAR. It would be good to show a comparison of PAR between old and new versions of LPJ-GUESS to just showcase the differences.

*The main motivation for the developments described in the paper is to be able to use LPJ-GUESS to sutdy feedbacks between the land vegetation ecosystems and the climate. In order to capture feedbacks between the climate and the surface radiation budget, upwelling short wave radiation (or, equivalently, surface albedo) must be calculated dynamically by the model. This quantity depends on surface vegetation cover, soil colour and water content and, importantly in the case of subdaily calculations, the solar zenith angle. Additionally, the optical properties of the canopy and the soil also have a spectral dependency, so it is necessary account for visible and near-infrared radiation separately. The radiative scheme in LPJ-GUESS deals only with the visible part of the spectrum (PAR), and assumes an average global, daily albedo of 17%. Therefore, it is unfit to calculate upwelling shortwave radiation on subdaily time steps. The sunlit/shaded leaves approach was chosen because it has been shown to be a reasonable compromise between accuracy and computational efficiency (Leuning et al., 1998).*

*To highlight the main motivation to change the radiative transfer scheme early in the manuscript, we suggest modifying lines 61-63 of the manuscript (at the end of the introduction) as follows:*

"To achieve [calculating diurnal energy fluxes to make coupling with an atmospheric model possible], we introduced several major modifications to LPJ-GUESS v4.0, namely: (a) a new radiative transfer scheme,  capable of calculating upwelling short wave radiation dynamically on a sub-daily time step, as well as accounting for direct and diffuse solar radiation separately; [...]"

*Lines 129-135 of the manuscript explain in more detail the advantages of introducing a sunlit/shaded partition, as well as the necessity to account for near-infrared radiation.*

*"The vertical layering of the canopy is kept in the radiation calculations, but the new scheme distinguishes direct and diffuse radiation and two separate wavebands (visible and near infrared). Infrared radiation does not contribute to photosynthetic assimilation, but needs to be accounted for in the energy balance calculations. A separate treatment of diffuse and direct radiation allows to resolve sunlit and shaded leaves. This approach has been shown to lead to predictions of fluxes of energy, water and $CO_2$ that are comparable in accuracy to those made by more complex, and considerably more computationally expensive, multi-layered canopy models (Wang and Leuning, 1998)."*

*To further stress the above points, we propose to modify lines 581-585 in the summary section as follows:*

*"The newly incorporated energy balance module resolves the diurnal cycle of energy and water fluxes between the canopy and the atmosphere, as opposed to LPJ-GUESS's daily calculations.*

*the shorter time step used by atmospheric models to be matched. The simple, Beer's law-based PAR absorption calculations were replaced with a more sophisticated two-stream radiative transfer scheme (Sellers, 1985; Dai et al., 2004), which allows for separate treatment of sunlit and shaded leaves in the canopy* Calculating these fluxes on a sub-daily basis is necessary to match the shorter time steps at which atmospheric models operate (typically one hour or shorter, depending on resolution). The original daily PAR absorption calculations were replaced with a more sophisticated radiative transfer scheme by adapting the models of Sellers (1985) and Dai et al. (2004) to LPJ-GUESS's multi-cohort, multi-layer canopy (some differences in PAR absorption calculated by both schemes are shown in the supplement). This enables the model to simulate the upwelling shortwave radiation flux on sub-daily time scales. Direct and diffuse radiation are treated separately, which allows to resolve sunlit and shaded leaves in the canopy. This approach offers a reasonable compromise between accuracy of the modeled fluxes and computational efficiency (Wang and Leuning, 1998)."*

*A comparison between PAR absorption calculated by the new radiative transfer scheme and standard LPJ-GUESS's algorithm is now shown in the supplement (see below).*

The implementation of more soil layers and sub-daily calculation of soil temperature and moisture may have an impact on soil C and N cycle and thus influence NEE or Rh, which might be better discussed in the light of Table 8.

*To address this point, we propose adding the following text to the summary section (line 589):*

*"The new physical schemes introduced in this work lead to discrepancies between the LSM and standard LPJ-GUESS, which stem from the different physical environments simulated in the models. Calculating assimilation at the newly simulated canopy temperature, rather than the air temperature, can lead to either higher or lower productivity, depending on the optimal photosynthetic temperature ranges of each PFT and the effect of temperature on N limitation (see Sec. 4.2). Canopy temperature also affects autothrophic respiration, while differences in the simulated soil humidity and temperature impact organic matter decomposition rates and heterotrophic respiration. The combination of these effects results in differences in the simulated equilibrium carbon and nitrogen pools (see supplement) and ecosystem-atmosphere carbon fluxes."*

*Additionally, selected spinup plots are now included in a supplement.*
* * *
The choice of leaving Vcmax outside of the sub-daily loop, needs to be explained and discussed. Choosing to update Vcmax on daily scale (not subdaily or even longer time scale) should be justified.

How does nitrogen limitation affect An? Does nitrogen limitation operate on a daily or sub-daily scale? These calculations might be similar to the old model version of LPJ-GUESS, but still it is useful to explain here.

Also in Section 2.2.4, it is not clearly explained how the net photosynthetic rate (An) is derived in the model.

Since there is a mixed use of daily and sub-daily variables in Section 2.2.4, it would be clearer if all the variables in the equations are clearly denoted whether they are daily or hourly variables to avoid mis-understanding.

*In both the original and the augmented model, Vmax is calculated daily, and is limited by the available nitrogen. The impact of N limitation on An is therefore through reduced Vmax values, and is updated on a daily time step. The time scale of readjustment of leaf nitrogen and its impact on photosynthetic rates is days to weeks (e.g. Reich et al., 1991; Irvin and Robinson, 2006), which is too slow to follow diurnal environmental variations. We therefore did not consider moving the Vmax calculation into the subdaily loop. Longer time scales were not considered because the original model updates this quantity daily. Assessing the impact of updating this quantity on longer time scales in both the original and the LSM versions of LPJ-GUESS was out of the scope of this work. In order to clarify these points we propose the following changes to the text:*

*- The description of the stomatal conductance models will be moved to the end of the section.*
*- We will reference the original works upon which LPJ-GUESS's photosynthetic scheme is based.*

*Lines 262 and following:*

*"In what follows, variables that are updated daily are denoted with the subscript 'day'. Daytime averages are denoted with the subscript 'dt'. All the other variables are computed on a subdaily basis. Photosynthetic assimilation is now calculated within the subdaily energy balance routine (Fig. 1). A net photosynthesis rate is computed for the sunlit and shaded leaves of each cohort separately. by calling the photosynthesis routine built in LPJ-GUESS. This calculation is based on the biochemical model of Collatz et al. (1991, 1992), the strong-optimality model of light use efficiency of Haxeltine and Prentice (1996), and the nitrogen limitation of the maximum carboxylation rate, $V_{max, day}$, introduced in Smith et al. (2014). For a given cohort i, $V^{(i)}_{max, day}$ is recalculated at the end of each simulation day and depends linearly on the total amount of daily absorbed photosynthetic active radiation, $PAR^{(i)}_{day}$ (Haxeltine and Prentice, 1996):*

*(…)*

*In this equation, $V^{(i)}_{max, day}$, is expressed per unit patch area. This potential rate is calculated by LPJ-GUESS for every cohort daily (Fig 1). The slope of the relationship, $f_v$, depends on environmental factors, including temperature and leaf nitrogen content. encodes the influence of temperature and nitrogen limitation. Updating $V^{(i)}_{max, day}$ on sub-daily time scales is not necessary because readjustment of leaf nitrogen content and photosynthetic traits occurs on time scales of days to weeks (e.g. Reich et al., 1991; Irvin and Robinson, 2006), and therefore cannot follow diurnal environmental variations."*

*We will also add the subscript 'day' to $V^{(i)}_{max,sun,day,leaf}$*

Are stomatal conductance and An co-determining each other? How CO2 impacts An?

*To clarify this, we suggest adding the following text after the stomatal conductance descriptions (currently line 275):*

*"The photosynthesis rate depends on the $CO_2$ concentration inside the stomatal cavity. This concentration is related to the atmospheric $CO_2$ concentration through a diffusion process across the stomatal opening and the leaf boundary layer, and therefore depends upon stomatal conductance, which in turn depends on the photosynthetic rate (Eqs. 41 and 42). Hence, photosynthetic rates and stomatal conductance are calculated simultaneously by iteration. A detailed description of the algorithm can be found in Bonan (2019)."*

\*\*\*

The comparison between the new and old versions of LPJ-GUESS is not very clear to me, because the differences between the two experiments can be due to either the newly implemented codes in this manuscript (direct effect) or the differences in the PFT cover fractions of the two experiments as shown in Table 6 (an indirect effect of the newly implemented codes). It would be interesting to tease out the direct and indirect effects of the newly implemented codes on heat fluxes. I am wondering if it is possible to do a set of sensitivity experiments using the new version of LPJ-GUESS but with prescribed PFT cover fractions from the old version of LPJ-GUESS?

*We agree with the reviewer that the difference in model predictions between the standard and the LSM versions of LPJ-GUESS owes to both direct and indirect effects. However, we think these are not easy to disentangle. In LPJ-GUESS, relative PFT coverage results from both productivity (since it depends on LAI) and competition, and cannot be prescribed within the area where competition occurs (the patch). An experiment similar to the one suggested by the reviewer could be set up by dividing the gridcell in LSM simulations into several tiles, whose relative size would be determined by the fractional FPC predicted by the standard LPJ-GUESS simulation. However, this would not help to tell direct effects from indirect effects:*

*- Since PFTs do not compete with each other in the tile setup, such experiment would also lead to different predictions in the standard version of LPJ-GUESS.*

*- FPC coverage within each tile would depend on the productivity of the PFT growing in a competition-free environment, so the resulting total gridcell coverage would still turn out different than the one predicted by the standard LPJ-GUESS simulation. This can be seen in the simulations where C4 grasses grow without competition. In three out of four sites, productivity per unit leaf area is higher in the LSM version, which is a direct effect of using leaf rather than air temperature to calculate Vmax. This results on a higher plant coverage (e.g. it grows from 82 to 94-95% in Au-DaP, Table 6), which will result in even higher productivity per unit patch area (an indirect effect).*

*The full impact of the new schemes on the model output will be better evaluated in combination with regional and global experiments (these are the spatial scales that LPJ-GUESS is designed to run at; a site-based study was chosen for this paper because of the focus on the evaluation of the energy fluxes against flux tower measurements).*

**Other comments:**

Line 56-57: Please shortly explain the deficiencies of the existing "coupled biosphere-atmosphere regional and global studies" using LPJ-GUESS. This will help the reader to better understand the importance of this work.

*Since some of the deficiencies arising from an indirect coupling approach have been already listed above these lines, we propose modifying the text as follows to stress the point:*

*Lines 49 and following:*

*"DGVMs are frequently*  *integrated into ESMs through an intermediary Land Surface Model (LSM), which facilitates the sub-daily energy, water and gas exchange calculations (e.g. Bonan et al., 2003; Krinner et al., 2005; Smith et al., 2011; Döscher et al., 2021). This is necessary because DGVMs run normally on a daily or longer time step, while atmospheric models may use time steps ranging from seconds to tens of minutes, depending on the required resolution. This indirect approach to coupling can, however, entail inconsistencies between the DGVM and the LSM, such as the use of different time steps and temperatures in photosynthetic calculations, duplicated or inconsistent soil water tracking, or different characterization of vegetation types. One possible important consequence of these inconsistencies is the failure to conserve carbon mass. In this work we modify the LPJ-GUESS DGVM (Smith et al., 2001, 2014) to enable coupling with an atmospheric model without the need for a mediating LSM. LPJ-GUESS simulates a wide range of land-biosphere processes, including vegetation growth, establishment and mortality, plant functional type (PFT) competition, disturbances, wildfires, and land use change. This model has been used in a broad range of applications, including coupled biosphere-atmosphere regional (Wramneby et al., 2010; Smith et al., 2011; Zhang et al., 2014, 2018; Wu et al., 2016, 2021) and global (Weiss et al., 2014; Alessandri et al., 2017; Forrest et al., 2020; Döscher et al., 2021) studies, although these suffer from the above-mentioned limitations of the indirect coupling approach. LPJ-GUESS is maintained by an international developer community and undergoes active development and evaluation, which makes it a suitable choice to study climate-biosphere interactions."*

Line 230: Why set the same optical properties for all the PFTs? There should be data available to help parameterize these parameters for different PFTs.

*We used the same optical properties for all PFTs in order to limit the degrees of freedom and keep the development tractable. In this way we can introduce them at a later stage and assess the impact of this change more easily (we note that in standard LPJ-GUESS PFTs do not have different optical properties).*

*We suggest adding:*

*"In* *order to keep the model development process tractable, we set the optical properties of the canopy to the following values, regardless of PFT:"*

Line 236-237: Have you considered the effect of soil moisture on soil optical properties?

*Yes. In order to clarify this point, we suggest replacing the sentence (line 236)*

*"Soil optical properties are from the dataset prepared by Lawrence and Chase (2007)."*

*with*

*"Soil albedo is calculated from the soil dry and moisture-saturated reflectances and the water content of the top soil layer following Oleson et al. (2010). Soil color classes are from Lawrence and Chase (2007), and were obtained from the dataset included in the CLM4.0 code (Lawrence et al., 2011)."*

Equation 47: "Vsun,day" should be "V max, sun, day"?

*We thank the reviewer for spotting the mislabeling in the equation. We will correct it in the revised manuscript.*

Line 328: Does heat capacity also depend on organic matter content?

*Soil heat capacity does not depend on organic matter content in this version of the model. We plan to introduce this dependency at a later stage in the development. In order to clarify this, we suggest adding the following text (line 327):*

*"Soil heat capacity is computed as a weighted sum of the heat capacities of the dry soil, which depends on texture, and water (de Vries, 1963). Soil organic matter does not contribute to soil heat capacity in the current version of the model."*

Line 359-360: Please explain in more details why such overestimation happens?

*The overestimation happens because the potential evaporation rates are calculated on the basis of the current canopy temperature in a given iteration. At the end of the iteration, with the updated canopy temperature, the energy fluxes are recalculated. We found that this recalculation before proceeding to the next iteration gives stability to the numerical scheme and reduces the number of iterations. Since fluxes are recalculated at the updated temperature, but evapotranspiration is partitioned between canopy evaporation and transpiration on the basis of the conductances derived from the potential evaporation rates before updating the temperatures, a mismatch happens and the canopy may not hold enough water to meet the newly calculated evaporation demand.*

*In order to correct this error, we simply recalculated the leaf conductances at the found energy balance temperatures and humidities just before exiting the energy balance iteration and repeated the simulations. This is the change in the code:*

```
// BEFORE:
while (!exit_loop) {
    …
    // If we are close to energy balance or niter >= NITERMAX -> exit loop
    if ((norm2(f,3) < 1.e-4) || distance < 1.e-3 || niter >= NITERMAX) {
        exit_loop = true;
    }
}

// AFTER:
while (!exit_loop) {
    …
    // If we are close to energy balance or niter >= NITERMAX -> exit loop
    if ((norm2(f,3) < 1.e-4) || distance < 1.e-3 || niter >= NITERMAX) {
        // Recalc conductances at new equilibrium temp and q
        double evap_pot_sha = -climate.rhoair*(patch.q_ca - q[SHA])/rb;
        double evap_pot_sun = climate.isdaytime ?
                              -climate.rhoair*(patch.q_ca - q[SUN])/rb : 0.;
        leaf_conductances_patch(climate, patch, ga_h, rab, rb, evap_pot_sha,
                                evap_pot_sun);
        exit_loop = true;
    }
}
```

*This recalculation made this error disappear. Water is now conserved on average to a 5x10^-12% of the precipitation input. This modification also improved the energy closure error slightly. Since the water conservation error is so small, we removed the second histogram on Fig. 5, which now is:*

[Figure]

Line 375-377: What time step of LPJ-GUESS is used in the simulation? Half hour or 1 hour or 3 hour? Why do hourly averaging for the forcing data instead of using the original half-hourly forcing data which might be more physically consistent for different forcing variables.

*We used a time step of 1h in the LSM runs. Since some of the sites provide only hourly averages of the climatology, we decided to average half-hourly data in order to have a consistent input time step across sites. We believe, however, that this choice should not affect the results significantly.*

*We suggest adding the following text to clarify the time step used in the simulations (line 375):*

*"We used the climate data collected at the tower sites to force the model. Half-hourly forcing data was converted to hourly averages to use a fixed time step of 1 hour in all simulations."*

Section 3.2: It is not clearly explained how soil properties were set for each site simulation.

*In section 2.2.5 (Soil Physics) we state:*

*(Line 327): "Soil heat capacity is computed as a weighted sum of the heat capacities of the dry soil, which depends on **soil texture**, and water (de Vries, 1963)."*

*(Line 333): "Hydraulic diffusivity and conductivity are calculated as a function of **soil texture** and soil water content by using the expressions derived by Clapp and Hornberger (1978) and Cosby et al (1984)."*

*To clarify where the the texture data come from, we suggest inserting the following text in line 378:*

*"Nitrogen deposition data is from Lamarque et al. (2013). Atmospheric CO2 concentration data is from McGuire et al. (2001). The soil texture data used to calculate soil hydraulic and thermal properties (as described in Sec. 2.2.5) at each site were as in Sitch et al. (2003), based on The Digitized Soil Map of the World (Zobler, 1986; FAO, 1991)."*

Figure 7-13: please add (a, b, c, d, e ….) for each subplot.

*The plots were corrected as suggested and will be included in the revised version of the manuscript.*

Line 422: "dry season": please specify which months.

*The text will be changed in the revised manuscript as follows:*

*"At the AU-DaS site (Fig. 7, upper right panel), the shapes of measured and simulated annual cycles match relatively well at the beginning and the end of the year, but diverge substantially during the dry season (July-November)."*

*Additionally, we suggest to include, for reference (to also address the next comment about the 'systematic overestimation of turbulent fluxes'), the average precipitation and the net radiation in the plot:*

[Figure]

Line 434-435: Where do "the systematic overestimation of sensible and latent heat fluxes" (i.e., excess energy) come from in the model?

*At this site, the model overestimates net (absorbed) radiation between March and May, but between June and November modeled absorbed radiation is very close to measurements. Since energy is balanced in the model, the excess energy in turbulent fluxes must be compensated by an underestimation of heat conducted into the ground. One possibility is that simulated upper soil moisture is lower than actual soil moisture at the site, which would lead to an underestimation of upper soil thermal conductivity in the model. Unfortunately, soil moisture data are not available for this site, so we could not to test this hypothesis. We suggest adding the following text to address this point in the paper:*

*Lines 434-435: "Sensible and latent heat fluxes are systematically overestimated by the model by ~ 10–20 Wm$^{-2}$. This overestimation takes place even when simulated net radiation is very close to observations (June to November), so it must be compensated by an underestimation of ground heat, possibly caused by an underestimation of upper soil moisture. Unfortunately, soil moisture measurements are not available for this site, so we were not able to test this hypothesis."*

[Figure]

Line 443: Please add (measured value) after 200 Wm-2.

*We will add it in the revised manuscript.*

Line 388-389: Please show the spin-up plot (e.g., in supplement). How does the new LSM version of LPJ-GUESS affect the C and N cycle in soil?

*The spinup plots are now included and briefly discussed in a supplement (see attached document).*

Section 3.4.4: I am wondering if this section should be moved to the front as section 3.4.1, so that the readers could have a rough picture about the vegetation cover in each simulation. Please also explain how much soil decomposition affects NEE in the LSM version of LPJ-GUESS. The whole section should be shortened.

*We added a new plot to the section showing the evolution of the PFT composition over the spinup period at selected sites:*

[Figure]

*We rewrote the whole section (see below) and brought the word count from ~1600 words down to ~1240, including a new paragraph explaining the new figure. The influence of the choice of stomatal conductance scheme and soil water uptake function is briefly discussed. We briefly discuss how carbon fluxes may be affected by the new schemes:*

*"The new physical schemes introduced in this work lead to discrepancies between the LSM and standard LPJ-GUESS, which stem from the different physical environments simulated in the models. Calculating assimilation at the newly simulated canopy temperature, rather than the air temperature, can lead to either higher or lower productivity, depending on the optimal photosynthetic temperature ranges of each PFT and the effect of temperature on N limitation (see Sec. 4.2). Canopy temperature also affects autothrophic respiration, while differences in the simulated soil humidity and temperature*

*impact organic matter decomposition rates and heterotrophic respiration. The combination of these effects results in differences in the simulated equilibrium carbon and nitrogen pools (see supplement) and ecosystem-atmosphere carbon fluxes."*

*We agree with the reviewer that moving the subsection section up to the beginning of the results section makes sense, and will make this change in the revised manuscript.*

Line 553 and 560: remove "somewhat".

*We will remove it in the revised manuscript.*

*NEW REFERENCES*

[revised manuscript text omitted]
 PAR absorbed by the vegetation, calculated using the new radiative transfer scheme and the PAR absorption scheme in standard LPJ-GUESS. Data are from the CLM/Med simulations described in the paper. PAR values are averages over the measurement period of the simulations, in MJ/year/m$^2$. The percent change is relative to the standard LPJ-GUESS run.

**1 Differences in PAR absorption between LPJ-GUESS and LPJ-GUESS/LSM**

Table 1 shows a comparison of average PAR absorption per unit LAI calculated by the new radiative transfer scheme and the PAR absorption algorithm in stadard LPJ-GUESS. The calculations were made in the CLM/Med simulation, i.e., PAR absorption is calculated with both schemes in the same modeled areas for the purpose of this comparison. In general, the new radiative transfer calculates higher absorbed PAR values than standard LPJ-GUESS at sites with low modeled LAI values, while both calculations yield similar results at sites with high LAI values. This behaviour can be understood by examining PAR absorption by individual cohorts. Figure 1 shows PAR absorption by the vegetation over 60 years during the spinup period at BR-Sa1, starting after a disturbance. Three tree cohorts (0, 1 and 2) and a grass individual (4) establish. Initially, grass has a high LAI, but, as trees grow and the canopy thickens, the grass LAI declines (panels $c$ and $d$). Calculated tree PAR absorption per leaf area is initially similar for both schemes (panel $a$), but as cohort 0 grows it shadows cohorts 1 and 2. The new radiative transfer scheme calculates lower PAR values for these two cohorts, but since their leaf area index is also declining, this does not contribute substantially to the overall difference, which is small and dominated by cohort 0 (panel $b$).

Figure 2 shows the same comparison for a patch at AU-Gin. In this case,

[Figure]

Figure 1: Comparison of PAR absorbed by the cohorts in a patch at BR-Sa1, calculated using the new radiative transfer scheme and the standard LPJ-GUESS PAR absorption scheme. *(a):* Annual absorbed PAR per leaf area; *(b):* Percent change in PAR absorption relative to standard LPJ-GUESS; *(c):* LAI; *(d):* Cohort height.

the tree cohorts have a lower leaf area index, so their leaves receive, on average, more direct sunlight than in the case of a thicker canopy. The new radiative scheme calculates higher values of absorbed PAR for these cohorts (panels *a* and *b*), and this feature dominates the overall difference between the two schemes in this site.

**2 Spinup information**

In a standard LPJ-GUESS simulation the 500-year spinup process proceeds as follows: the first 100 years, the model runs without nitrogen uptake to allow build up of soil nitrogen pools. All vegetation in the patch is then reset, and plant nitrogen uptake is turned on. Between years 140 and 220, information on the rates of change of C and N pools is collected. This information is then used

[Figure]

Figure 2: Comparison of PAR absorbed by the cohorts in a patch at BR-Sa1, calculated using the new radiative transfer scheme and the standard LPJ-GUESS PAR absorption scheme. *(a):* Annual absorbed PAR per leaf area; *(b):* Percent change in PAR absorption relative to standard LPJ-GUESS; *(c):* LAI; *(d):* Cohort height.

to calculate carbon and nitrogen steady-state pool sizes analitically, assuming an equilibration time of 40000 years for the soil organic matter pools. The model then runs for another 280 years, a period considered long enough for the vegetation C and N pools to reach steady state.

In general, the steady-state size of the carbon and nitrogen pools is determined by the balance between the rate of carbon input to the system (NPP) and the turnover rates of the soil organic matter pools. The LSM implementation changes the physical environment at which these processes take place in the model. Calculating photosynthesis rates at the newly simulated leaf temperature can lead to higher or lower carbon assimilation, depending on the PFT's optimal photosynthetic temperature range. It can also boost productivity by mitigating the effect of N limitation (see paper, Section 4.2). Soil organic matter decomposition is affected by soil temperature and humidity; higher (lower) tem-

| | BB | | | | | Med | | | | |
|---|---|---|---|---|---|---|---|---|---|---|
| | NPP | Temp | AWC | Soil C | Soil N | NPP | Temp | AWC | Soil C | Soil N |
| AU-Emr | -66.7 | 13.1 | -30.0 | -74.1 | -74.1 | -68.7 | 14.4 | -49.6 | -75.3 | -75.3 |
| ES-Amo | 6.1 | 3.3 | -11.6 | -12.9 | -12.9 | 0.6 | 3.7 | -29.9 | -12.9 | -12.7 |
| AU-DaP | 1.8 | 4.5 | 46.2 | -12.8 | -10.2 | 25.5 | 4.4 | 19.8 | 21.7 | 25.7 |
| AU-Stp | -42.8 | 8.5 | -24.1 | -50.5 | -50.4 | -23.0 | 9.0 | -37.0 | -33.6 | -33.4 |
| CG-Tch | 87.6 | 2.5 | 51.5 | 6.8 | 9.2 | 92.0 | 1.8 | 52.1 | 10.4 | 12.9 |
| PA-SPs | 34.7 | 1.2 | 6.7 | 22.2 | 24.3 | 38.9 | 1.2 | 3.5 | 27.2 | 29.4 |
| AU-DaS | -11.8 | 0.9 | 42.1 | -0.9 | 3.5 | -6.5 | 1.7 | 16.5 | 15.9 | 21.3 |
| AU-Dry | -7.1 | 3.8 | 47.1 | -2.0 | 1.0 | 1.4 | 3.3 | 11.8 | 4.5 | 6.5 |
| SD-Dem | -13.1 | -0.4 | 85.0 | -45.9 | -47.2 | 34.8 | -0.3 | -0.0 | 43.6 | 46.6 |
| AU-Ade | -13.4 | 0.8 | 34.9 | 8.1 | 15.6 | -8.7 | 0.8 | 16.6 | 21.4 | 29.5 |
| AU-Gin | 0.6 | 6.2 | 56.3 | -32.7 | -32.3 | -9.1 | 6.0 | 25.7 | -25.3 | -23.6 |
| AU-How | -13.8 | -0.5 | 37.9 | 2.9 | 10.2 | -10.6 | -0.0 | 20.8 | 18.6 | 27.8 |
| AU-RDF | 4.6 | 3.3 | 45.4 | 15.0 | 19.9 | 12.0 | 3.6 | 21.0 | 28.7 | 33.1 |
| AU-Rob | 4.4 | 1.1 | 18.3 | -6.4 | -6.2 | 3.3 | 1.1 | 13.0 | -2.1 | -1.4 |
| BR-Sa1 | -25.2 | -0.5 | 8.1 | -16.0 | -15.3 | -20.5 | -0.6 | 5.7 | -14.2 | -13.3 |
| BR-Sa3 | -11.1 | -2.4 | 5.6 | -8.7 | -8.4 | -6.2 | -2.5 | -3.2 | -6.5 | -6.3 |
| GF-Guy | -14.6 | 0.3 | 9.9 | -14.7 | -14.5 | -11.3 | 0.6 | 6.0 | -12.6 | -12.3 |
| GH-Ank | -23.9 | -0.3 | 13.3 | -15.6 | -13.4 | -18.8 | -0.7 | 11.4 | -13.8 | -11.5 |
| MY-PSO | -30.1 | 0.3 | 59.2 | -43.1 | -42.6 | -23.2 | 0.5 | 54.4 | -37.5 | -37.4 |
| PA-SPn | -15.8 | 0.6 | 8.5 | -20.6 | -18.5 | -12.1 | 0.7 | 5.5 | -16.0 | -13.6 |
| ZM-Mon | -1.6 | 3.9 | 68.9 | -18.1 | -12.9 | -3.8 | 1.8 | 39.5 | -20.1 | -15.3 |

Table 2: Percent change in steady-state NPP, average soil temperature over the top 50 cm of soil, average water content over the top 50 cm of soil, soil carbon content, and soil nitrogen content, relative to standard LPJ-GUESS. Steady state values are taken as the average of the last 100 years of spinup.

peratures and humidities lead to higher (lower) turnover rates. Table 2 shows a comparison of these factors in LSM and standard LPJ-GUESS simulations for all the sites considered in this study.

We show two examples of the build-up of the soil organic matter pools at BR-Sa1 (Fig. 3) and SD-Dem (Fig. 4), for the standard LPJ-GUESS, the CLM/BB, and the CLM/Med runs. At BR-Sa1 in the BB simulation, equilibrium NPP is lower than in standard LPJ-GUESS by $\sim 25\%$ (Table 2). Soil temperature is similar to standard LPJ-GUESS, but soil moisture is $\sim 8\%$ larger. This leads to lower equilibrium soil carbon ($\sim -16\%$) and nitrogen ($\sim -15\%$) content. The CLM/Med simulation behaves similarly at this site (and at most forest sites).

At SD-Dem the BB and Med simulations show very different behaviours. In the BB simulation, NPP is lower than in LPJ-GUESS, while the higher stomatal resistance given by the Ball-Berry scheme (see paper, Fig. 3) causes higher soil moisture content. This leads to lower equilibrium soil organic matter content values (a $\sim 46\%$ decrease compared to standard LPJ-GUESS). In the Med simulation, equilibrium NPP is substantially higher than in the standard LPJ-GUESS run, while lower soil moisture retention leads to slower decomposition rates, resulting in soil organic matter pools $\sim 44\%$ larger than in standard LPJ-GUESS.

[Figure]

Figure 3: Comparison of the build up of carbon and nitrogen pools in the CLM/BB *(a) and (b)* and the CLM/Med *(c) and (d)* simulations with standard LPJ-GUESS, at BR-Sa1.

[Figure]

Figure 4: Comparison of the build up of carbon and nitrogen pools in the CLM/BB *(a) and (b)* and the CLM/Med *(c) and (d)* simulations with standard LPJ-GUESS, at SD-Dem.